# In-Context Learning Is Provably Bayesian Inference: A Generalization Theory for Meta-Learning

**Tomoya Wakayama** [1]  **Taiji Suzuki** [1][2]

## Abstract

This paper develops a finite-sample statistical theory for in-context learning (ICL), analyzed within a meta-learning framework that accommodates mixtures of diverse task types. We leverage a Bayes risk identity that separates the total ICL risk into two orthogonal components: Bayes Gap and Posterior Variance. The Bayes Gap quantifies how well the trained model approximates the Bayes-optimal in-context predictor. For a uniform-attention Transformer, we derive a non-asymptotic upper bound on this gap, which explicitly clarifies the dependence on the number of pretraining prompts and their context length. The Posterior Variance is a model-independent risk representing the intrinsic task uncertainty. Our key finding is that this term is determined solely by the difficulty of the true underlying task, while the uncertainty arising from the task mixture vanishes exponentially fast with only a few in-context examples. Together, these results provide a unified view of ICL: the uniform-attention Transformer selects the optimal meta-algorithm during pretraining and rapidly converges to the optimal algorithm for the true task at test time.

## 1. Introduction

Large language models (LLMs) have moved far beyond classic NLP benchmarks into complex, real-world workflows (Naveed et al., 2024; Zhao et al., 2025) such as code assistants and generators (GitHub, 2025) in software engineering, Med-PaLM 2 (Singhal et al., 2025) in healthcare, text-to-SQL systems (Gao et al., 2024; Shi et al., 2024) in business

[1]RIKEN Center for Advanced Intelligence Project (AIP), Tokyo, Japan [2]Department of Mathematical Informatics, Graduate School of Information Science and Technology, The University of Tokyo, Tokyo, Japan. Correspondence to: Tomoya Wakayama <tomoya.wakayama@riken.jp>.

*Proceedings of the 43rd International Conference on Machine Learning*, Seoul, South Korea. PMLR 306, 2026. Copyright 2026 by the author(s).

intelligence, and vision-language-action models (Kim et al., 2024b; Zitkovich et al., 2023) in robotics. In particular, since GPT-3, modern LLMs have demonstrated a striking ability to adapt to new tasks from only a handful of input-output exemplars, without parameter updates (Brown et al., 2020). This phenomenon, known as in-context learning (ICL), appears across diverse datasets and task formats and is at the heart of these workflows (Min et al., 2022; Dong et al., 2024). These deployments share common constraints: inference-time (test-time) prompts are short, and upstream pretraining covers heterogeneous task types. A concrete, finite-sample account of predictive error under these constraints is therefore of key importance to practitioners.

Numerous studies investigate the behavior of ICL. Wang et al. (2023); Akyürek et al. (2023); von Oswald et al. (2023); Li et al. (2023); Bai et al. (2023); Garg et al. (2022); Mahankali et al. (2024) have empirically or theoretically shown that (simplified) Transformers can implement canonical estimators and learning procedures in context (e.g., least squares, ridge and gradient-descent steps), sometimes achieving near-Bayes-optimal performance on linear tasks. Concurrently, Jeon et al. (2024) provide an information-theoretic analysis, and Kim et al. (2024a) present non-parametric rates for a linear-attention Transformer, with subsequent progress (Wang et al., 2024; Oko et al., 2024; Nishikawa et al., 2025). A compelling perspective frames ICL as a form of implicit Bayesian inference (Xie et al., 2022; Wang et al., 2023; Panwar et al., 2024; Arora et al., 2025; Reuter et al., 2025; Zhang et al., 2025). Although this viewpoint provides an explanatory framework for ICL's capabilities, the aforementioned theories have not fully leveraged the theoretical relationship between ICL and Bayes. Hence, they lack a statistical theory that can (i) jointly couple pretraining size $N$ and prompt length $p$ and (ii) accommodate heterogeneous mixtures of task types, a regime common in LLM pretraining.

We adopt a Bayes-centric framework that offers a concrete account of the sources of error and clarifies how they shrink with $p$ and $N$. Specifically, viewing ICL risk as the Bayes risk (e.g., §5.3.1.2 of Murphy, 2022), we treat the Bayes-optimal predictor as the optimal in-context predictor and leverage the following classical identity under squared loss

(Proposition 3.1):

$$\text{ICL risk} = \text{Bayes Gap} + \text{Posterior Variance},$$

where the *Bayes Gap* measures the discrepancy between a pretrained model and the optimal in-context (Bayes) predictor, and the *Posterior Variance* is independent of the model and shrinks as the observed context length grows. Conceptually, performance limits at inference time are governed by Bayesian uncertainty about the test task (i.e., the task at inference time), not by pretraining alone. Building on this view, we make the following contributions:

1. **Provide non-asymptotic upper bounds that couple the number of pretraining prompts $N$ and their context length $p$ (Theorem 3.2).** For uniform-attention Transformers, we leverage sequential learning theory (Rakhlin et al., 2010), develop optimal transport-based approximation theory, and then obtain (ignoring logarithmic factors)

$$\mathbb{E}R_{\text{BG}}(M_{\hat{\theta}}) \lesssim \underbrace{m^{-2\alpha/d_{\text{eff}}}}_{\text{approximation}} + \underbrace{m(pN)^{-1} + N^{-1}}_{\text{pretraining generalization}}.$$

Here $m$ is the number of learned features in the Transformer, $d_{\text{eff}}$ is the effective dimension, and $\alpha$ is a Hölder exponent. The rate $\propto m/(pN)$ clarifies the dependence on both $p$ and $N$, which earlier theories on ICL (Kim et al., 2024a; Wu et al., 2024; Zhang et al., 2024) have not fully captured. Importantly, the result suggests that **uniform-attention Transformers select a Bayes-optimal meta-algorithm during pretraining**.

2. **Explain in-context error via the test-task difficulty (Theorem 3.3).** In a mixture of task types, the posterior over the task index concentrates exponentially fast with respect to the observed context length, and the irreducible term $R_{\text{PV}}$ is upper bounded by the minimax risk of the test (true) task family. Without assuming specific algorithms (Akyürek et al., 2023; Bai et al., 2023; Zhang et al., 2024), our result implies that even in mixed-task settings **the Bayes-optimal meta-algorithm rapidly converges to the optimal algorithm for the true task at inference time**. This finding is consistent with empirical reports (Panwar et al., 2024; Arora et al., 2025), which show that ICL often behaves like Bayesian inference, particularly in task-mixture settings.

3. **Characterize stability under input-distribution shift (Theorem 3.4).** We demonstrate that under input-distribution shift from pretraining data to inference-time prompt, the Bayes Gap incurs an out-of-distribution (OOD) penalty proportional to the Wasserstein distance between the distributions, while the Posterior Variance is intrinsic to the target domain. Zhang et al. (2024) have noted that ICL is vulnerable to input-distribution shift in some settings, whereas our results specifically show

that only the Bayes Gap increases in proportion to the magnitude of the shift.

The paper is organized as follows. Section 2 formalizes the meta-learning prompt model, introduces a uniform-attention Transformer, and states assumptions, followed by a primer on the Bayes-optimal in-context predictor. Section 3 presents the risk decomposition and then analyzes (i) the Bayes Gap (Section 3.1), (ii) the Posterior Variance (Section 3.2), and (iii) OOD stability under input-distribution shift (Section 3.3). Section 4 provides the numerical experiments, supporting our theories. Section 5 concludes with limitations and future work. The Appendix contains a list of notation, experimental details, all technical proofs, auxiliary lemmas, and extended discussions.

**Related Work**

*(A) ICL as Bayesian inference.* ICL has been framed as (implicit) Bayesian inference under structured pretraining. Xie et al. (2022) show that mixtures of hidden Markov model-style documents enable Transformers to perform posterior prediction; Panwar et al. (2024) show that Transformers mimic Bayes across task mixtures. Lin & Lee (2024) reconcile task retrieval versus task learning with a probabilistic pretraining model. Wang et al. (2023) view LLMs as latent-variable predictors enabling principled exemplar selection. Reuter et al. (2025) empirically show full Bayesian posterior inference in-context, and Arora et al. (2025) demonstrate Bayesian scaling laws predicting many-shot reemergence of suppressed behaviors. Our results explicitly use Bayesian properties for ICL theory and provide a concrete non-asymptotic validation both in pretraining and at inference time. Note that Ma et al. (2025) independently and concurrently analyze Bayesian adaptivity for softmax-attention Transformers, while our theory focuses on the uniform-attention class and derives explicit $p$–$N$ scaling, minimax-risk reduction, and prompt-level Wasserstein OOD stability.

*(B) ICL as Meta-Learning.* ICL is widely understood as meta-learning (Brown et al., 2020). Transformers implement gradient-descent-style updates within their forward pass, acting as meta-optimizers that perform implicit fine-tuning (von Oswald et al., 2023; Dai et al., 2023). Models can be meta-trained to execute general-purpose in-context algorithms across tasks (Kirsch et al., 2022). From a learning-to-learn perspective, ICL's expressivity explains few-shot strength while exposing generalization limits (Wu et al., 2025). Beyond single tasks, meta-in-context learning shows recursive adaptation of ICL strategies without parameter updates (Coda-Forno et al., 2023). From this perspective, we theoretically clarify how ICL identifies the task at inference time and solves the true task.

## 2. Problem Setup

### 2.1. Meta-Learning: Mixture of Regression Types

We consider a meta-learning framework that accommodates a finite number of distinct task types (task families), which reflects heterogeneous pretraining prompts.

**Definition 2.1** (Prompt-Generating Process)**.** The data-generating process for prompts proceeds as follows:

1. Sample a task type: $I \sim \mathcal{P}_I = \mathrm{Categorical}(\boldsymbol{\alpha})$, i.e., $\Pr(I = i) = \alpha_i > 0$ for $i = 1, \ldots, T$.

2. Given $I = i$, sample a task function: $f \sim \mathcal{P}_{F_i}$, where $\mathcal{P}_{F_i}$ is a distribution on the $i$-th function space $F_i \subset \{f : \mathbb{R}^{d_{\mathrm{feat}}} \to \mathbb{R}\}$.

3. For $k = 1, \ldots, p + 1$:
   - Sample an $\mathbb{R}^{d_{\mathrm{feat}}}$-dimensional input: $\boldsymbol{x}_k \stackrel{\mathrm{i.i.d.}}{\sim} \mathcal{P}_X$.
   - Generate output: $y_k = f(\boldsymbol{x}_k) + \varepsilon_k$, where $\varepsilon_k \stackrel{\mathrm{i.i.d.}}{\sim} \mathcal{P}_\varepsilon$ is sub-Gaussian random noise with $\mathbb{E}[\varepsilon_k] = 0$, $\mathrm{Var}(\varepsilon_k) = \sigma_\varepsilon^2$, and $\varepsilon_k \perp (f, \boldsymbol{x}_k)$.

4. Form the length-$p$ (complete) prompt: $P = (\underbrace{\boldsymbol{x}_1, y_1, \ldots, \boldsymbol{x}_p, y_p,}_{\text{context } D^p} \underbrace{\boldsymbol{x}_{p+1}}_{\text{query}})$.

This setting allows for a mixture of $T\,(< \infty)$ different task types (task families), such as linear regression type $F = \{\boldsymbol{x} \mapsto \boldsymbol{w}^\top \boldsymbol{x} + b\}$ and basis-function regression type $F = \{\boldsymbol{x} \mapsto \sum_{j=0}^R a_j g_j(\boldsymbol{x})\}$, where $g_j$ are, for example, Hermite polynomials. Note that Step 1 of Definition 2.1 selects the task family $F_i$ (via $I$), and Step 2 samples a particular function $f$ from that family; in the linear-regression case, this corresponds to choosing coefficients such as $\boldsymbol{w}$ and $b$. While many existing ICL theories (e.g., Zhang et al., 2024; Kim et al., 2024a; Oko et al., 2024) focus on i.i.d. prompts from a single task family, our setting is more general.

A length-$k$ partial prompt[1] is denoted by $P^k = (\boldsymbol{x}_1, y_1, \ldots, \boldsymbol{x}_k, y_k, \boldsymbol{x}_{k+1})$ and its context dataset (examples) by $D^k = \{(\boldsymbol{x}_j, y_j)\}_{j=1}^k \in \mathbb{R}^{k d_{\mathrm{eff}}}$, where $d_{\mathrm{eff}} := d_{\mathrm{feat}} + 1$. We fix a maximum context length $p$. At inference time, after observing $k \le p$ examples, we sequentially evaluate the risk of predicting $y_{k+1}$ from $P^k$.

### 2.2. Transformer Architecture

We begin by briefly reviewing the standard Transformer architecture. The standard Transformer (Vaswani et al., 2017) processes sequences through self-attention mechanisms: $\mathrm{Attention}(Q, K, V) = \mathrm{softmax}\big(\frac{QK^\top}{\sqrt{d_k}}\big)V$, where

queries $Q$, keys $K$, and values $V$ are linear projections of the input embeddings. Each Transformer layer consists of self-attention and a position-wise feed-forward network.

In this work, we adopt a specialized uniform-attention ($Q = K = 0$) Transformer architecture. Conditional on the task function (Definition 2.1), the examples in each prompt are i.i.d. and hence exchangeable. Thus, a permutation-invariant mechanism like uniform attention is sufficient, which motivates our choice of the following architecture. Further justification is provided in Appendix C.

**Definition 2.2** (Uniform-attention Transformer Architecture)**.** We study a uniform-attention (mean-pooling) Transformer of the form:

$$M_\theta(P^k) := \rho_\theta\Big( \frac{1}{k} \sum_{i=1}^k \phi_\theta(\boldsymbol{x}_i, y_i), \boldsymbol{x}_{k+1} \Big).$$

Here, the feature encoder $\phi_\theta : \mathcal{U} \to \Delta^{m-1}$ and the decoder $\rho_\theta : \Delta^{m-1} \times \mathcal{C} \to \mathbb{R}$, where $\Delta^{m-1}$ denotes the $(m-1)$-dimensional probability simplex, $\mathcal{U}$ denotes the example domain (the space of $(\boldsymbol{x}_i, y_i) \in \mathbb{R}^{d_{\mathrm{eff}}}$) and $\mathcal{C}$ denotes the query domain (the space of $\boldsymbol{x}_{k+1}$), have the following structures:

*Feature Encoder Network $\phi_\theta$:* The feature encoder consists of a depth-$D_\phi$ feedforward ReLU network followed by a renormalization layer:

$$\phi_\theta(\boldsymbol{x}, y) := \mathrm{Renorm}_\tau \circ g_\theta(\boldsymbol{x}, y),$$
$$g_\theta(\boldsymbol{u}) := W^{(D_\phi)} \sigma\big( \cdots \sigma\big(W^{(1)}\boldsymbol{u} + \boldsymbol{b}^{(1)}\big) \cdots \big) + \boldsymbol{b}^{(D_\phi)},$$

where $\boldsymbol{u} = [\boldsymbol{x}^\top, y]^\top \in \mathbb{R}^{d_{\mathrm{eff}}}$, $\sigma(\cdot) = \max\{0, \cdot\}$ is the ReLU activation applied element-wise, $W^{(\ell)} \in \mathbb{R}^{n_\ell \times n_{\ell-1}}$ are weight matrices with $n_0 = d_{\mathrm{eff}}$ and $n_{D_\phi} = m$, and the renormalization layer is defined as $\mathrm{Renorm}_\tau(\boldsymbol{s}) = \frac{\sigma(\boldsymbol{s}) + \frac{\tau}{m}\mathbf{1}}{\mathbf{1}^\top \sigma(\boldsymbol{s}) + \tau}$ $(\tau \in (0, 1])$. This ensures $\phi_\theta(\boldsymbol{x}, y) \in \Delta^{m-1}$.

*Decoder Network $\rho_\theta$:* The decoder is a depth-$D_\rho$ feedforward ReLU network that jointly processes the aggregated features and query:

$$\rho_\theta(\boldsymbol{z}, \boldsymbol{c}) := \mathrm{clip}_{[-B_M, B_M]}\big(h_\theta(\boldsymbol{z}, \boldsymbol{c})\big),$$
$$h_\theta(\boldsymbol{v}) := W^{(D_\rho)} \sigma\big( \cdots \sigma\big(W^{(1)}\boldsymbol{v} + \boldsymbol{b}^{(1)}\big) \cdots \big) + b^{(D_\rho)},$$

where $\boldsymbol{v} = [\boldsymbol{z}^\top, \boldsymbol{c}^\top]^\top \in \mathbb{R}^{m + d_{\mathrm{feat}}}$, $W^{(\ell)} \in \mathbb{R}^{n_\ell \times n_{\ell-1}}$ with $n_0 = m + d_{\mathrm{feat}}$ and $n_{D_\rho} = 1$, and the clipping operation ensures $|M_\theta(P^k)| \le B_M$.

*Size of the Networks:* Throughout, $\|\cdot\|_2$ denotes the Euclidean norm for vectors and the spectral norm for matrices. For depth-$D$ ReLU network $\mathcal{T}_\theta$, define the spectral product $S(\mathcal{T}_\theta) := \prod_{d=1}^D \big\|W^{(d)}\big\|_2$. There exist fixed

---

[1] In standard statistical terminology, a complete prompt corresponds to a dataset together with a new query input, a partial prompt corresponds to a prefix of the dataset with the next query, and a task family corresponds to a model/function class equipped with a prior.

constants $C_\phi$, $C_\rho > 0$ (independent of $p, N$) such that $S(\phi_\theta) \leq C_\phi m^{1/d_{\text{eff}}}$ and $S(\rho_\theta) \leq C_\rho m^{1/2}$. For the feature encoder $\phi_\theta$, we assume that a depth of $D_\phi = O(\log m)$ and the number of trainable parameters is $O(m \log m)$. Finally, let $\Theta$ denote the parameter space that satisfies these conditions.

## 2.3. Risk, Training, and Assumptions

Throughout, we use the squared loss $\ell(u, v) = (u - v)^2$. The *ICL risk* of a predictor $M$ is defined as the expected loss averaged over sequence lengths $k = 1, \ldots, p$ under the generative process in Definition 2.1:

$$R(M) = \frac{1}{p} \sum_{k=1}^{p} \mathbb{E}_{I,f,D^k,\boldsymbol{x}_{k+1}} \left[ \ell\big(f(\boldsymbol{x}_{k+1}), M(P^k)\big) \right],$$

where the expectation is taken over the joint distribution of the task $I \sim \mathcal{P}_I$, the function $f \sim \mathcal{P}_{F_I}$, the context $D^k \sim \mathcal{P}_{X,Y|f}^{\otimes k}$ (where $\mathcal{P}_{X,Y|f}$ denotes the joint distribution conditional on $f$), and the query $\boldsymbol{x}_{k+1} \sim \mathcal{P}_X$. In practice, pretraining is performed using a dataset of $N$ prompts of length $p$. The empirical risk minimizer (ERM) is given by

$$\hat{\theta} = \arg\min_{\theta \in \Theta} \frac{1}{pN} \sum_{j=1}^{N} \sum_{k=1}^{p} \ell\big(y_{j,k+1}, M_\theta(P_j^k)\big). \quad (1)$$

*Remark* 2.3 (Meta-train/test protocol). The pretraining dataset consists of $N$ i.i.d. prompts $\{P_j\}_{j=1}^{N}$, each generated by first sampling $I_j$, next drawing $f_j$, and then sampling context examples and a query from the same $\mathcal{P}_X$. At inference time, $I^{\text{test}}$ and $f^{\text{test}}$ are drawn from the same mixture, and the risk $R(M)$ is averaged over new prompts from the same meta-distribution.

For the subsequent analysis, we make the following assumptions about the task function and the inputs.

**Assumption 2.4** (Bounded task functions)**.** There exists $B_f > 0$ such that for any $i$ and $f \in F_i$, $|f(\boldsymbol{x})| \leq B_f$ for all $\boldsymbol{x}$ in the support of $\mathcal{P}_X$.

**Assumption 2.5** (Bounded inputs and conditional independence)**.** There exists $B_X < \infty$ such that $\|\boldsymbol{x}\|_2 \leq B_X$, $\mathcal{P}_X$-almost surely. $\{\boldsymbol{x}_k\}_k$ are i.i.d. samples from $\mathcal{P}_X$, and, conditional on a sampled task function $f$, the pairs $\{(\boldsymbol{x}_k, y_k)\}_k$ are conditionally independent across $k$.

## 2.4. Primer on the Bayes-Optimal In-Context Predictor

In this section, we characterize the optimal predictor that minimizes the ICL risk. Since the ICL risk is equivalent to the Bayes risk (e.g., §5.3.1.2 of Murphy, 2022), the theoretically optimal in-context predictor is the Bayes predictor, i.e., the posterior mean of the function value given the context in this setting. We explain this point below.

The ICL risk minimization problem is to find a predictor $M$ that solves $\min_M R(M)$. Using the law of total expectation, we can rewrite $R(M)$ as an expectation over the context $D^k$. For each context, we aim to minimize the conditional expectation of the loss:

$$\min_M \mathbb{E}_{D^k} \Big[ \mathbb{E}_{I,f,\boldsymbol{x}_{k+1}|D^k} \big[ \ell\big(f(\boldsymbol{x}_{k+1}), M(P^k)\big) \big] \Big],$$

where the inner expectation considers $I \sim \mathcal{P}_{I|D^k}$, $f \sim \mathcal{P}_{F_I|D^k}$, and $\boldsymbol{x}_{k+1} \sim \mathcal{P}_X$. To minimize the overall expectation, it suffices to minimize the inner conditional expectation for each fixed context $D^k$. The minimizer is exactly the definition of the *Bayes estimator* (Bernardo & Smith, 1994; Robert, 2007) because the inner conditional expectation is the Bayes risk, which is the expected predictive loss $\mathbb{E}_{\boldsymbol{x}_{k+1} \sim \mathcal{P}_X} \big[ \ell\big(f(\boldsymbol{x}_{k+1}), M(P^k)\big) \big]$ with respect to the *Bayes posterior distribution* $\mathbb{E}_{I \sim \mathcal{P}_{I|D^k}}[\mathcal{P}_{F_I|D^k}]$. Specifically, for the squared error loss, the value $M(P^k)$ that minimizes the conditional mean squared error, $\mathbb{E}_{I \sim \mathcal{P}_{I|D^k}} \mathbb{E}_{f \sim \mathcal{P}_{F_I|D^k}} \mathbb{E}_{\boldsymbol{x}_{k+1} \sim \mathcal{P}_X} \big[ \ell\big(f(\boldsymbol{x}_{k+1}), M(P^k)\big) \big]$, is the Bayes posterior mean (e.g., Murphy, 2022; Lehmann & Casella, 1998). Thus, the optimal predictor $M_{\text{Bayes}}$ that minimizes the ICL risk is the posterior mean:

$$M_{\text{Bayes}}(P^k) := \mathbb{E}_{I \sim \mathcal{P}_{I|D^k}} \mathbb{E}_{f \sim \mathcal{P}_{F_I|D^k}}[f(\boldsymbol{x}_{k+1})]$$
$$\equiv \arg\min_M R(M).$$

This Bayes predictor serves as the theoretical target during pretraining (Figure 1), and the **Bayes Gap**, which we introduce next, measures how well the pretrained model $M_{\hat{\theta}}$ emulates this predictor. Also, the Bayes predictor can be viewed as performing implicit prompt learning (Li & Liang, 2021; Lester et al., 2021): given a context $D^k$, it infers a task-specific representation ($\mathbb{E}_{I \sim \mathcal{P}_{I|D^k}}[\mathcal{P}_{F_I|D^k}]$) which is analogous to a learned prompt in prompt learning.

**Posterior notation.**
Let $\pi_i(D^k) := \Pr(I = i \mid D^k)$ and $\mathcal{P}(f \mid D^k) = \sum_{i=1}^{T} \pi_i(D^k)\mathcal{P}_{F_i}(f \mid D^k, I = i)$. We write the Bayes predictor as $M_{\text{Bayes}}(P^k) = \mathbb{E}_{f \sim \mathcal{P}(f|D^k)}[f(\boldsymbol{x}_{k+1})]$. Throughout, we work on standard Borel spaces so that regular conditional distributions exist, suggesting $\Pr(f \in \cdot \mid D^k)$, $\mathbb{E}[f(\boldsymbol{x}_{k+1}) \mid D^k]$ and $\text{Var}(f(\boldsymbol{x}_{k+1}) \mid D^k)$ are well-defined. Note that as the query $\boldsymbol{x}_{k+1}$ is drawn independently of $f$ and $D^k$, $\mathcal{P}(f \mid P^k) = \mathcal{P}(f \mid D^k)$.

**Permutation invariance of the Bayes predictor.**
For each $k$, we write $\boldsymbol{u}_k = (\boldsymbol{x}_k, y_k) \in \mathcal{U}$ and $\boldsymbol{c} = \boldsymbol{x}_{k+1} \in \mathcal{C}$ and view the Bayes predictor $M_{\text{Bayes}}(P^k)$ as $M_{\text{Bayes}}(\boldsymbol{u}_{1:k}, \boldsymbol{c})$ here. Since the posterior $\mathcal{P}(f \mid D^k)$ depends on $D^k$ only through the multiset $\{(\boldsymbol{x}_i, y_i)\}_{i=1}^{k}$, for any permutation $\pi$ of $\{1, \ldots, k\}$, $M_{\text{Bayes}}(\boldsymbol{u}_{1:k}, \boldsymbol{c}) = M_{\text{Bayes}}(\boldsymbol{u}_{\pi(1)}, \ldots, \boldsymbol{u}_{\pi(k)}, \boldsymbol{c})$. Thus, the Bayes predictor is a symmetric set functional, which justifies using the

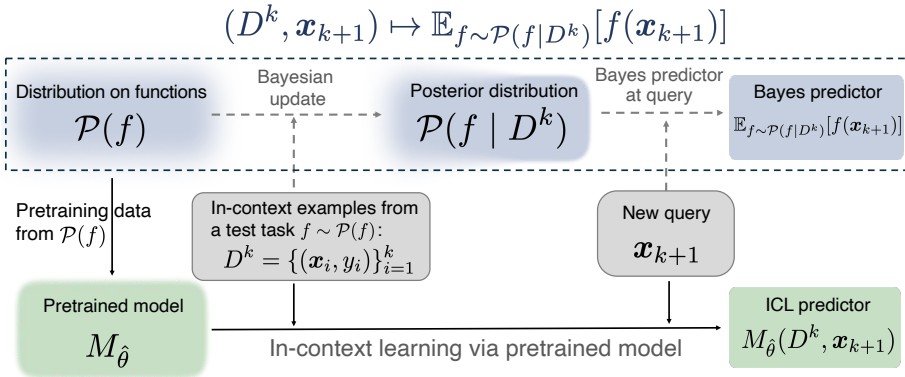

$$(D^k, \boldsymbol{x}_{k+1}) \mapsto \mathbb{E}_{f \sim \mathcal{P}(f|D^k)}[f(\boldsymbol{x}_{k+1})]$$

*Figure 1.* Bayesian view of ICL. The upper path: the process of computing the optimal prediction is $(D^k, \boldsymbol{x}_{k+1}) \mapsto \mathbb{E}_{f \sim \mathcal{P}(f|D^k)}[f(\boldsymbol{x}_{k+1})]$ given $\mathcal{P}(f)$. The lower path: since $\mathcal{P}(f)$ is unknown, the model $M_{\hat{\theta}}$, pretrained on data from $\mathcal{P}(f)$, aims to emulate this process via $(D^k, \boldsymbol{x}_{k+1}) \mapsto M_{\hat{\theta}}(D^k, \boldsymbol{x}_{k+1})$.

uniform-attention Transformer to emulate it. Specifically, averaging simplex-valued features produces a summary statistic that is permutation-invariant and has a fixed total mass of 1 irrespective of $k$. See Appendix C for more details.

When prompts exhibit cross-example dependencies, the Bayes predictor can be order-dependent, and capturing it may require non-uniform attention. Still, the exchangeable prompt settings cover a standard regime where demonstrations are randomly sampled and their order is not semantically important (Brown et al., 2020; Mukherjee et al., 2021).

## 3. Risk Analysis of In-Context Learning

In this section, we first present a risk identity (Proposition 3.1), then control each term separately: Section 3.1 bounds the Bayes Gap (pretraining approximation and generalization), while Section 3.2 analyzes the Posterior Variance (inference-time uncertainty in mixtures).

The following identity decomposes the ICL risk into a model-dependent term and a model-independent term.

**Proposition 3.1** (Risk identity for in-context learning). *Consider the prompt-generating process from Definition 2.1 and assume that Assumption 2.4 holds. For a measurable, bounded map $M$, the ICL risk decomposes as*

$$R(M) = \underbrace{R_{\text{BG}}(M)}_{\text{Bayes Gap}} + \underbrace{R_{\text{PV}}}_{\text{Posterior Variance}}$$

*where:*

- ***Bayes Gap:*** $R_{\text{BG}}(M) := \frac{1}{p}\sum_{k=1}^{p} \mathbb{E}_{P^k}\left[\left(M(P^k) - M_{\text{Bayes}}(P^k)\right)^2\right]$. *This measures how closely the model $M$ approximates the optimal Bayes predictor. In other words, this is the excess risk relative to the Bayes predictor.*

- ***Posterior Variance:*** $R_{\text{PV}} := \frac{1}{p}\sum_{k=1}^{p} \mathbb{E}_{P^k}\left[\text{Var}_{f \sim \mathcal{P}(f|D^k)}(f(\boldsymbol{x}_{k+1}))\right]$, *which is*

*independent of $M$ and irreducible. This represents the behavior of the Bayes estimator given the context.*

This decomposition reveals how each term can be reduced. The Bayes Gap is the model-dependent excess risk over the Bayes predictor, controlled through architecture design and pretraining scale $(N, p)$. In contrast, the Posterior Variance stems from the inference-time uncertainty of the test task and can be reduced only by increasing the context length $k$ at inference time because $\mathbb{E}[\text{Var}_{f \sim \mathcal{P}(f|D^{k+1})}(f)] \leq \mathbb{E}[\text{Var}_{f \sim \mathcal{P}(f|D^k)}(f)]$ follows from the law of total variance. Therefore, under sufficiently large pretraining, the final error bottleneck is the latter.

Importantly, the identity itself is model-agnostic (Proposition 3.1); architectural assumptions are required only to derive an upper bound for the Bayes Gap. Also, it can be extended to a broad class of losses that admit an analogous decomposition with the Bayes estimator, such as Bregman-type losses (Adlam et al., 2022; Pfau, 2025).

### 3.1. Bayes Gap: Pretraining Generalization Error and Approximation Error

This section answers "*Can $M_\theta$ emulate the hypothetical map $P^k \mapsto \mathbb{E}_{f \sim \mathcal{P}(f|D^k)}[f(\boldsymbol{x}_{k+1})]$ ?*"

For the uniform-attention Transformers, the following theorem decomposes the Bayes Gap into an approximation term and a pretraining generalization term, and provides a non-asymptotic upper bound. This is architecture-specific and applies to the uniform-attention class in Definition 2.2. Its proof relies on the mean-pooling approximation of the Bayes predictor developed in Appendix C.

**Theorem 3.2** (Bayes Gap upper bound). *Consider the prompt-generating process defined in Definition 2.1 and the model class in Definition 2.2 under Assumptions 2.4 and 2.5. For $k = 1, \dots, p$, assume the Bayes pre-*

dictor $M_{\text{Bayes}} : (\mathbb{R}^{d_{\text{eff}}})^k \times \mathbb{R}^{d_{\text{feat}}} \to \mathbb{R}$ *satisfies the Hölder condition:* $\left| M_{\text{Bayes}}(\boldsymbol{u}_{1:k}, \boldsymbol{c}) - M_{\text{Bayes}}(\boldsymbol{u}'_{1:k}, \boldsymbol{c}') \right| \leq L \frac{1}{k} \sum_{i=1}^k \left\| (\boldsymbol{u}_i, \boldsymbol{c}) - (\boldsymbol{u}'_i, \boldsymbol{c}') \right\|_2^\alpha$ *for all bounded* $\boldsymbol{u}_i, \boldsymbol{u}'_i \in \mathcal{U}, \boldsymbol{c}, \boldsymbol{c}' \in \mathcal{C},$ *and* $\alpha \in (0, 1]$. *Then, for any* $p \geq 2$, $\hat{\theta}$ *in* (1) *satisfies*

$$\mathbb{E}[R_{\text{BG}}(M_{\hat{\theta}})] \lesssim \underbrace{m^{-\frac{2\alpha}{d_{\text{eff}}}}}_{\text{Approximation error}}$$

$$+ \underbrace{\frac{m}{pN} \text{polylog}(pN) + \frac{1}{N} \text{polylog}(pN),}_{\text{Pretraining generalization error}}$$

*where the expectation is taken over* $\{\{(P_j^k, y_{j,k+1})\}_{k=1}^p\}_{j=1}^N$ *and* $\text{polylog}(\cdot) \asymp \log^r(\cdot)$ *with some* $r \in \mathbb{N}$. *Choosing* $m^\star \asymp (pN)^{\frac{d_{\text{eff}}}{d_{\text{eff}}+2\alpha}}$ *and ignoring* $\text{polylog}(pN)$ *yields* $\mathbb{E}[R_{\text{BG}}(M_{\hat{\theta}})] \lesssim ((pN)^{-\frac{2\alpha}{d_{\text{eff}}+2\alpha}} + N^{-1})$.

*Proof Idea*: Regarding the pretraining generalization error, we handle the $N$ meta-training prompts via conventional learning theory across $j$ (van der Vaart & Wellner, 2023; Shalev-Shwartz & Ben-David, 2014), and the $p$ context examples per prompt via a sequential learning theory across $k$ (Rakhlin et al., 2015; Block et al., 2021). Concerning the approximation error, we build a mollified partition-of-unity ("soft histogram") over the example domain $\mathcal{U}$ and mean-pool it to encode prompts. Then the Bayes predictor on empirical measures is approximated by a decoder defined via a McShane extension over a discrete 1-Wasserstein metric between histograms (Peyré & Cuturi, 2019), yielding a Lipschitz, piecewise-linear target. Both encoder and decoder are then realized by moderate-size ReLU networks. As these proof ideas do not depend on a specific Bayesian formulation, the result holds under milder data assumptions compared to prior Bayesian analyses (Xie et al., 2022; Zhang et al., 2025).

Theorem 3.2 shows that the Bayes Gap decomposes into two orthogonal sources of error: (i) an approximation term $m^{-2\alpha/d_{\text{eff}}}$ and (ii) a pretraining generalization error term $\tilde{O}(m/(pN) + 1/N)$. The first term captures the expressiveness of the Transformer class: larger feature dimension $m$ yields a finer approximation of the Bayes predictor (while architectural depth and width enter only implicitly through $m$ and the Lipschitz constants of the encoder and decoder; see Lemmas G.2-G.3). The second term quantifies estimation error from finite pretraining data. Here, $p$ represents the amount of information within one task, while $N$ represents the coverage of the meta-distribution. The rate $\propto m/(pN)$ makes explicit the joint effect of $pN$, which earlier non-asymptotic theories on ICL (Kim et al., 2024a; Wu et al., 2024; Zhang et al., 2024) have not fully captured, as they typically considered the effect of $p$ and $N$ separately or focused on only one of them.

While existing results in linear settings suggest that Transformers emulate specific procedures such as ridge regression or gradient descent (e.g., Akyürek et al., 2023; Bai et al., 2023; Zhang et al., 2024), our bound implies a broader conclusion. In nonparametric, nonlinear, meta-learning regimes, pretraining can drive the model toward the Bayes-optimal in-context predictor, i.e., it selects the optimal meta-algorithm non-asymptotically.

We also highlight its ability to avoid the curse of dimensionality with respect to context length $p$. Since the Bayes predictor is unchanged no matter the order in which the context arrives, we can compress a long input sequence into a single mean vector without losing information, and the network only needs to handle that fixed-length vector of dimension $d_{\text{eff}}$ rather than $pd_{\text{eff}} + d_{\text{feat}}$.

The Hölder condition holds, intuitively, if (i) each task function is smooth (e.g., Hölder) with respect to the input, (ii) inputs and responses are effectively bounded (e.g., sub-Gaussian noise), and (iii) Bayesian updates are stable (e.g., distributions of parameters are light-tailed or log-concave), so perturbing any single context point by $O(\delta)$ changes the posterior mean by at most $O(\delta^\alpha/k)$ under the prompt metric. These conditions are typically met for mixtures of common task families (e.g., linear regression, basis-function regression, finite convex-dictionary regression). Further discussion is deferred to Appendix E. Moreover, the rate $(pN)^{-\frac{2\alpha}{d_{\text{eff}}+2\alpha}}$ matches the minimax lower bound for estimating, for example, the density of the joint distribution of $(\boldsymbol{x}_i, y_i) \in \mathcal{U}$ under the standard Hölder smoothness assumption (Tsybakov, 2009).

In practice, as the token budget used for pretraining LLMs is enormous (say, infinite), the only risk that essentially remains is the other component of the ICL risk, $R_{\text{PV}}$ analyzed in the next section.

### 3.2. Posterior Variance: Inference-time Error

We now focus on the Posterior Variance, $R_{\text{PV}}$. This term represents the irreducible error of the Bayes predictor itself. A key question is: *How does this Posterior Variance, arising from a mixture of $T$ task types, relate to the intrinsic difficulty of the true task at inference time?*

The following theorem suggests that, under some assumptions on the data (discussed later), the Bayes predictor identifies the true task type from a few-shot context.

**Theorem 3.3** (Gap between Posterior Variance and minimax risk of the true task type). *Suppose Assumption 2.4 holds. Let $i^\star$ be the true task index and recall $\alpha_{i^\star} = \Pr(I = i^\star)$ from Definition 2.1. For each wrong task $j \neq i^\star$ and each $t \geq 1$, define the predictive log-likelihood ratio increment $Z_{j,t} := \log \frac{p_j(y_t|\boldsymbol{x}_t, D^{t-1})}{p_{i^\star}(y_t|\boldsymbol{x}_t, D^{t-1})}$. Under the true task, there exist a task-type divergence $D_j > 0$*

*and constants $(\nu_j, b_j)$ such that, for all $t \geq 1$ and the filtration $\mathcal{G}_{t-1}$, $\mathbb{E}[Z_{j,t} \mid \mathcal{G}_{t-1}, I = i^\star] \leq -D_j$ and $\mathbb{E}[\exp\{\lambda(Z_{j,t} + D_j)\} \mid \mathcal{G}_{t-1}, I = i^\star] \leq \exp(\lambda^2 \nu_j^2/2)$ hold for all $|\lambda| \leq 1/b_j$. Let $D_{\min} := \min_{j \neq i^\star} D_j > 0$ and $C := \min_{j \neq i^\star} \frac{D_j^2}{8(\nu_j^2 + b_j D_j/2)} > 0$. Then, for all $k \geq 1$,*

$$
\mathbb{E}_{D^k, \boldsymbol{x}|I=i^\star} \Big[ \underbrace{\mathrm{Var}_{f|D^k}\{f(\boldsymbol{x})\}}_{\text{mixture Posterior Variance}} \Big]
$$

$$
\leq \underbrace{\inf_M \sup_{f \in F_{i^\star}} \mathbb{E}_{P^k} \Big[ \big(f(\boldsymbol{x}_{k+1}) - M(P^k)\big)^2 \Big| f \Big]}_{\text{the true task type's minimax risk}}
$$

$$
+ \underbrace{5B_f^2 \left( \frac{1 - \alpha_{i^\star}}{\alpha_{i^\star}} e^{-D_{\min}k/2} + (T-1)e^{-Ck} \right)}_{\text{task-type identification error}}.
$$

This theorem quantitatively justifies the empirical observation that ICL can quickly adapt[2] to the specific task, even when pretrained on a diverse mixture. Concretely, the posterior distribution over the task index, $\mathcal{P}_{I|D^k}$, concentrates exponentially fast on the true index $i^\star$ as $k$ grows. This is consistent with empirical demonstrations. For instance, Panwar et al. (2024) show that in hierarchical mixtures, Transformers mimic the Bayes predictor based on the true task distribution. Also, the above theorem explains the "Bayesian scaling laws" of Arora et al. (2025), which model ICL's error curves as repeated Bayesian updates, and under an ideal Bayesian learner, show the task posterior converges to the true task as context grows through experiments.

Compared to prior ICL theories, Theorem 3.3 can be seen as the general form of the result in Kim et al. (2024a), which showed that even when the function class used at pretraining is wider than the one at inference, the inference error depends only on the hardness of the latter class. Although Jeon et al. (2024) also mention an irreducible error of ICL, our addition is to show that it manifests as Posterior Variance and that it approaches the minimax risk for the "true family" up to a small gap. Moreover, this phenomenon of ICL "selecting algorithms on the fly" is consistent with the theoretical results on in-context algorithm selection in generalized linear models and the Lasso (Akyürek et al., 2023; Bai et al., 2023; Zhang et al., 2024). Our result proves that even without assuming a specific algorithmic form, behavior close to optimal algorithm selection emerges through posterior concentration in mixture settings.

Also, the result implies that ICL performance transitions from being dominated by task-type identification (small $k$) to within-family generalization (large $k$), and hence, few-

---

[2]Here, adaptation does not refer to test-time parameter updates; the model parameters remain fixed after pretraining. Rather, the posterior over the latent task type concentrates on the true family as the context grows.

shot benchmarks mainly measure task discrimination, while longer ones assess learning within the identified family.

The assumptions are fairly standard in the theory of sequential data and ensure that the in-context examples provide sufficient signal to rapidly rule out incorrect task types: (i) the supermartingale condition $\mathbb{E}[Z_{j,t} \mid \mathcal{G}_{t-1}, I = i^\star] \leq -D_j < 0$ (Williams, 1991) means each new observation, on average, decreases the predictive log-likelihood ratio of any wrong type $j$ against the true type; (ii) the Bernstein-type condition $\mathbb{E}[\exp\{\lambda(Z_{j,t} + D_j)\} \mid \mathcal{G}_{t-1}, I = i^\star] \leq \exp(\lambda^2 \nu_j^2/2)$ (Bercu et al., 2015) yields the concentration of the cumulative log-likelihood ratio, so occasional misleading samples cannot outweigh the overall trend. Note that $D_j$ is the per-step information gap that favors the true task over the wrong type $j$, $\nu_j$ is the sub-exponential scale of the log-likelihood ratio, $b_j$ bounds the tail via the moment-generating-function (smaller means heavier tails), and $\min_{j \neq i^\star} \frac{D_j^2}{8(\nu_j^2 + b_j D_j/2)}$ sets the uniform exponential rate at which posterior mass on wrong types decays with more context.

In Appendix F, we consider a concrete regression problem (linear vs. series regression) and specify $\nu_j, b_j, D_j$ and $C$ that appear in Theorem 3.3.

Remark that if the likelihood does not have a density function (with respect to Lebesgue measure), assume that all predictive distributions $P_i(y_t \mid \boldsymbol{x}_t, D^{t-1})$ are dominated by a common reference measure so that the Radon-Nikodym derivative exists. Then $Z_{j,t}$ can be rigorously defined as a log-likelihood ratio.

### 3.3. OOD Stability of the ICL Risk

This section investigates how the ICL risk changes under a distributional shift in the input between pretraining data and inference-time prompt. Note that the task distribution and the noise distribution are unchanged. Since $R_{\mathrm{PV}}$ represents the uncertainty of the task at inference time, it depends only on the prompt distribution at inference time. In contrast, the Bayes Gap $R_{\mathrm{BG}}(M_{\hat{\theta}})$, which measures the performance of the pretrained model $M_{\hat{\theta}}$, is directly affected by the discrepancy between the pretraining (source domain) and inference-time (target domain) distributions.

To formalize the problem, let $\mathsf{P}$ and $\mathsf{Q}$ denote the probability measures governing the prompt-generating process with the source input distribution $\mathcal{P}_X$ (pretraining) and the target input distribution $\mathcal{Q}_X$ (inference), respectively. Denote the Bayes Gap evaluated under $\mathsf{R} \in \{\mathsf{P}, \mathsf{Q}\}$ by $R_{\mathrm{BG}}^{(\mathsf{R})}(M_\theta) := \frac{1}{p} \sum_{k=1}^{p} \mathbb{E}_{P^k \sim \mathcal{L}_{\mathsf{R}}(P^k)}[\{M_\theta(P^k) - M_{\mathrm{Bayes}}(P^k)\}^2]$, where $\mathcal{L}_{\mathsf{R}}(Z) := \mathsf{R} \circ Z^{-1}$ is the induced distribution of a random variable $Z$.

We measure the shift at the prompt level. For $0 <$

$\alpha \leq 1$ and $k \in \{1, \ldots, p\}$, define the ground metric $\overline{d}_{k,\alpha}\big((\boldsymbol{u}_{1:k}, \boldsymbol{c}), (\boldsymbol{u}'_{1:k}, \boldsymbol{c}')\big) := \frac{1}{k}\sum_{i=1}^{k}\|\boldsymbol{u}_i - \boldsymbol{u}'_i\|_2^\alpha + \|\boldsymbol{c} - \boldsymbol{c}'\|_2^\alpha$, and the associated 1-Wasserstein distance $\mathsf{W}_\alpha^{(k)}\big(\mathcal{L}_\mathsf{P}(P^k), \mathcal{L}_\mathsf{Q}(P^k)\big) := W_1\big(\mathcal{L}_\mathsf{P}(P^k), \mathcal{L}_\mathsf{Q}(P^k); \overline{d}_{k,\alpha}\big)$. Assume $\mathcal{U}$ and $\mathcal{C}$ have finite diameters (e.g., by truncating on a high-probability event under the sub-Gaussian noise model) for brevity. Note that the decoder is uniformly Lipschitz in its two arguments with constants $(L_s, L_c)$, while $\mathrm{Lip}(\phi_\theta)$ denotes the encoder's Lipschitz constant.

**Theorem 3.4** (Wasserstein stability of the Bayes Gap). *Consider the prompt-generating process defined in Definition 2.1 under Assumptions 2.4 and 2.5 for $\mathcal{P}_X$ and $\mathcal{Q}_X$. Suppose that the Bayes predictor satisfies the same $\alpha$-Hölder condition as in Theorem 3.2 with exponent $\alpha \in (0, 1]$ and constant $L$. Then, for every parameter $\theta$,*

$$\big|R_{\mathrm{BG}}^{(\mathsf{Q})}(M_\theta) - R_{\mathrm{BG}}^{(\mathsf{P})}(M_\theta)\big|$$
$$\leq \frac{2(B_M + B_f)}{p}\sum_{k=1}^{p}\big(L + \Lambda_\alpha\big)\mathsf{W}_\alpha^{(k)}\big(\mathcal{L}_\mathsf{P}(P^k), \mathcal{L}_\mathsf{Q}(P^k)\big),$$

*where $\Lambda_\alpha := \big(L_s\mathrm{Lip}(\phi_\theta) + L_c\big)\big(\mathrm{diam}(\mathcal{U}) + \mathrm{diam}(\mathcal{C})\big)^{1-\alpha}$.*

This result implies that the Bayes Gap is distributionally Lipschitz. Its change across domains is controlled by (i) the smoothness $L$ of the Bayes predictor and (ii) the architectural regularity of $M_\theta$ through $L_s$, $L_c$, and $\mathrm{Lip}(\phi_\theta)$. In line with the findings of Zhang et al. (2024) that ICL is susceptible to input-distribution shifts, we find that only the Bayes Gap is affected by such shifts and quantify the magnitude of this effect. When moderate domain shift is unavoidable, limiting sensitivity (e.g., via spectral normalization and output clipping) may help mitigate performance degradation.

For additional theories and discussions, see Appendix D.

**Scope of the results.** We end this section by clarifying the scope of our theorems. Proposition 3.1 is model-agnostic; it is just a Bayes-risk identity under squared loss and can be extended to other losses including Bregman-type losses. Theorem 3.3 and the permutation-invariance results in Appendix C are also independent of the architecture, and use only the exchangeable prompt model. By contrast, Theorem 3.2 is specific to the uniform-attention class in Definition 2.2, because its proof uses mean pooling. Theorem 3.4 is intermediate: it only needs regularity quantities such as Lipschitz constants, so the same argument can apply to another architecture once similar bounds are proved.

# 4. Experiments

To validate our theories, we conduct pretraining and ICL experiments using a GPT-2 architecture (Radford et al., 2019), as in Oko et al. (2024) and Garg et al. (2022). Full experi-

*Table 1.* Power-law fits to the pooled Bayes-gap values from the $N$- and $p$-sweeps.

| Model | Surrogate | $R^2$ |
|---|---|---|
| joint | $a + b(pN)^{-\beta} + c/N$ | 0.912 |
| $N$-only | $a + bN^{-\beta}$ | 0.030 |
| $p$-only | $a + cp^{-\gamma}$ | 0.016 |

mental details (data generation, architecture, and optimization) are deferred to Appendix B.

**Brief setup.** We use a two-family synthetic regression mixture (linear vs. non-linear; conjugate Bayes in closed form), enabling exact Bayes (mixture/oracle) baselines and direct Bayes-gap measurement. The Transformer is pretrained on $N$ sets of length-$p$ prompts sampled from the mixture (Appendix B).

**Bayes baselines.** We report two Bayes reference curves. *Bayes (mixture)* is the Bayes predictor under the mixture prior over task families: $M_{\mathrm{Bayes}}(P^k) := \mathbb{E}\big[f(\boldsymbol{x}_{k+1})\big|D^k\big] = \sum_{i=1}^{T}\pi_i(D^k)\mathbb{E}\big[f(\boldsymbol{x}_{k+1})\,\big|\,D^k, I = i\big]$. *Bayes (oracle)* assumes the true task family is given at test time and uses the within-family posterior mean $M_{\mathrm{Bayes}}^{\mathrm{oracle}}(P^k) := \mathbb{E}\big[f(\boldsymbol{x}_{k+1})\big|D^k, I = i^\star\big]$. Thus, the gap between $M_{\mathrm{Bayes}}$ and $M_{\mathrm{Bayes}}^{\mathrm{oracle}}$ represents the task-type identification uncertainty, which our theory predicts to vanish fast in $k$ (Theorem 3.3).

**Findings.** *(1) Bayes Gap shrinks with both $N$ and $p$.* The left and middle panels in Figure 2 report the Bayes Gap, estimated on 1000 independent test prompts sampled from the same mixture. Across $N$-sweeps (left) and $p$-sweeps (middle), the Bayes Gap $R_{\mathrm{BG}}(M_{\hat{\theta}})$ decreases as either the number of pretraining prompts $N$ or the prompt length $p$ increases, supporting the coupled dependence on $pN$ in Theorem 3.2.

As an additional quantitative check, we pooled the Bayes-gap values from the $N$- and $p$-sweeps in the left and middle panels of Figure 2. We then fit the coupled model suggested by Theorem 3.2,

$$\mathrm{BG}(p, N) \approx a + b(pN)^{-\beta} + \frac{c}{N},$$

and compared it with single-variable surrogates. Table 1 shows that the joint model explains the pooled Bayes-gap values much better than the $N$-only or $p$-only alternatives.

The result suggests that the coupled $p$–$N$ power-law dependence is visible at the level of empirical trends. We interpret this as order-level support for Theorem 3.2, rather than as evidence that the constants or exponents in the upper bound are tight.

*(2) Rapid approach to the oracle Bayes curve at inference*

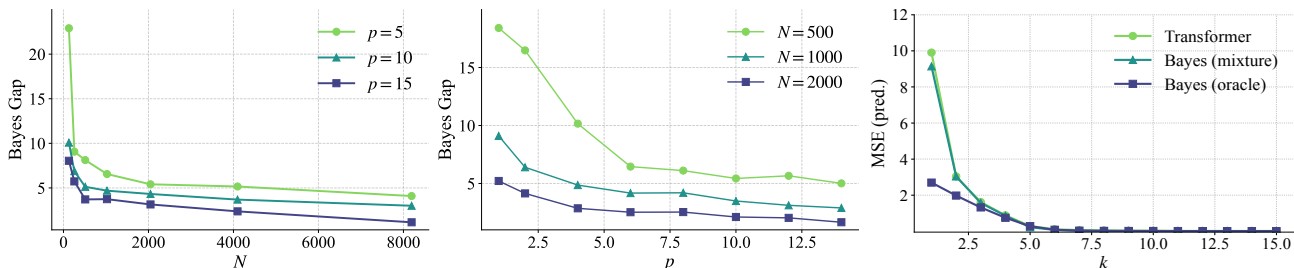

*Figure 2.* **Behavior of the Bayes Gap (left: $N$-sweep, middle: $p$-sweep).** The left panel fixes $p \in \{5, 10, 15\}$ and varies the number of pretraining prompts $N$; the middle panel fixes $N \in \{500, 1000, 2000\}$ and varies the context length $p$ in pretraining. In both cases, the Bayes Gap decreases generally as $N$ or $p$ increases. **Inference-time error under task mixtures (right).** At inference time, the sufficiently pretrained Transformer closely tracks *Bayes (mixture)* $M_{\text{Bayes}}$ and, with a few in-context examples, it approaches *Bayes (oracle)* $M_{\text{Bayes}}^{\text{oracle}}$ that knows the task family, demonstrating the rapid task-type identification.

*time.* The right panel shows the MSE averaged over 1000 independent test prompts for each $k$. With sufficiently large pretraining (so that the Bayes Gap is negligible), the Transformer's test-time MSE closely tracks *Bayes (mixture)*. As the number of examples $k$ grows, the Transformer's curve quickly approaches *Bayes (oracle)* after only a few examples. This demonstrates that the task-type identification error vanishes quickly, so the remaining error is the intrinsic error of the true task family, which aligns with Theorem 3.3.

As a further test beyond the two-family setting, Appendix B reports a harder three-family experiment with a noise-level sweep. The same qualitative pattern persists: higher noise slows task-family identification, while the Transformer continues to track the Bayes-mixture predictor and move toward the oracle curve after a few in-context examples.

Overall, our experiments support the main qualitative predictions of the theory. While our theory focuses on a uniform-attention architecture, we empirically observe similar qualitative trends in a more practical Transformer, suggesting that these phenomena are not specific to the simplified model.

## 5. Conclusion

We develop a finite-sample theory of ICL under meta-learning with task mixtures. A risk identity splits ICL error into a model-dependent Bayes Gap and a model-independent Posterior Variance, isolating what pretraining can improve versus what only additional context can reduce. For simplified Transformers, we give non-asymptotic Bayes-Gap bounds that couple prompt length $p$ and the number of pretraining prompts $N$. Furthermore, we demonstrate that ICL rapidly identifies the true task within heterogeneous mixtures, effectively reducing inference error to the true task's intrinsic uncertainty. We also quantify OOD sensitivity: only the Bayes Gap grows with Wasserstein shift. These findings provide a theoretical framework that validates ICL's capability to efficiently adapt to diverse tasks. The experiments with a GPT-2 architecture provide qualita-

tive evidence that the phenomena described by our theory also appear in a standard Transformer.

**Practical implications.** Although our analysis is stylized, it suggests several practical guidelines for ICL. First, at inference time, very long prompts are not always necessary: in our experiments, about three examples were often enough for the model to identify the task (right panel of Figure 2). For harder problems, however, additional context can still help by reducing the true-family minimax risk, as described in Theorem 3.3. Second, under distribution shift, it is preferable to match the test input distribution to the pretraining distribution whenever possible. Theorem 3.4 suggests that such shifts mainly degrade the Bayes gap; under moderate unavoidable shift, controlling Transformer Lipschitz constants, for example by spectral normalization, and then increasing $p$ and $N$ is a more principled strategy. Third, for benchmark design, small-$k$ evaluations mainly test task-family identification, while larger-$k$ evaluations also probe learning within the identified family.

**Limitations and Future Work.** Our finite-sample pretraining bound (Theorem 3.2) is proved only for the uniform-attention class. The main obstacle to extending this result to standard softmax attention is that the attention weights are prompt-dependent and couple all context tokens through logits and normalization, so the effective hypothesis class becomes data-dependent. A plausible route is to treat softmax attention as a data-dependent weighted empirical summary and combine Lipschitz control of the attention map with sequential covering / sequential Rademacher bounds for this compositional class. Another direction is to move beyond a finite and fixed number of task families. In Theorem 3.3, the task-identification term contains a factor of order $T$; under fixed separation constants, keeping this term small requires a context length that grows only logarithmically with $T$. In more realistic settings with many or infinitely many task types, this finite factor should be replaced by a complexity measure of the task space, such as a covering number, metric entropy, or a prior-mass term.

## Acknowledgments

We thank the anonymous reviewers for their constructive comments. TW was partially supported by JSPS KAKENHI (26K21188), JST ACT-X (JPMJAX23CS) and RIKEN Incentive Research Project. TS was partially supported by JSPS KAKENHI (24K02905) and JST CREST (PMJCR2015). This research is supported by the National Research Foundation, Singapore and the Ministry of Digital Development and Information under the AI Visiting Professorship Programme (award number AIVP-2024-004). Any opinions, findings and conclusions or recommendations expressed in this material are those of the author(s) and do not reflect the views of National Research Foundation, Singapore and the Ministry of Digital Development and Information.

## Impact Statement

This paper presents work whose goal is to advance the field of Machine Learning. There are many potential societal consequences of our work, none of which we feel must be specifically highlighted here.

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

# Appendix

## A. Notation and Definitions

This section provides a comprehensive list of notations and definitions used throughout the paper for ease of reference.

### General Mathematical Notation

- $\mathbb{R}^d$: The $d$-dimensional Euclidean space.

- $\|\cdot\|_2$: The Euclidean ($\ell_2$) norm for vectors and the spectral (operator) norm for matrices.

- $\|\cdot\|_1$: The $\ell_1$ norm of a vector.

- $\|\cdot\|_0$: The $\ell_0$ pseudo-norm of a vector, counting the number of non-zero elements.

- $\mathbf{1}$: A vector of all ones, with its dimension inferred from the context.

- $\Delta^{m-1}$: The standard probability simplex in $\mathbb{R}^m$, defined as $\Delta^{m-1} = \{s \in [0,1]^m : \sum_{j=1}^m s_j = 1\}$.

- $\mathcal{U}, \mathcal{C}$: The example domain (the space of $(x_i, y_i)$) and the query domain (the space of $x_{k+1}$), respectively.

- $F_i$: The function space for tasks of type $i$.

- $\Theta$: The parameter space for the neural network model $M_\theta$.

- $\mathrm{diam}(A) := \sup_{x,y \in A} \|x - y\|_2$: The diameter of a set $A$.

- $B(a, R)$: The closed Euclidean ball of radius $R \geq 0$ centered at $a$, $B(a, R) := \{x \in \mathbb{R}^d : \|x - a\|_2 \leq R\}$, where the ambient dimension $d$ is understood from context.

- $\mathrm{Lip}(f)$: The Lipschitz constant of a function $f$.

- $f \asymp g$: Indicates that $f$ and $g$ are of the same order, i.e., there exist constants $c_1, c_2 > 0$ such that $c_1 g \leq f \leq c_2 g$.

- $f \lesssim g$: Indicates that $f$ is less than or equal to $g$ up to a constant factor, i.e., $f \leq Cg$ for some universal constant $C > 0$.

- $\tilde{O}(\cdot)$: Asymptotic notation that hides polylogarithmic factors.

- $\mathrm{polylog}(\cdot) := (\log(\cdot))^{O(1)}$, i.e., $\log^c(\cdot)$ for some constant $c > 0$.

- $\sigma(\cdot)$: The Rectified Linear Unit (ReLU) activation function, $\sigma(u) = \max\{u, 0\}$, applied element-wise.

- $\mathrm{clip}_{[a,b]}(x) := \max(a, \min(b, x))$: The clipping function.

- $\circ$: The composition operator for maps. For maps $f, g$ with compatible domains or codomains, $(f \circ g)(x) = f(g(x))$. For a probability measure $\mu$ and a measurable map $T$,

$\mu \circ T^{-1}$ denotes the pushforward measure (law) of $T$ under $\mu$, i.e., $(\mu \circ T^{-1})(A) = \mu(T^{-1}(A))$ for measurable sets $A$.

### Probability and Statistics

- $\mathcal{P}_X, \mathcal{P}_\varepsilon, \ldots$: Probability distributions of random variables $X, \varepsilon, \ldots$.

- $\mathbb{E}_{X \sim \mathcal{P}_X}[\cdot]$ or simply $\mathbb{E}[\cdot]$: The expectation with respect to the distribution of the random variable(s) specified in the subscript. If no subscript is present, the expectation is taken over all relevant random variables.

- $\mathrm{Var}(\cdot)$: The variance of a random variable.

- $\mathrm{Emp}_k(u_{1:k}) := \frac{1}{k} \sum_{t=1}^k \delta_{u_t}$: The empirical measure of the context.

- $\Sigma_X := \mathbb{E}[(x - \mathbb{E}x)(x - \mathbb{E}x)^\top]$: The covariance matrix of $x$.

- $\mathrm{Pr}(\cdot)$: The probability of an event.

- $X \sim \mathcal{P}_X$: The random variable $X$ is drawn from the distribution $\mathcal{P}_X$.

- $\overset{\text{i.i.d.}}{\sim}$: A symbol for "is independently and identically distributed as".

- $X \perp Y$: The random variables $X$ and $Y$ are statistically independent.

- $\mathcal{P}_{X,Y|f}$: The joint distribution of $(X, Y)$ conditional on a function $f$.

- $\mathcal{P}^{\otimes k}$: The $k$-fold product measure, corresponding to $k$ i.i.d. draws from the distribution $\mathcal{P}$.

- $I \sim \mathrm{Categorical}(\alpha)$: $I$ is a discrete random variable on $\{1, \ldots, T\}$ with $\mathrm{Pr}(I = i) = \alpha_i$ and $\sum_{i=1}^T \alpha_i = 1$, and $I \perp \{x_k\}_{k \geq 1}, \{\varepsilon_k\}_{k \geq 1}$.

- $\mathcal{P}(f \mid D^k)$: The marginal posterior distribution of the task function $f$ given the context data $D^k$.

- $\pi_i(D^k) := \mathrm{Pr}(I = i \mid D^k)$: The marginal posterior probability of task type (task family) $i$ given the context $D^k$.

- $\mathcal{G}_k$: The $\sigma$-algebra generated by the random variables $D^k$, representing the information available at step $k$.

- $\mathcal{G}_k'$: The $\sigma$-algebra generated by the random variables $(D^k, x_{k+1})$.

- Sub-Gaussian: A centered random variable $X$ is sub-Gaussian with proxy variance $\sigma^2$ if $\mathbb{E}e^{\lambda X} \leq \exp(\sigma^2 \lambda^2 / 2)$ for all $\lambda \in \mathbb{R}$.

- Sub-exponential: A centered random variable $X$ is $(\nu, b)$-sub-exponential if $\mathbb{E}e^{\lambda X} \leq \exp(\nu^2 \lambda^2 / 2)$ for all $|\lambda| \leq 1/b$.

- KL$(P\|Q)$: Kullback–Leibler divergence between distributions $P$ and $Q$, used to quantify separation between task types.

- $m_i(D^k) := \int \prod_{t=1}^{k} p(y_t \mid \boldsymbol{x}_t, f) \, \mathcal{P}_{F_i}(\mathrm{d}f)$: The marginal likelihood of the context $D^k$ under task type $i$.

- $\mu_{i,t}(\boldsymbol{x})$, $s_{i,t}^2(\boldsymbol{x})$: Predictive mean and variance for task type $i$ after observing $t-1$ examples.

- $\mathcal{N}(\mu, \sigma^2)$: Gaussian (normal) distribution with mean $\mu$ and variance $\sigma^2$.

- Truncated Gaussian: A Gaussian distribution restricted to a bounded support set and renormalized to integrate to 1.

- $\frac{\mathrm{d}P}{\mathrm{d}Q}$: Radon–Nikodym derivative of $P$ with respect to $Q$ (when $P$ is absolutely continuous with respect to $Q$).

**Meta-learning Setup**

- $T$: The total number of distinct task types (task families).

- $p$: The maximum number of in-context examples (i.e., the context length).

- $i^\star$: The index of the true task type at inference time.

- $N$: The number of prompts in the pretraining dataset.

- $d_{\text{feat}}$: The dimensionality of the input features $\boldsymbol{x}$.

- $d_{\text{eff}} := d_{\text{feat}} + 1$: The effective dimensionality of an example pair $(\boldsymbol{x}_i, y_i)$.

- $m$: The dimensionality of the feature vector produced by the encoder, $\phi_\theta(\boldsymbol{x}_i, y_i)$.

- $P = (\boldsymbol{x}_1, y_1, \ldots, \boldsymbol{x}_p, y_p, \boldsymbol{x}_{p+1})$: A full prompt of length $p$.

- $P^k = (\boldsymbol{x}_1, y_1, \ldots, \boldsymbol{x}_k, y_k, \boldsymbol{x}_{k+1})$: A partial prompt of length $k$.

- $D^k = \{(\boldsymbol{x}_j, y_j)\}_{j=1}^{k}$: The context data, consisting of $k$ example pairs. Under our prompt-generating process, conditional on $(I, f)$, the pairs $\{(\boldsymbol{x}_j, y_j)\}_{j=1}^{k}$ are i.i.d. and hence exchangeable. Accordingly, we freely identify $D^k$ with the corresponding multiset (equivalently, its empirical measure) whenever order is irrelevant.

- $M, M_\theta, M_{\hat{\theta}}$: A generic predictor, the uniform-attention Transformer parameterized by $\theta$, and the uniform-attention Transformer obtained by empirical risk minimization (ERM), respectively.

- $\phi_\theta$: The feature encoder network that maps an example $(\boldsymbol{x}, y)$ to a feature vector in $\Delta^{m-1}$.

- $\rho_\theta$: The decoder network that predicts the output from the aggregated features and the query input.

- $\ell(u, v) = (u - v)^2$: The squared error loss function used throughout the paper.

- $S(\mathcal{T}_\theta) = \prod_{\ell=1}^{L} \|W^{(\ell)}\|_2$: The spectral product of the weight matrices of a neural network $\mathcal{T}_\theta$.

- $\text{Renorm}_\tau(\boldsymbol{s})$: A specific renormalization layer that maps a vector $\boldsymbol{s} \in \mathbb{R}^m$ to the probability simplex $\Delta^{m-1}$.

- $B_f, B_X, B_M$: Uniform bounds on $|f(\boldsymbol{x})|$, $\|\boldsymbol{x}\|_2$, and $|M(P^k)|$, respectively (as assumed).

- $S(\phi_\theta), S(\rho_\theta)$: Layerwise spectral-product bounds (or induced Lipschitz budgets) for the feature network $\phi_\theta$ and decoder $\rho_\theta$ used in generalization and stability analyses.

**Theoretical Quantities**

- $R(M)$: The in-context learning (ICL) risk of a predictor $M$. $R(M) := \frac{1}{p} \sum_{k=1}^{p} \mathbb{E}[(M(P^k) - f(\boldsymbol{x}_{k+1}))^2]$.

- $M_{\text{Bayes}}(P^k) := \mathbb{E}[f(\boldsymbol{x}_{k+1}) \mid D^k, \boldsymbol{x}_{k+1}]$: The Bayes predictor, which corresponds to the posterior mean of the query output given the context and is the optimal predictor for the squared error loss.

- $R_{\text{BG}}(M)$: The Bayes Gap, measuring the squared difference between the predictor $M$ and the Bayes predictor, averaged over prompts. This term is reducible by training the model.

- $R_{\text{BG},k}(M) := \mathbb{E}[\{M(P^k) - M_{\text{Bayes}}(P^k)\}^2]$, $R_{\text{PV},k} := \mathbb{E}[\text{Var}(f(\boldsymbol{x}_{k+1}) \mid D^k)]$: Per-$k$ versions used in the risk decomposition.

- $R_{\text{BG}}^{(\text{P})}(M) := \frac{1}{p} \sum_{k=1}^{p} \mathbb{E}_{P^k \sim \mathcal{L}_{\text{P}}(P^k)}[\{M(P^k) - M_{\text{Bayes}}(P^k)\}^2]$: Bayes Gap evaluated under a prompt distribution P (used in OOD analysis).

- $R_{\text{PV}}$: The Posterior Variance, which is the irreducible error corresponding to the variance of the posterior predictive distribution. This term is independent of the model $M$.

- $R_k^\star(F_{i^\star})$: The minimax risk for predicting a function from the true task class $F_{i^\star}$ given $k$ examples.

- $R_k^\star(F_{i^\star}; \text{R})$: The minimax risk for predicting a function from the true task class $F_{i^\star}$ under prompt distribution R (default R is the pretraining domain).

- $L, \alpha$: Constants that define the Hölder condition on the Bayes predictor (see Lemma G.3 and Theorem 3.2).

- $p_i(y_t \mid \boldsymbol{x}_t, D^{t-1})$: Posterior predictive density under task family $i$.

- $Z_{j,t} := \log \frac{p_j(y_t \mid \boldsymbol{x}_t, D^{t-1})}{p_{i^\star}(y_t \mid \boldsymbol{x}_t, D^{t-1})}$: The predictive log-likelihood ratio increment.

- $D_j$, $\nu_j$, $b_j$, $C$: Identification-rate constants for the wrong task $j \neq i^\star$; $D_j$ is the negative drift, $(\nu_j, b_j)$ are sub-exponential parameters, and $C$ controls exponential concentration of the posterior mass on incorrect types.

- $S_k$: The symmetric group on $\{1, \ldots, k\}$; $\mathcal{S}[M]$ denotes the symmetrized predictor obtained by averaging $M$ over all permutations in $S_k$.

- Predictable tree: A depth-$p$ tree $z = \{z_t(\xi_{1:t-1})\}_{t \leq p}$ whose node $z_t$ depends only on past signs $\xi_{1:t-1} \in \{\pm 1\}^{t-1}$.

- $\ell_2$ sequential metric: For a depth-$p$ tree $z$ and predictable sequences $v, v'$, and for a path $\xi \in \{\pm 1\}^p$, define $d_{2,\xi}(v, v'; z) := \left[ \frac{1}{p} \sum_{t=1}^{p} \left\{ v_t(z_t(\xi_{1:t-1})) - v'_t(z_t(\xi_{1:t-1})) \right\}^2 \right]^{1/2}$.

- $N_2^{\text{seq}}(\alpha, \mathcal{F}; z)$: The sequential covering number (Rakhlin et al., 2010; 2015) is the minimal size of a predictable $\alpha$-cover on a predictable tree $z$ with respect to $d_{2,\xi}(\cdot, \cdot; z)$ such that, for all $\xi \in \{\pm 1\}^p$ and all $f \in \mathcal{F}$, there exists $v$ in the cover with $d_{2,\xi}(f \circ z, v; z) \leq \alpha$.

- $N_2^{\text{seq}}(\alpha, \mathcal{F}, p)$: The depth-$p$ $\ell_2$ sequential covering number is the worst–tree version $N_2^{\text{seq}}(\alpha, \mathcal{F}, p) := \sup_z N_2^{\text{seq}}(\alpha, \mathcal{F}; z)$, where the supremum ranges over all predictable trees $z$ of depth $p$.

- Sequential Rademacher complexity: $\mathfrak{R}_p^{\text{seq}}(\mathcal{F}) := \sup_z \mathbb{E}_\xi \left[ \sup_{f \in \mathcal{F}} \frac{1}{p} \sum_{t=1}^{p} \xi_t f\big(z_t(\xi_{1:t-1})\big) \right]$, where $\xi_t \overset{\text{i.i.d.}}{\sim} \text{Unif}\{\pm 1\}$.

- $W_1(\mu, \nu; d)$: The 1-Wasserstein distance between probability measures $\mu, \nu$ with ground metric $d$. The specialized distances $W_\alpha^{(u)}$ and $\mathsf{W}_\alpha^{(k)}$ below are instances of $W_1(\cdot, \cdot; \cdot)$ with particular choices of $d$.

- $W_\alpha^{(u)}(\boldsymbol{s}, \boldsymbol{t})$: Discrete 1-Wasserstein on $\Delta^{m-1}$ with grid $\{\boldsymbol{r}_j\} \subset \mathcal{U}$ and cost $c^{(u)}(j, \ell) = \|\boldsymbol{r}_j - \boldsymbol{r}_\ell\|_2^\alpha$ $(0 < \alpha \leq 1)$; $W_\alpha^{(u)}(\boldsymbol{s}, \boldsymbol{t}) = \min_{\pi \geq 0} \sum_{j,\ell} c^{(u)}(j, \ell) \pi_{j\ell}$ subject to the usual marginal constraints. On the simplex, $W_\alpha^{(u)}(\boldsymbol{s}, \boldsymbol{t}) \leq \frac{\text{diam}(\mathcal{U})^\alpha}{2} \|\boldsymbol{s} - \boldsymbol{t}\|_1$.

- $\mathcal{P}_X$, $\mathcal{Q}_X$: Source (pretraining) and target (test) input distributions used in OOD analysis.

- $\mathcal{L}_\mathsf{P}(P^k)$, $\mathcal{L}_\mathsf{Q}(P^k)$: Distributions of length-$k$ prompts under the source and target domains, respectively.

- $\bar{d}_{k,\alpha}\big((\boldsymbol{u}_{1:k}, \boldsymbol{c}), (\boldsymbol{u}'_{1:k}, \boldsymbol{c}')\big) := \frac{1}{k} \sum_{i=1}^{k} \|\boldsymbol{u}_i - \boldsymbol{u}'_i\|_2^\alpha + \|\boldsymbol{c} - \boldsymbol{c}'\|_2^\alpha$: Prompt-level ground metric $(0 < \alpha \leq 1)$.

- $\mathsf{W}_\alpha^{(k)}\big(\mathcal{L}_\mathsf{P}(P^k), \mathcal{L}_\mathsf{Q}(P^k)\big) := W_1\big(\mathcal{L}_\mathsf{P}(P^k), \mathcal{L}_\mathsf{Q}(P^k); \bar{d}_{k,\alpha}\big)$: Prompt-level Wasserstein distance used in OOD bounds.

- $\mathsf{P}$, $\mathsf{Q}$: Generic prompt distributions used when evaluating risks.

## B. Details of the Experiments

In this section, we describe the experimental details and present additional plots and discussion.

**Prompt Generation** We set the feature dimension $d_{\text{feat}} = 5$. Tasks are generated from a mixture of two regression families: Family 1 is linear regression $y = \boldsymbol{w}^\top \boldsymbol{x} + \varepsilon$, and Family 2 is regression via an element-wise nonlinear map $\varphi(\boldsymbol{x}) = \tanh(\boldsymbol{x})$, i.e., $y = \boldsymbol{w}^\top \varphi(\boldsymbol{x}) + \varepsilon$. The mixture weights are $(\alpha_1, \alpha_2) = (0.5, 0.5)$. Inputs are sampled i.i.d. as $\boldsymbol{x} \sim \mathcal{N}(0, I_{d_{\text{feat}}})$. The weight vector $\boldsymbol{w}$ for each family is drawn from $\mathcal{N}(\boldsymbol{\mu}_i, \tau^2 I)$ with $\tau = 1$, $\boldsymbol{\mu}_1 = (3,3,3,3,3)$ and $\boldsymbol{\mu}_2 = (-3,-3,-3,-3,-3)$. The observation noise $\varepsilon$ is zero-mean Gaussian with $\sigma_\varepsilon = 0.1$. This process generates prompts $P^k = (\boldsymbol{x}_1, y_1, \ldots, \boldsymbol{x}_k, y_k, \boldsymbol{x}_{k+1}) \in (\mathbb{R}^{d_{\text{feat}}} \times \mathbb{R})^k \times \mathbb{R}^{d_{\text{feat}}}$ and the target $y_{k+1} \in \mathbb{R}$.

**Architecture** As in Oko et al. (2024) and Garg et al. (2022), we adopt a GPT-2 model (Radford et al., 2019; Wolf et al., 2020), with 12 layers, 8 heads, a hidden size of 256, and positional embeddings removed. The input is a token sequence of length $2k + 1$: $P^k = (\boldsymbol{x}_1, y_1, \ldots, \boldsymbol{x}_k, y_k, \boldsymbol{x}_{k+1})$.

**Evaluation of Bayes Gap** For any prompt $(D^k, \boldsymbol{x}_{k+1})$, since each family admits a conjugate Bayesian update, we can compute the mixture posterior $\pi_i(D^k)$ and the Bayes predictor $M_{\text{Bayes}}$ in closed form (Gelman et al., 2013). At inference time, using token sequences of length $2p + 1$, we compute the squared difference between the model's prediction and the Bayes prediction $M_{\text{Bayes}}(D^k, \boldsymbol{x}_{k+1})$. We average this over 1000 trials and report the mean as the Bayes Gap.

- $N$-**Sweep (fixed $p$):** For $p \in \{5, 10, 15\}$, we train the model on data with $N \in \{128, 256, \ldots, 8192\}$ and report the Bayes Gap.

- $p$-**Sweep (fixed $N$):** For $N \in \{500, 1000, 2000\}$, we vary $p \in \{1, 2, 4, 6, 8, 10, 12, 14\}$ and report the Bayes Gap.

During pretraining, we perform the ERM as in (1) using Adam (Kingma & Ba, 2015; Paszke et al., 2019) with a learning rate of 0.00005, a batch size of 64, and 10000 steps.

Figure 3 illustrates how the Bayes Gap depends jointly on the number of pretraining prompts $N$ and the context length $p$. In the $N$-sweep (left panel), larger $p$ values start with smaller gaps and maintain this advantage as $N$ grows. In the $p$-sweep (right panel), increasing $p$ steadily reduces the gap for any fixed $N$, with higher $N$ curves lying lower overall.

**Evaluation of In-Context Error** We pretrain the model on a dataset with $N = 12800000$ and $p = 15$ using Adam,

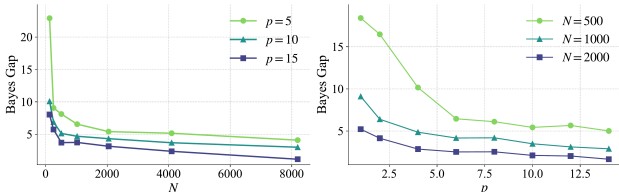

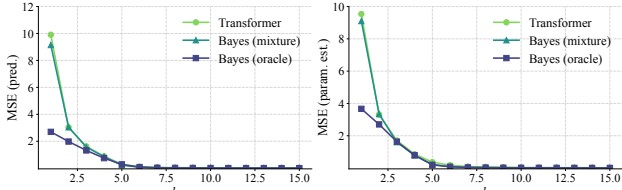

*Figure 3.* **Behavior of the Bayes Gap (left:** $N$**-sweep, right:** $p$**-sweep).** The left panel fixes $p \in \{5, 10, 15\}$ and varies the number of pretraining prompts $N$; the right panel fixes $N \in \{500, 1000, 2000\}$ and varies the context length $p$ in pretraining. In both cases, the Bayes Gap decreases generally as $N$ or $p$ increases, demonstrating that longer contexts and more pretraining improve approximation to the Bayes predictor.

*Figure 4.* **In-context error under task mixtures (left: predictive MSE; right: parameter-estimation MSE).** As the context length $k$ increases, both predictive error for the next label $y_{k+1}$ from $P^k$ (left) and parameter-estimation error (right) decrease monotonically. The Transformer closely tracks the mixture Bayes predictor and, with sufficient context, approaches the oracle Bayes curve that knows the true task family. This demonstrates the rapid concentration of the task-index posterior under growing context and the corresponding shrinkage of the irreducible term.

which involves 200000 steps of online learning, sampling 64 data points per step, similar to (Garg et al., 2022). On 1000 test prompts, the Bayes Gap is below $10^{-2}$, so the pretrained Transformer reasonably approximates $M_{\text{Bayes}}$ (cf. Theorem 3.2). We then investigate: (i) the prediction accuracy of the Transformer via the mean squared error (MSE) with respect to $y$, and (ii) Bayes-like behavior of the Transformer via linear probing: generating inputs, recovering the regression coefficients implied by the Transformer's outputs (inverting the induced linear map), and comparing against the optimal Bayesian coefficients. The probed coefficients are averaged over 2000 independent input–output sets.

**Bayes baselines.** We use two Bayes reference predictors computed in closed form (conjugate Gaussian updates). For each family $i \in \{1, 2\}$, let $m_i(D^k, \boldsymbol{x}_{k+1}) := \mathbb{E}\big[f(\boldsymbol{x}_{k+1}) \mid D^k, I = i\big]$. Let the family posterior be $\pi_i(D^k) := \Pr(I = i \mid D^k) \propto \alpha_i\, p(D^k \mid I = i)$. We report two Bayes predictors:

- **Bayes (mixture):** $M_{\text{Bayes}}(P^k) = \sum_{i=1}^{2} \pi_i(D^k)\, m_i(D^k, \boldsymbol{x}_{k+1})$.
- **Bayes (oracle):** $M_{\text{Bayes}}^{\text{oracle}}(P^k) = m_{i^\star}(D^k, \boldsymbol{x}_{k+1})$ (true family $i^\star$ is known).

The difference between these curves reflects task-type identification uncertainty.

Figure 4 illustrates the inference-time behavior of the Transformer. First, with sufficiently large pretraining, the Transformer tracks the Bayes (mixture) curve almost exactly and its error decays to nearly zero, consistent with our theory (Theorem 3.2). In terms of parameter estimation, the Transformer likewise reproduces the Bayes-mixture behavior, supporting the view that it performs Bayesian inference in context. Moreover, the gap between the Transformer and Bayes (oracle) diminishes rapidly as the number of in-context examples $k$ increases and essentially vanishes around $k \approx 3\text{–}4$. This indicates that the task type identification error quickly diminishes and the remaining bottleneck is the intrinsic (optimal) error of the true task family (Theo-

rem 3.3).

**Harder three-family experiment.** To test whether the same mechanism persists beyond the two-family mixture, we also performed a more challenging synthetic experiment. We replaced the original two-family mixture by a three-family mixture consisting of (i) linear regression, (ii) tanh-feature regression, and (iii) odd-quadratic-feature regression with the element-wise feature map $\varphi_{\text{odd}}(\boldsymbol{x}) = \boldsymbol{x} \odot |\boldsymbol{x}|$. We further introduced a difficulty sweep by increasing the observation noise over $\sigma \in \{0.1, 0.3, 0.5\}$. This keeps exact Bayes mixture and oracle baselines available while making both task-family identification and within-family prediction more challenging.

Table 2 shows two clear patterns. First, as the noise level increases, the posterior mass outside the true family decays more slowly at the same context length. Thus, harder settings require more in-context examples for task-family identification. Second, the Transformer exhibits the same qualitative behavior as the Bayes-mixture predictor: the prediction error drops rapidly in the first few in-context examples, and the Transformer quickly approaches the Bayes oracle. This indicates that the Transformer identifies the true task family after only a few examples; after that point, the remaining error is mainly the task-specific intrinsic error. Overall, the qualitative mechanism predicted by Theorem 3.3 persists in the harder three-family setting and under the noise-difficulty sweep.

## C. Permutation Invariance and Justification for Uniform-Attention Transformers

This section formalizes the permutation invariance of the Bayes predictor under the prompt-generating process (Definition 2.1). In summary, by Proposition C.6, the Bayes predictor depends on $D^k$ only via its empirical measure. Hence, any architecture that can approximate functionals of

*Table 2.* Harder three-family mixture experiment under an observation-noise sweep. Wrong-family mass is $\sum_{i \neq i^\star} \pi_i(D^k)$. Each MSE entry is reported as Transformer / Bayes (mixture) / Bayes (oracle).

| Noise $\sigma$ | Wrong-family mass @ $k = 3$ | Wrong-family mass @ $k = 5$ | MSE @ $k = 1$ Trans. / Mix / Oracle | MSE @ $k = 3$ Trans. / Mix / Oracle | MSE @ $k = 5$ Trans. / Mix / Oracle |
|---|---|---|---|---|---|
| 0.1 | 0.067 | 0.0010 | 19.15/18.027/4.063 | 1.814/1.738/1.136 | 0.156/0.109/0.013 |
| 0.3 | 0.080 | 0.0025 | 23.27/22.737/4.773 | 3.135/2.879/2.110 | 0.282/0.224/0.152 |
| 0.5 | 0.093 | 0.0054 | 23.94/22.426/6.124 | 3.291/2.908/2.364 | 0.338/0.317/0.314 |

empirical distributions, e.g., uniform-attention Transformers, matches the symmetry of the optimal predictor. Moreover, in view of Theorem C.2, replacing any non-invariant model by its permutation average never increases risk.

Recall that the loss is the squared error, and all random objects live on standard Borel spaces, so regular conditional distributions exist.

Under Definition 2.1, once the task $(I, f)$ is fixed, the context pairs $(\boldsymbol{x}_t, y_t)$ are i.i.d. draws. The following lemma says that the context can be treated as a multiset rather than an ordered list.

**Lemma C.1** (Conditional exchangeability). *Fix $(I, f)$ with $I \in \{1, \ldots, T\}$ and $f \in F_I$. Under Definition 2.1, the context pairs $(\boldsymbol{x}_t, y_t)_{t=1}^k$ are i.i.d. from $\mathcal{P}_{X,Y|f}$; hence for any permutation $\pi$ of $\{1, \ldots, k\}$,*

$$\mathcal{L}((\boldsymbol{x}_1, y_1, \ldots, \boldsymbol{x}_k, y_k) \mid I, f)$$
$$= \mathcal{L}((\boldsymbol{x}_{\pi(1)}, y_{\pi(1)}, \ldots, \boldsymbol{x}_{\pi(k)}, y_{\pi(k)}) \mid I, f),$$

*where $\mathcal{L}(Z \mid W)$ denotes the conditional law (distribution) of $Z$ given $W$.*

If the order of the context is uninformative, averaging any predictor over all permutations should not increase risk. This symmetrization principle justifies restricting attention to permutation-invariant models.

**Theorem C.2** (Risk–reducing symmetrization). *For any measurable predictor $M$ and any $k \in \{1, \ldots, p\}$, define the permutation–averaged predictor $\mathcal{S}[M](P^k) := \mathbb{E}_\Pi[M((\boldsymbol{x}_{\Pi(1)}, y_{\Pi(1)}, \ldots, \boldsymbol{x}_{\Pi(k)}, y_{\Pi(k)}), \boldsymbol{x}_{k+1}) \mid P^k]$, where $\Pi$ is uniform on the symmetric group $S_k$ and independent of everything else. Then the ICL risk satisfies*

$$R(\mathcal{S}[M]) \leq R(M).$$

Hence, by convexity of the squared loss and conditional exchangeability (Lemma C.1), permutation-averaging is a risk-reducing ensembling step. This is an instance of Rao–Blackwellization by group averaging under convex loss (Lehmann & Casella, 1998). Therefore, uniform-attention (mean-pooling) architectures are not only natural but also without loss of optimality in this setting.

*Proof of Theorem C.2.* Write $R(M) = \frac{1}{p} \sum_{k=1}^p R_k(M)$

with $R_k(M) := \mathbb{E}[(f(\boldsymbol{x}_{k+1}) - M(P^k))^2]$. Fix $k$ and condition on $(D^k, \boldsymbol{x}_{k+1}, I, f)$. By Jensen's inequality applied to the convex map $v \mapsto (f(\boldsymbol{x}_{k+1}) - v)^2$,

$$\mathbb{E}_\Pi[(f(\boldsymbol{x}_{k+1}) - M_\Pi)^2 \mid D^k, \boldsymbol{x}_{k+1}, I, f]$$
$$\geq (f(\boldsymbol{x}_{k+1}) - \mathcal{S}[M])^2,$$

where $M_\Pi := M((\boldsymbol{x}_{\Pi(1)}, y_{\Pi(1)}), \ldots, (\boldsymbol{x}_{\Pi(k)}, y_{\Pi(k)}), \boldsymbol{x}_{k+1})$. By Lemma C.1, $\mathbb{E}_\Pi[(f(\boldsymbol{x}_{k+1}) - M_\Pi)^2 \mid I, f] = \mathbb{E}[(f(\boldsymbol{x}_{k+1}) - M(P^k))^2 \mid I, f]$. Taking expectations proves $R_k(\mathcal{S}[M]) \leq R_k(M)$ and summing over $k$ yields the claim. $\square$

The previous theorem immediately yields that an optimal predictor can be chosen permutation-invariant:

**Corollary C.3** (Existence of permutation-invariant minimizers). *There exists a risk minimizer that is permutation-invariant in the $k$ context items. In particular, when analyzing architectures it is without loss of generality to restrict to permutation-invariant models, e.g., uniform-attention (mean-pooling) Transformers.*

Analytically, we may restrict our hypothesis class to set-function architectures (e.g., uniform-attention Transformers) without sacrificing optimality.

With a mixture over task families, the optimal predictor must both identify the task type and perform within-family inference. Bayes' rule exposes this computational structure explicitly.

**Theorem C.4** (Hierarchical posterior factorization). *Assume there exists a $\sigma$-finite reference measure $\mu$ such that, for all $f \in \cup_{i=1}^T F_i$ and all $x$, the conditional law $P(\cdot \mid x, f)$ is dominated by $\mu$ and admits the Radon–Nikodym derivative $p(y \mid x, f) := \frac{dP(\cdot|x,f)}{d\mu}(y)$. Then for any context $D^k$,*

$$\mathcal{P}_{F_i}(df \mid D^k, I = i) \propto \left\{ \prod_{t=1}^k p(y_t \mid \boldsymbol{x}_t, f) \right\} \mathcal{P}_{F_i}(df),$$

$$\pi_i(D^k) = \frac{\alpha_i \, m_i(D^k)}{\sum_{j=1}^T \alpha_j \, m_j(D^k)},$$

*where $m_i(D^k) := \int \prod_{t=1}^k p(y_t \mid \boldsymbol{x}_t, f) \mathcal{P}_{F_i}(df)$ and $\pi_i(D^k) := \Pr(I = i \mid D^k)$. Consequently, the Bayes*

*predictor decomposes as*

$$M_{\text{Bayes}}(P^k) = \sum_{i=1}^{T} \pi_i(D^k) \, \mathbb{E}_{f \sim \mathcal{P}_{F_i}(\cdot \mid D^k, I=i)}[f(\boldsymbol{x}_{k+1})].$$

The Bayes predictor is a mixture of within-family posterior means with weights $\pi_i(D^k)$ determined by marginal likelihoods $m_i(D^k)$. Because these weights depend on the product of likelihood factors, they are invariant to permutations of the context, foreshadowing the permutation invariance results below and validating architectures that first summarize the context before decoding.

*Proof of Theorem C.4.* Bayes' rule and conditional i.i.d. of $(\boldsymbol{x}_t, y_t)$ given $(I, f)$ yield the displayed formulas. The final expression follows from the tower property applied to $\mathbb{E}[f(\boldsymbol{x}_{k+1}) \mid D^k, \boldsymbol{x}_{k+1}]$. $\qquad\square$

**Corollary C.5** (Permutation invariance of the Bayes predictor). *For any permutation $\pi$ of $\{1, \ldots, k\}$,*

$$M_{\text{Bayes}}((\boldsymbol{x}_1, y_1), \ldots, (\boldsymbol{x}_k, y_k), \boldsymbol{x}_{k+1})$$
$$= M_{\text{Bayes}}((\boldsymbol{x}_{\pi(1)}, y_{\pi(1)}), \ldots, (\boldsymbol{x}_{\pi(k)}, y_{\pi(k)}), \boldsymbol{x}_{k+1}).$$

*Proof of Corollary C.5.* In Theorem C.4, both the within-family posterior $\mathcal{P}_{F_i}(\cdot \mid D^k, I=i)$ and the weight $\pi_i(D^k)$ depend on $D^k$ only through the product $\prod_{t=1}^{k} p(y_t \mid \boldsymbol{x}_t, f)$, which is invariant under reindexing $t \mapsto \pi(t)$. Substituting this into the mixture formula yields the claim. $\qquad\square$

**Proposition C.6** (Empirical–measure representation). *Let* $\text{Emp}_k : \mathcal{U}^k \to \mathcal{P}(\mathcal{U})$ *be the empirical measure map* $\text{Emp}_k(\boldsymbol{u}_{1:k}) = \frac{1}{k} \sum_{t=1}^{k} \delta_{\boldsymbol{u}_t}$, *where $\mathcal{P}(\mathcal{U})$ is endowed with the Borel $\sigma$-algebra of the weak topology. Then there exists a measurable map $\Psi : \mathcal{P}(\mathcal{U}) \times \mathcal{C} \to \mathbb{R}$ such that, for any* $\boldsymbol{u}_{1:k} \in \mathcal{U}^k$ *and* $\boldsymbol{c} \in \mathcal{C}$,

$$M_{\text{Bayes}}(\boldsymbol{u}_{1:k}, \boldsymbol{c}) = \Psi(\text{Emp}_k(\boldsymbol{u}_{1:k}), \boldsymbol{c}).$$

Once the context is summarized as an empirical distribution, a mean-pooled soft histogram is a faithful finite-dimensional proxy. This is precisely the representation approximated by our feature map $\phi_\theta$ and decoder $\rho_\theta$ (cf. Lemma G.3, Theorem 3.2). It also explains why the analysis avoids a dependence on the sequence length $p$ beyond averaging.

*Proof of Proposition C.6.* By Corollary C.5, $M_{\text{Bayes}}(\cdot, \boldsymbol{c})$ is invariant under permutations on $\mathcal{U}^k$. The quotient of a standard Borel space by a finite group action is standard Borel; thus any measurable, permutation-invariant map factors measurably through the canonical invariant $\text{Emp}_k$. Define $\Psi(\mu, \boldsymbol{c})$ to be the common value of $M_{\text{Bayes}}(\boldsymbol{u}_{1:k}, \boldsymbol{c})$ on the fiber $\{\boldsymbol{u}_{1:k} : \text{Emp}_k(\boldsymbol{u}_{1:k}) = \mu\}$; well-definedness follows from invariance, measurability from the quotient factorization. $\qquad\square$

*Remark C.7* (When permutation invariance may fail). The invariance arguments rely on the conditional i.i.d. structure of Definition 2.1. If inputs are chosen adaptively (active learning, bandit-style data acquisition), or if the within-prompt distribution drifts over time, $(\boldsymbol{x}_t, y_t)$ are not conditionally i.i.d. given $(I, f)$ and order may carry information. In such cases, non-uniform attention or explicitly sequential models can be beneficial, and our uniform-attention analysis should be viewed as the principled baseline for the i.i.d. prompt regime.

For further investigation of exchangeability of the Bayes predictor in the standard Bayesian statistics context, refer to (Bernardo & Smith, 1994; Gelman et al., 2013; Ghosal & van der Vaart, 2017).

# D. Further Details on Out-of-Distribution Generalization

Recall that $\mathcal{P}_X$ denotes the source input distribution used during pretraining and $\mathcal{Q}_X$ denotes an input distribution at inference time. The task distribution and the noise distribution are unchanged. We work with ground metrics and assume compact supports (finite diameters) so that constants remain finite; in particular, $\text{diam}(\mathcal{U}) < \infty$ and $\text{diam}(\mathcal{C}) < \infty$. (Unit-diameter rescaling, e.g., $\text{diam}(\mathcal{U}) \leq 1$ and $\text{diam}(\mathcal{C}) \leq 1$, is a convenient normalization and only rescales constants.)

*Remark D.1* (High-probability boundedness). As in Step 0 of Theorem 3.2, define $\mathcal{E}_\delta := \{\max_{t \leq p+1} |\varepsilon_t| \leq t_\delta\}$ with $t_\delta = \sigma_\varepsilon \sqrt{2 \log(4p/\delta)}$. On $\mathcal{E}_\delta$, the domains $\mathcal{U}, \mathcal{C}$ have finite diameters since $|y| \leq B_f + t_\delta$. Theorems in this section can thus be established on $\mathcal{E}_\delta$, while the contribution of $\mathcal{E}_\delta^\complement$ is controlled by sub-Gaussian tails, yielding an additional $O(\delta \log(1/\delta))$ term.

For a metric space $(\mathcal{Z}, d)$, we write $W_1(\mu, \nu; d) := \inf_{\pi \in \Pi(\mu, \nu)} \int d(z, z') \pi(\mathrm{d}z, \mathrm{d}z')$ for the 1-Wasserstein distance with ground metric $d$. For $0 < \alpha \leq 1$ and $k \in \mathbb{N}_+$, define the prompt-level ground metric

$$\bar{d}_{k,\alpha}\big((\boldsymbol{u}_{1:k}, \boldsymbol{c}), (\boldsymbol{u}'_{1:k}, \boldsymbol{c}')\big) := \frac{1}{k} \sum_{i=1}^{k} \|\boldsymbol{u}_i - \boldsymbol{u}'_i\|_2^\alpha + \|\boldsymbol{c} - \boldsymbol{c}'\|_2^\alpha,$$

and abbreviate $\mathsf{W}_\alpha^{(k)}(\cdot, \cdot) := W_1(\cdot, \cdot; \bar{d}_{k,\alpha})$. Likewise, for a random pair $\boldsymbol{U} = (\boldsymbol{X}, Y)$, we write $\mathsf{W}_\alpha(\cdot, \cdot) := W_1(\cdot, \cdot; \|\cdot\|_2^\alpha)$. Note that for any metric $d$ and $0 < \alpha \leq 1$, $d^\alpha$ is a metric by concavity of $t \mapsto t^\alpha$. Since $|f| \leq B_f$, we have $|M_{\text{Bayes}}| \leq B_f$ and hence $|M_\theta - M_{\text{Bayes}}| \leq B_M + B_f$.

We restate Theorem 3.4 here for a self-contained proof.

**Theorem D.2** (Wasserstein stability: OOD upper bound for the Bayes Gap). *Under Definition 2.1, Definition 2.2 and Assumptions 2.4–2.5 for $\mathcal{P}_X$ and $\mathcal{Q}_X$, assume the Bayes*

*predictor* $M_{\text{Bayes}}$ *satisfies the same* $\alpha$-*Hölder condition as in Theorem 3.2 with exponent* $\alpha \in (0,1]$ *and constant* $L$. *Then, for any* $\theta$,

$$
\left| R_{\text{BG}}^{(\mathsf{Q})}(M_\theta) - R_{\text{BG}}^{(\mathsf{P})}(M_\theta) \right|
$$
$$
\leq \frac{2(B_M + B_f)}{p} \sum_{k=1}^{p} \left(L + \Lambda_\alpha\right) \mathsf{W}_\alpha^{(k)}\left(\mathcal{L}_{\mathsf{P}}(P^k), \mathcal{L}_{\mathsf{Q}}(P^k)\right),
$$

*where* $\Lambda_\alpha := \left(L_s \text{Lip}(\phi_\theta) + L_c\right)\left(\text{diam}(\mathcal{U}) + \text{diam}(\mathcal{C})\right)^{1-\alpha}$. *In particular, when* $\alpha = 1$, $\Lambda_1 = L_s \text{Lip}(\phi_\theta) + L_c$.

*Proof of Theorem D.2.* Fix $k \in \{1, \ldots, p\}$ and abbreviate $\boldsymbol{z} = (\boldsymbol{u}_{1:k}, \boldsymbol{c}) \in \mathcal{U}^k \times \mathcal{C}$, $s(\boldsymbol{z}) := \frac{1}{k} \sum_{i=1}^{k} \phi_\theta(\boldsymbol{u}_i) \in \Delta^{m-1}$, and $M_\theta(\boldsymbol{z}) := \text{clip}_{[-B_M, B_M]}\left(\rho_\theta(s(\boldsymbol{z}), \boldsymbol{c})\right)$. Write the Bayes predictor as $M_{\text{Bayes}}(\boldsymbol{z}) := M_{\text{Bayes}}(\boldsymbol{u}_{1:k}, \boldsymbol{c})$ and introduce $g_k(\boldsymbol{z}) := \left(M_\theta(\boldsymbol{z}) - M_{\text{Bayes}}(\boldsymbol{z})\right)^2$ and $h_k(\boldsymbol{z}) := M_\theta(\boldsymbol{z}) - M_{\text{Bayes}}(\boldsymbol{z})$.

**Step 1 (Lipschitz modulus of $M_\theta$ under $\overline{d}_{k,\alpha}$).** By the network size assumption and the 1-Lipschitzness of clipping,

$$
\left| M_\theta(\boldsymbol{z}) - M_\theta(\boldsymbol{z}') \right| \leq L_s \left\| s(\boldsymbol{z}) - s(\boldsymbol{z}') \right\|_2 + L_c \|\boldsymbol{c} - \boldsymbol{c}'\|_2.
$$

Let $L_\phi := \text{Lip}(\phi_\theta)$ (for our encoder with $\text{Renorm}_\tau$, $L_\phi \leq \frac{2\sqrt{m}}{\tau} S(g_\theta)$). Since $\phi_\theta$ is $L_\phi$–Lipschitz,

$$
\|s(\boldsymbol{z}) - s(\boldsymbol{z}')\|_2 \leq \frac{1}{k} \sum_{i=1}^{k} \|\phi_\theta(\boldsymbol{u}_i) - \phi_\theta(\boldsymbol{u}_i')\|_2
$$
$$
\leq \frac{L_\phi}{k} \sum_{i=1}^{k} \|\boldsymbol{u}_i - \boldsymbol{u}_i'\|_2.
$$

Let $D_U := \text{diam}(\mathcal{U})$ and $D_C := \text{diam}(\mathcal{C})$, and put $D := D_U + D_C$. For $0 < \alpha \leq 1$ and any $t \in [0, D]$ one has $t \leq D^{1-\alpha} t^\alpha$; hence

$$
\frac{1}{k} \sum_{i=1}^{k} \|\boldsymbol{u}_i - \boldsymbol{u}_i'\|_2 \leq D_U^{1-\alpha} \frac{1}{k} \sum_{i=1}^{k} \|\boldsymbol{u}_i - \boldsymbol{u}_i'\|_2^\alpha,
$$
$$
\|\boldsymbol{c} - \boldsymbol{c}'\|_2 \leq D_C^{1-\alpha} \|\boldsymbol{c} - \boldsymbol{c}'\|_2^\alpha.
$$

Using the prompt-level metric $\overline{d}_{k,\alpha}(\boldsymbol{z}, \boldsymbol{z}') = \frac{1}{k} \sum_{i=1}^{k} \|\boldsymbol{u}_i - \boldsymbol{u}_i'\|_2^\alpha + \|\boldsymbol{c} - \boldsymbol{c}'\|_2^\alpha$, we obtain

$$
\left| M_\theta(\boldsymbol{z}) - M_\theta(\boldsymbol{z}') \right| \leq \left(L_s \text{Lip}(\phi_\theta) + L_c\right) D^{1-\alpha} \overline{d}_{k,\alpha}(\boldsymbol{z}, \boldsymbol{z}')
$$
$$
= \Lambda_\alpha \overline{d}_{k,\alpha}(\boldsymbol{z}, \boldsymbol{z}').
$$

**Step 2 (Lipschitz modulus of $h_k$ and $g_k$).** Note that the Hölder condition in Theorem 3.2 implies $|M_{\text{Bayes}}(\boldsymbol{z}) - M_{\text{Bayes}}(\boldsymbol{z}')| \leq L \overline{d}_{k,\alpha}(\boldsymbol{z}, \boldsymbol{z}')$, because, for $\alpha \in (0,1]$, $\|(\boldsymbol{u}, \boldsymbol{c}) - (\boldsymbol{u}', \boldsymbol{c}')\|_2^\alpha \leq \|\boldsymbol{u} - \boldsymbol{u}'\|_2^\alpha + \|\boldsymbol{c} - \boldsymbol{c}'\|_2^\alpha$. By this observation, $M_{\text{Bayes}}$ is $\overline{d}_{k,\alpha}$–Lipschitz with constant $L$, hence

$$
|h_k(\boldsymbol{z}) - h_k(\boldsymbol{z}')| \leq (\Lambda_\alpha + L) \overline{d}_{k,\alpha}(\boldsymbol{z}, \boldsymbol{z}').
$$

Because $|M_\theta| \leq B_M$ and $|M_{\text{Bayes}}| \leq B_f$, the range of $h_k$ is contained in $[-(B_M + B_f), B_M + B_f]$. Therefore, using $|a^2 - b^2| = |(a-b)(a+b)| \leq 2(B_M + B_f)|a - b|$,

$$
\left| g_k(\boldsymbol{z}) - g_k(\boldsymbol{z}') \right|
$$
$$
\leq 2(B_M + B_f) \left| h_k(\boldsymbol{z}) - h_k(\boldsymbol{z}') \right|
$$
$$
\leq 2(B_M + B_f)(L + \Lambda_\alpha) \overline{d}_{k,\alpha}(\boldsymbol{z}, \boldsymbol{z}').
$$

Thus $g_k$ is $\overline{d}_{k,\alpha}$–Lipschitz with modulus $2(B_M + B_f)(L + \Lambda_\alpha)$.

**Step 3 (Kantorovich–Rubinstein duality and averaging over $k$).** Let $\mathcal{L}_{\mathsf{P}}(P^k)$ and $\mathcal{L}_{\mathsf{Q}}(P^k)$ denote the distributions of the length-$k$ prompts under the source and target domains, respectively. Kantorovich–Rubinstein duality for $W_1(\cdot, \cdot; \overline{d}_{k,\alpha})$ implies, for any Lipschitz $g_k$,

$$
\left| \mathbb{E}_Q g_k(P^k) - \mathbb{E}_P g_k(P^k) \right|
$$
$$
\leq \text{Lip}_{\overline{d}_{k,\alpha}}(g_k) \mathsf{W}_\alpha^{(k)}\left(\mathcal{L}_{\mathsf{P}}(P^k), \mathcal{L}_{\mathsf{Q}}(P^k)\right),
$$

where $\mathsf{W}_\alpha^{(k)} := W_1(\cdot, \cdot; \overline{d}_{k,\alpha})$. Combining it with the Lipschitz bound from Step 2 yields

$$
\left| R_{\text{BG}}^{(\mathsf{Q})}(M_\theta) - R_{\text{BG}}^{(\mathsf{P})}(M_\theta) \right|
$$
$$
\leq \frac{2(B_M + B_f)}{p} \sum_{k=1}^{p} \left(L + \Lambda_\alpha\right) \mathsf{W}_\alpha^{(k)}\left(\mathcal{L}_{\mathsf{P}}(P^k), \mathcal{L}_{\mathsf{Q}}(P^k)\right),
$$

which is exactly the claimed inequality. $\square$

The prompt $P^k = (\boldsymbol{U}_1, \ldots, \boldsymbol{U}_k, \boldsymbol{C})$ contains dependent coordinates in general, because the context responses $\boldsymbol{U}_i = (\boldsymbol{X}_i, Y_i)$ share the latent task function $f$ within a prompt. Therefore, a direct product of coordinate-wise optimal couplings is not a valid coupling of the prompt distributions. The following conditional coupling fixes this.

*Remark* D.3 (Prompt-level Wasserstein via conditional coupling). Let $S$ be a latent seed that is shared across domains and determines the task index and task function. For instance, one may take $S = (I, f)$. Conditional on $S$, the prompt coordinates factorize as $\mathcal{L}_{\mathsf{P}}(P^k \mid S) = \left(\mathcal{L}_{\mathsf{P}}(U \mid S)\right)^{\otimes k} \times \mathcal{P}_X$ and $\mathcal{L}_{\mathsf{Q}}(P^k \mid S) = \left(\mathcal{L}_{\mathsf{Q}}(U \mid S)\right)^{\otimes k} \times \mathcal{Q}_X$, where $U = (\boldsymbol{X}, Y)$ and $\mathcal{P}_X, \mathcal{Q}_X$ are the (source/target) input distributions. In particular, conditional on $S$ the $k$ context pairs are i.i.d. under each domain. (If one prefers to carry a coupling of the additive noise across domains, introduce an exogenous noise seed that determines the noise distribution but not its realized sample path; this preserves conditional i.i.d.)

**Lemma D.4** (Conditional product-type upper bound for prompt-level Wasserstein). *Under the setting of Remark D.3, for every* $k \geq 1$ *and* $0 < \alpha \leq 1$,

$$
\mathsf{W}_\alpha^{(k)}\left(\mathcal{L}_{\mathsf{P}}(P^k), \mathcal{L}_{\mathsf{Q}}(P^k)\right)
$$
$$
\leq \mathbb{E}_S \left[ \mathsf{W}_\alpha\left(\mathcal{L}_{\mathsf{P}}(U \mid S), \mathcal{L}_{\mathsf{Q}}(U \mid S)\right)\right] + \mathsf{W}_\alpha(\mathcal{P}_X, \mathcal{Q}_X),
$$

*where the prompt-level ground metric is*

$$\overline{d}_{k,\alpha}\big((\boldsymbol{u}_{1:k}, \boldsymbol{c}), (\boldsymbol{u}'_{1:k}, \boldsymbol{c}')\big) := \frac{1}{k}\sum_{i=1}^{k}\|\boldsymbol{u}_i - \boldsymbol{u}'_i\|_2^\alpha + \|\boldsymbol{c} - \boldsymbol{c}'\|_2^\alpha,$$

*and for single pairs $\boldsymbol{U} = (\boldsymbol{X}, Y)$ we write $\mathsf{W}_\alpha(\cdot, \cdot) := W_1(\cdot, \cdot; \|\cdot\|_2^\alpha)$.*

*Proof of Lemma D.4.* **Step 1 (conditional product coupling).** Fix $S = s$. By Remark D.3, under each domain the $k$ context coordinates are i.i.d. with common conditional distribution $\mathcal{L}_\mathsf{P}(\boldsymbol{U} \mid s)$ (resp. $\mathcal{L}_\mathsf{Q}(\boldsymbol{U} \mid s)$), while the query coordinate has distribution $\mathcal{P}_X$ (resp. $\mathcal{Q}_X$) independent of the context. Let $\pi_U^s$ be an optimal coupling for $\mathsf{W}_\alpha\big(\mathcal{L}_\mathsf{P}(\boldsymbol{U} \mid s), \mathcal{L}_\mathsf{Q}(\boldsymbol{U} \mid s)\big)$ with ground metric $d_\alpha(\boldsymbol{u}, \boldsymbol{u}') := \|\boldsymbol{u} - \boldsymbol{u}'\|_2^\alpha$, and let $\pi_C$ be an optimal coupling for $\mathsf{W}_\alpha(\mathcal{P}_X, \mathcal{Q}_X)$ with ground metric $d_\alpha(\boldsymbol{c}, \boldsymbol{c}') := \|\boldsymbol{c} - \boldsymbol{c}'\|_2^\alpha$. Construct a coupling $\Pi_s$ of $\mathcal{L}_\mathsf{P}(P^k \mid s)$ and $\mathcal{L}_\mathsf{Q}(P^k \mid s)$ by drawing $(\boldsymbol{U}_i, \boldsymbol{U}'_i) \overset{\text{i.i.d.}}{\sim} \pi_U^s$ for $i = 1, \ldots, k$ and $(\boldsymbol{C}, \boldsymbol{C}') \sim \pi_C$, all independent across coordinates. Then, by the definition of the prompt-level ground metric,

$$\mathbb{E}_{\Pi_s}\Big[\overline{d}_{k,\alpha}\big((\boldsymbol{U}_{1:k}, \boldsymbol{C}), (\boldsymbol{U}'_{1:k}, \boldsymbol{C}')\big)\Big]$$
$$= \frac{1}{k}\sum_{i=1}^{k}\mathbb{E}_{\pi_U^s}\big[d_\alpha(\boldsymbol{U}_i, \boldsymbol{U}'_i)\big] + \mathbb{E}_{\pi_C}\big[d_\alpha(\boldsymbol{C}, \boldsymbol{C}')\big]$$
$$= \mathsf{W}_\alpha\big(\mathcal{L}_\mathsf{P}(\boldsymbol{U} \mid s), \mathcal{L}_\mathsf{Q}(\boldsymbol{U} \mid s)\big) + \mathsf{W}_\alpha(\mathcal{P}_X, \mathcal{Q}_X).$$

Therefore,

$$\mathsf{W}_\alpha^{(k)}\big(\mathcal{L}_\mathsf{P}(P^k \mid s), \mathcal{L}_\mathsf{Q}(P^k \mid s)\big)$$
$$\leq \mathsf{W}_\alpha\big(\mathcal{L}_\mathsf{P}(\boldsymbol{U} \mid s), \mathcal{L}_\mathsf{Q}(\boldsymbol{U} \mid s)\big) + \mathsf{W}_\alpha(\mathcal{P}_X, \mathcal{Q}_X).$$

**Step 2 (disintegration and convexity).** Write the unconditional prompt distributions as mixtures over $S$: $\mathcal{L}_\mathsf{P}(P^k) = \int \mathcal{L}_\mathsf{P}(P^k \mid s)\nu(\mathrm{d}s)$ and $\mathcal{L}_\mathsf{Q}(P^k) = \int \mathcal{L}_\mathsf{Q}(P^k \mid s)\nu(\mathrm{d}s)$, where $\nu$ is the (shared) distribution of $S$ under both domains (task distribution and noise distribution are kept the same across domains). By convexity of $W_1(\cdot, \cdot; \overline{d}_{k,\alpha})$ in each argument,

$$\mathsf{W}_\alpha^{(k)}\big(\mathcal{L}_\mathsf{P}(P^k), \mathcal{L}_\mathsf{Q}(P^k)\big)$$
$$\leq \int \mathsf{W}_\alpha^{(k)}\big(\mathcal{L}_\mathsf{P}(P^k \mid s), \mathcal{L}_\mathsf{Q}(P^k \mid s)\big)\nu(\mathrm{d}s)$$
$$\leq \mathbb{E}_S\big[\mathsf{W}_\alpha\big(\mathcal{L}_\mathsf{P}(\boldsymbol{U} \mid S), \mathcal{L}_\mathsf{Q}(\boldsymbol{U} \mid S)\big)\big] + \mathsf{W}_\alpha(\mathcal{P}_X, \mathcal{Q}_X),$$

which is the desired bound. □

**Corollary D.5** (Input-only reduction under Lipschitz tasks). *Assume $Y = f(X) + \varepsilon$ with a shared noise coupling across domains (possibly conditional on $S$) and a task family that is uniformly $L_f$-Lipschitz in $x$: $|f(\boldsymbol{x}) - f(\boldsymbol{x}')| \leq L_f\|\boldsymbol{x} - \boldsymbol{x}'\|_2$ for all tasks. Then for every $k \geq 1$ and $0 < \alpha \leq 1$,*

$$\mathsf{W}_\alpha^{(k)}\big(\mathcal{L}_\mathsf{P}(P^k), \mathcal{L}_\mathsf{Q}(P^k)\big) \leq (2 + L_f^\alpha)\mathsf{W}_\alpha(\mathcal{P}_X, \mathcal{Q}_X).$$

*Proof of Corollary D.5.* Under the shared-noise coupling, by subadditivity of $t \mapsto t^\alpha$ for $\alpha \in (0, 1]$, $\|\boldsymbol{U} - \boldsymbol{U}'\|_2^\alpha \leq \|\boldsymbol{X} - \boldsymbol{X}'\|_2^\alpha + |f(\boldsymbol{X}) - f(\boldsymbol{X}')|^\alpha \leq (1 + L_f^\alpha)\|\boldsymbol{X} - \boldsymbol{X}'\|_2^\alpha$. Hence $\mathsf{W}_\alpha(\mathcal{L}_\mathsf{P}(\boldsymbol{U} \mid S), \mathcal{L}_\mathsf{Q}(\boldsymbol{U} \mid S)) \leq (1 + L_f^\alpha)\mathsf{W}_\alpha(\mathcal{P}_X, \mathcal{Q}_X)$ for every $S$. Plug this into Lemma D.4 and add the $\mathsf{W}_\alpha(\mathcal{P}_X, \mathcal{Q}_X)$ term for the query coordinate $\boldsymbol{C}$. □

Define the domain-specific ICL risk

$$R^{(\mathsf{R})}(M) := \frac{1}{p}\sum_{k=1}^{p}\mathbb{E}_{P^k \sim \mathcal{L}_\mathsf{R}(P^k)}\big[(f(\boldsymbol{x}_{k+1}) - M(P^k))^2\big].$$

Combining Proposition 3.1, Theorem 3.2, Theorem D.2 with either Lemma D.4 or Corollary D.5, and absorbing polylogarithms into $\tilde{O}(\cdot)$, yields the same end-to-end OOD risk bound as in the main text, with the prompt-level Wasserstein term. The additional terms quantify distribution shift incurred during pretraining (via $\mathcal{L}_\mathsf{P}(P^k)$ vs. $\mathcal{L}_\mathsf{Q}(P^k)$); once $\theta$ is fixed, the inference-time predictor risk $R_{\mathrm{PV}}$ is evaluated under the target domain alone and does not carry extra estimation error from pretraining. Putting everything together, for the target domain $\mathcal{Q}_X$ we obtain

$$\mathbb{E}R^{(\mathsf{Q})}(M_{\hat{\theta}})$$
$$\leq \underbrace{\frac{1}{p}\sum_{k=1}^{p}R_k^\star(F_{i^\star}; \mathcal{Q}_X)}_{\text{oracle risk under the true task type in the target domain}}$$
$$+ \tilde{O}\left(m^{-\frac{2\alpha}{d_{\mathrm{eff}}}} + \frac{m}{pN} + \frac{1}{N}\right)$$
$$+ \underbrace{\frac{2(B_M + B_f)}{p}\sum_{k=1}^{p}\big(L + \Lambda_\alpha\big)\mathsf{W}_\alpha^{(k)}\big(\mathcal{L}_\mathsf{P}(P^k), \mathcal{L}_\mathsf{Q}(P^k)\big)}_{\text{OOD penalty on the Bayes Gap}}$$
$$+ \underbrace{\frac{5B_f^2}{p}\left(\frac{\frac{1-\alpha_{i^\star}}{\alpha_{i^\star}}}{e^{D_{\min}/2} - 1} + \frac{T - 1}{e^C - 1}\right)}_{\text{mixture identification remainder}},$$

where $R_k^\star(F_{i^\star}; \mathcal{Q}_X)$ denotes the minimax risk for predicting a function from the true task class $F_{i^\star}$ under prompt distribution $\mathcal{Q}_X$.

# E. On the Hölder Condition of the Bayes Predictor

Theorem 3.2 assumes a Hölder condition of the Bayes predictor on a bounded set. Since our final bound is in expectation, it is sufficient that this Hölder condition holds on a high-probability event under the prompt distribution; the contribution of the complement event can be controlled by sub-Gaussian tails (cf. Step 0 in the proof of Theorem 3.2). Therefore, the conditions below are meant as convenient

high-probability sufficient (not necessary) conditions, rather than global ones holding for every possible context realization.

In addition to Assumptions 2.4-2.5, assume the noise is Gaussian for simplicity. Under the additional regularity conditions below (holding on a high-probability event), the Bayes predictor $M_{\mathrm{Bayes}}$ is Hölder ($\alpha = 1$), with the family-specific Hölder constants listed below:

- Linear regression. $f_{(w,b)}(\boldsymbol{x}) = \boldsymbol{w}^\top \boldsymbol{x} + b$ with $\|\boldsymbol{w}\|_2 \leq B_w$, $|b| \leq B_b$ and feature map $\psi(\boldsymbol{x}) = [\boldsymbol{x}^\top, 1]^\top$. Then $\boldsymbol{x} \mapsto f_{(w,b)}(\boldsymbol{x})$ is $B_w$-Lipschitz.

- Finite-order series regression. $f_a(\boldsymbol{x}) = \sum_{j=1}^R a_j g_j(\boldsymbol{x})$ with $\|\boldsymbol{a}\|_1 \leq A$, basis functions satisfying $\|g_j\|_\infty \leq 1$ and $\|\nabla g_j(\boldsymbol{x})\|_2 \leq L_g$ uniformly; take $\psi(\boldsymbol{x}) = [g_1(\boldsymbol{x}), \ldots, g_R(\boldsymbol{x})]^\top$. Then $\boldsymbol{x} \mapsto f_a(\boldsymbol{x})$ is $AL_g$-Lipschitz.

- Finite convex dictionary. $f_a = \sum_{j=1}^J a_j f^{(j)}$ with $\boldsymbol{a} \in \Delta^{J-1}$, each atom obeying $|f^{(j)}(\boldsymbol{x})| \leq B_f$ and $\|\nabla f^{(j)}(\boldsymbol{x})\|_2 \leq L_f$ uniformly; take $\psi(\boldsymbol{x}) = [f^{(1)}(\boldsymbol{x}), \ldots, f^{(J)}(\boldsymbol{x})]^\top$. Then $\boldsymbol{x} \mapsto f_a(\boldsymbol{x})$ is $L_f$-Lipschitz. (An example of a distribution on $\Delta^{J-1}$ is the logistic-normal distribution (Aitchison & Shen, 1980).)

We consider these three regression models. For these models, we assume the following conditions hold with high probability:

- For task family $i$, there exist a dimension $d_i \in \mathbb{N}$ and a parameter space $\Theta_i \subset \mathbb{R}^{d_i}$ such that $f_{\boldsymbol{\theta}} : \mathcal{C} \to \mathbb{R}$ for every $\boldsymbol{\theta} \in \Theta_i$. Moreover, the model is uniformly bounded and Hölder in the query: $\sup_{\boldsymbol{\theta} \in \Theta_i} \sup_{\boldsymbol{x} \in \mathcal{C}} |f_{\boldsymbol{\theta}}(\boldsymbol{x})| \leq B_f$ and $\sup_{\boldsymbol{\theta} \in \Theta_i} \sup_{\boldsymbol{x} \neq \boldsymbol{x}'} \frac{|f_{\boldsymbol{\theta}}(\boldsymbol{x}) - f_{\boldsymbol{\theta}}(\boldsymbol{x}')|}{\|\boldsymbol{x} - \boldsymbol{x}'\|_2^\alpha} \leq L_{f,i}$.

- The distribution on $\boldsymbol{\theta}$ given $I = i$ has a density $\varpi_i(\boldsymbol{\theta}) \propto \exp\{-V(\boldsymbol{\theta})\}$ on $\Theta_i$, where $V$ is twice continuously differentiable and $\nabla^2 V(\boldsymbol{\theta}) \succeq \lambda_0 I_{d_i}$ for all $\boldsymbol{\theta} \in \Theta_i$, for some $\lambda_0 > 0$. In particular, a Gaussian distribution $\mathcal{N}(0, \Lambda_0^{-1})$ satisfies this with $\lambda_0 = \lambda_{\min}(\Lambda_0)$.

- Let $u = (\boldsymbol{x}, y) \in \mathcal{U}$ and define the per-sample loss $\tilde{\ell}(\boldsymbol{\theta}; u) := \frac{1}{2\sigma_\varepsilon^2}(f_{\boldsymbol{\theta}}(\boldsymbol{x}) - y)^2$. There exists a constant $L_{\boldsymbol{\theta},i} < \infty$ such that, for all $\boldsymbol{\theta} \in \Theta_i$ and all $\boldsymbol{u}, \tilde{\boldsymbol{u}} \in \mathcal{U}$, $\|\nabla_{\boldsymbol{\theta}} \tilde{\ell}(\boldsymbol{\theta}; \boldsymbol{u}) - \nabla_{\boldsymbol{\theta}} \tilde{\ell}(\boldsymbol{\theta}; \tilde{\boldsymbol{u}})\|_2 \leq L_{\boldsymbol{\theta},i} \|\boldsymbol{u} - \tilde{\boldsymbol{u}}\|_2^\alpha$.

- There exists $b$ such that $\frac{1}{k} \sum_{t=1}^k \psi(\boldsymbol{x}_t) \psi^\top(\boldsymbol{x}_t) \succeq bI$ (as in Bai et al. (2023)).

In the mixture setting, the Bayes predictor decomposes as $M_{\mathrm{Bayes}}(P^k) = \sum_{i=1}^T \pi_i(D^k) \mu_i(D^k, \boldsymbol{c})$ with $\mu_i(D^k, \boldsymbol{c}) := \mathbb{E}[f(\boldsymbol{c}) \mid D^k, I = i]$ and $\pi_i(D^k) \propto \alpha_i m_i(D^k)$, $m_i(D^k) := \int \exp\{-(2\sigma_\varepsilon^2)^{-1} \sum_{r=1}^k (f_{\boldsymbol{\theta}}(\boldsymbol{x}_r) - y_r)^2\} \, \mathrm{d}\mathcal{P}_i(\boldsymbol{\theta})$. The above single-family arguments and the assumptions imply that each $\mu_i$ is $\alpha$-Hölder with a constant $L_{\mu,i}$ independent of $k$.

However, the mixture weights $\pi_i(D^k)$ depend on the context through the marginal evidences $m_i(D^k)$. In identifiable mixtures (the regime of Theorem 3.3), the task posterior concentrates exponentially fast on the true index $i^\star$, which makes the gating increasingly stable. Concretely, under the log-likelihood-ratio conditions of Theorem 3.3, there exists an event $\mathcal{E}_k$ with $\Pr(\mathcal{E}_k) \geq 1 - (T-1)e^{-Ck}$ such that on $\mathcal{E}_k$, $1 - \pi_{i^\star}(D^k) = \sum_{i \neq i^\star} \pi_i(D^k) \leq \frac{1-\alpha_{i^\star}}{\alpha_{i^\star}} e^{-D_{\min}k/2}$. As a result, the mixture correction to the modulus of continuity of $M_{\mathrm{Bayes}}$ is exponentially suppressed, and the effective Hölder constant is independent of $k$ up to an exponentially small term. Also, a uniform margin assumption $\min_{i \neq j} \frac{1}{k} |\log m_i(D^k) - \log m_j(D^k)| \geq \gamma > 0$ implies the same conclusion with $e^{-\gamma k}$.

# F. Details of Theorem 3.3

This appendix is illustrative rather than required for Theorem 3.3: it specializes the theorem to a concrete linear-vs-series regression mixture and verifies the abstract sequential conditions under transparent sufficient conditions.

Before presenting concrete examples, we record two technical remarks for clarity and rigor.

*Remark* F.1 (Filtration for the log-likelihood ratio increments). Let $\mathcal{G}_{t-1} := \sigma(D^{t-1})$ and $\mathcal{G}'_{t-1} := \sigma(D^{t-1}, X_t)$. The increment $Z_{j,t}$ depends on the fresh covariate $X_t$. Since $X_t \overset{\text{i.i.d.}}{\sim} \mathcal{P}_X$ is independent of $D^{t-1}$ under our PGP, conditioning on $\mathcal{G}_{t-1}$ averages over $X_t \sim \mathcal{P}_X$. Moreover, for any integrable $\varphi$,

$$\mathbb{E}[\varphi(Z_{j,t}) \mid \mathcal{G}_{t-1}] = \mathbb{E}\big[\mathbb{E}[\varphi(Z_{j,t}) \mid \mathcal{G}'_{t-1}] \big| \mathcal{G}_{t-1}\big].$$

Hence, any drift or mgf bounds stated conditional on $\mathcal{G}'_{t-1}$ imply the corresponding $\mathcal{G}_{t-1}$-conditional ones by the tower property; in particular, the drift condition in Theorem 3.3 can be interpreted as an $X_t$-averaged separation.

*Remark* F.2 (Dominating measure and definition of $Z_{j,t}$). If the predictive distributions do not admit Lebesgue densities, assume they are all dominated by a common $\sigma$-finite reference measure $\mu$. Then the Radon–Nikodym derivatives $p_i(\cdot \mid x_t, D^{t-1}) = \frac{dP_i(\cdot \mid x_t, D^{t-1})}{d\mu}$ exist ($\mu$-a.e.), and the log-likelihood ratio increment $Z_{j,t} := \log \frac{p_j(Y_t \mid X_t, D^{t-1})}{p_{i^\star}(Y_t \mid X_t, D^{t-1})}$ is well-defined.

We concretely investigate Theorem 3.3 for a pair of task families: *linear regression* versus a *series (basis) regression* that excludes constant and linear terms.

**Standing assumptions.**

- Inputs are bounded and i.i.d.: $\boldsymbol{X} \sim \mathcal{P}_X$ with $\|\boldsymbol{X}\|_2 \leq B_X$ a.s. and $\mathbb{E}[\boldsymbol{X}] = 0$. Let $\Sigma_X := \mathbb{E}[\boldsymbol{X}\boldsymbol{X}^\top]$, which we assume is positive definite on $\mathbb{R}^{d_{\mathrm{feat}}}$ with $\lambda_{\min}(\Sigma_X) > 0$.

- Noise is Gaussian (a special case of sub-Gaussian): $\varepsilon \overset{\text{i.i.d.}}{\sim} \mathcal{N}(0, \sigma_\varepsilon^2)$ independent of $(f, \boldsymbol{X})$.

- Boundedness of tasks. For the linear class

$$F_{\text{lin}} = \left\{ f_{\boldsymbol{w}, b}(\boldsymbol{x}) = \boldsymbol{w}^\top \boldsymbol{x} + b : \ \|\boldsymbol{w}\|_2 \le B_w, \ |b| \le B_b \right\},$$

we have $|f_{w,b}(\boldsymbol{x})| \le B_w B_X + B_b =: B_f$ on the support of $\mathcal{P}_X$. For the series class

$$F_{\text{ser}} = \left\{ f_a(\boldsymbol{x}) = \sum_{r=r_0}^{R_{\max}} a_r g_r(\boldsymbol{x}) : \ \|\boldsymbol{a}\|_2 \le B_a \right\},$$

assume $r_0 \ge 2$ (so constant and linear terms are excluded), the basis $\{g_r\}_{r=r_0}^{R_{\max}}$ is orthonormal in $L^2(\mathcal{P}_X)$, orthogonal to linear functions, and bounded pointwise, i.e. $\sup_x |g_r(\boldsymbol{x})| \le G_{\max}$. Then $|f_a(\boldsymbol{x})| \le \|\boldsymbol{a}\|_2 \cdot \|(g_r(\boldsymbol{x}))_{r=r_0}^{R_{\max}}\|_2 \le B_a \sqrt{R_{\max} - r_0 + 1} G_{\max} =: B_f$.

- Within each family, we use a truncated Gaussian parameter distribution supported on the above bounded parameter sets (to respect $|f| \le B_f$) and otherwise conjugate: $\boldsymbol{\theta}_{\text{lin}} := (\boldsymbol{w}, b) \sim \mathcal{N}(0, \tau_{\text{lin}}^2 \mathbf{I})$ truncated to $\{\|\boldsymbol{w}\| \le B_w, \ |b| \le B_b\}$ and $\boldsymbol{\theta}_{\text{ser}} := \boldsymbol{a} \sim \mathcal{N}(0, \tau_{\text{ser}}^2 \mathbf{I})$ truncated to $\{\|\boldsymbol{a}\|_2 \le B_a\}$. The truncation preserves boundedness; the standard Gaussian formulas below give upper bounds (hence valid constants) for the truncated case because the posterior covariances are $\preceq$ their untruncated analogues on the bounded domain.

**A generic Gaussian–predictive bound for $Z_{j,t}$**  Fix a time $t$ and condition on $\mathcal{G}_{t-1}$ and $\boldsymbol{X}_t = \boldsymbol{x}$. Under any task type (task family) $i$, the (posterior) predictive distribution is Gaussian

$$p_i(y \mid \boldsymbol{x}, \mathcal{G}_{t-1}) = \mathcal{N}(\mu_{i,t}(\boldsymbol{x}), \ s_{i,t}^2(\boldsymbol{x})),$$

with mean $\mu_{i,t}(\boldsymbol{x})$ (the posterior mean of $f(\boldsymbol{x})$) and predictive variance

$$s_{i,t}^2(\boldsymbol{x}) = \sigma_\varepsilon^2 + \text{Var}(f(\boldsymbol{x}) \mid \mathcal{G}_{t-1}, I = i).$$

For our conjugate priors, $\mu_{\text{lin},t}(\boldsymbol{x}) = \phi(\boldsymbol{x})^\top \boldsymbol{m}_{t-1}$, and $s_{\text{lin},t}^2(\boldsymbol{x}) = \sigma_\varepsilon^2 + \phi(\boldsymbol{x})^\top \Sigma_{t-1} \phi(\boldsymbol{x})$, $\phi(\boldsymbol{x}) := \begin{bmatrix} \boldsymbol{x} \\ 1 \end{bmatrix}$, and similarly $\mu_{\text{ser},t}(\boldsymbol{x}) = \psi(\boldsymbol{x})^\top \tilde{\boldsymbol{m}}_{t-1}$, and $s_{\text{ser},t}^2(\boldsymbol{x}) = \sigma_\varepsilon^2 + \psi(\boldsymbol{x})^\top \tilde{\Sigma}_{t-1} \psi(\boldsymbol{x})$, $\psi(\boldsymbol{x}) := (g_r(\boldsymbol{x}))_{r=r_0}^{R_{\max}}$. Because $\|\phi(\boldsymbol{x})\|_2 \le B_\phi := \sqrt{B_X^2 + 1}$ and $\|\psi(\boldsymbol{x})\|_2 \le B_\psi := \sqrt{R_{\max} - r_0 + 1} G_{\max}$ and the posterior covariances are bounded by the prior covariances, we have a uniform variance upper bound

$$s_{i,t}^2(\boldsymbol{x}) \le \sigma_\varepsilon^2 + \bar{V}, \qquad \bar{V} := \max\{\tau_{\text{lin}}^2 B_\phi^2, \ \tau_{\text{ser}}^2 B_\psi^2\}. \quad (2)$$

For two types $i$ (true) and $j$ (wrong), define the log-predictive increment

$$Z_{j,t} := \log \frac{p_j(Y_t \mid \boldsymbol{X}_t, \mathcal{G}_{t-1})}{p_i(Y_t \mid \boldsymbol{X}_t, \mathcal{G}_{t-1})}.$$

A direct Gaussian calculation (writing $Y_t = \mu_{i,t}(\boldsymbol{X}_t) + s_{i,t}(\boldsymbol{X}_t)\varepsilon$ with $\varepsilon \sim \mathcal{N}(0,1)$) yields

$$\mathbb{E}[Z_{j,t} \mid \mathcal{G}_{t-1}, \boldsymbol{X}_t]$$
$$= -\text{KL}\Big(\mathcal{N}(\mu_{i,t}, s_{i,t}^2) \,\Big\|\, \mathcal{N}(\mu_{j,t}, s_{j,t}^2)\Big)$$
$$= -\frac{1}{2}\left\{ \log \frac{s_{j,t}^2}{s_{i,t}^2} + \frac{s_{i,t}^2}{s_{j,t}^2} - 1 + \frac{(\mu_{i,t} - \mu_{j,t})^2}{s_{j,t}^2} \right\}.$$

Consequently, for every $(t, \boldsymbol{x})$,

$$\mathbb{E}[Z_{j,t} \mid \mathcal{G}_{t-1}, \boldsymbol{X}_t = \boldsymbol{x}]$$
$$\le -\frac{(\mu_{i,t}(\boldsymbol{x}) - \mu_{j,t}(\boldsymbol{x}))^2}{2\, s_{j,t}^2(\boldsymbol{x})}$$
$$\le -\frac{(\mu_{i,t}(\boldsymbol{x}) - \mu_{j,t}(\boldsymbol{x}))^2}{2(\sigma_\varepsilon^2 + \bar{V})}. \quad (3)$$

Moreover, the centered increment $Z_{j,t} + D_{j,t}$ with $D_{j,t} := -\mathbb{E}[Z_{j,t} \mid \mathcal{G}_{t-1}, \boldsymbol{X}_t]$ is a quadratic polynomial in a standard normal variable $Z_{j,t} + D_{j,t} = a_t \varepsilon + b_t (\varepsilon^2 - 1)$ with $a_t := -\frac{(\mu_{i,t} - \mu_{j,t}) s_{i,t}}{s_{j,t}^2}$ and $b_t := -\frac{1}{2}\left(\frac{s_{i,t}^2}{s_{j,t}^2} - 1\right)$, hence sub-exponential. Calculating the mgf

$$\mathbb{E} e^{\lambda(a_t \varepsilon + b_t(\varepsilon^2 - 1))}$$
$$= e^{-\lambda b_t}(1 - 2\lambda b_t)^{-1/2} \exp\left(\frac{\lambda^2 a_t^2}{2(1 - 2\lambda b_t)}\right),$$

and the elementary bound $-\ln(1 - u) - u \le u^2$ valid for $|u| \le 1/2$ (note that $|b_t| \le \bar{V}/(2\sigma_\varepsilon^2)$, so $u = 2\lambda b_t \in [-1/2, 1/2]$ whenever $|\lambda| \le 1/b_j$), we obtain the uniform sub-exponential parameters $(\nu_j, b_j)$ in Theorem 3.3 with

$$\nu_j^2 \le \frac{8 B_f^2 (\sigma_\varepsilon^2 + \bar{V})}{\sigma_\varepsilon^4} + \frac{\bar{V}^2}{\sigma_\varepsilon^4}, \qquad b_j := \frac{2\bar{V}}{\sigma_\varepsilon^2}.$$

**Pair A: true linear vs. wrong series (degree $\ge 2$)**  Assume the data are generated by some $f^\star(\boldsymbol{x}) = \boldsymbol{w}_\star^\top \boldsymbol{x} + b_\star \in F_{\text{lin}}$ and the wrong family is $F_{\text{ser}}$ with orthonormal $\{g_r\}_{r=r_0}^{R_{\max}}, r_0 \ge 2$, orthogonal to 1 and to all linear functionals of $\boldsymbol{X}$. Let $\Pi_{\text{ser}}$ denote the $L^2(\mathcal{P}_X)$–orthogonal projection onto $\text{span}\{g_r\}$.

By orthogonality, $\Pi_{\text{ser}} f^\star \equiv 0$, hence the $L^2$–gap between the true function and the wrong family is

$$\Delta_{\text{lin} \to \text{ser}}^2 := \|f^\star - \Pi_{\text{ser}} f^\star\|_{L^2(\mathcal{P}_X)}^2$$
$$= \mathbb{E}[(\boldsymbol{w}_\star^\top \boldsymbol{X} + b_\star)^2]$$
$$= \boldsymbol{w}_\star^\top \Sigma_X \boldsymbol{w}_\star + b_\star^2.$$

For conjugate normal models with bounded regressors $\phi, \psi$ and positive definite design covariances, standard ridge-risk bounds in series regression (§3.4 in van der Vaart & Wellner, 2023) give $\|\mu_{\text{lin},t} - f^\star\|^2_{L^2(\mathcal{P}_X)} = O\big(\frac{d_{\text{feat}}+1}{t}\big)$ and $\|\mu_{\text{ser},t} - \Pi_{\text{ser}}f^\star\|^2_{L^2(\mathcal{P}_X)} = O\big(\frac{R_{\max}-r_0+1}{t}\big)$. Thus, taking $t_0 = \tilde{O}\left(\frac{d_{\text{feat}}+R_{\max}-r_0+1}{\Delta^2_{\text{lin}\to\text{ser}}}\right)$, for all $t \geq t_0$, we have

$$\mathbb{E}_X\big[(\mu_{\text{lin},t}(\boldsymbol{X}) - \mu_{\text{ser},t}(\boldsymbol{X}))^2\big] \geq \frac{1}{2}\Delta^2_{\text{lin}\to\text{ser}}.$$

Combining with (3) and $s^2_{\text{ser},t} \leq \sigma^2_\varepsilon + \bar{V}$ gives the uniform negative drift (for all $t \geq t_0$)

$$\mathbb{E}[Z_{j,t} \mid \mathcal{G}_{t-1}] \leq -D_j, \qquad D_j := \frac{\Delta^2_{\text{lin}\to\text{ser}}}{4(\sigma^2_\varepsilon + \bar{V})}.$$

From Theorem 3.3, the posterior mass on the wrong family is

$$\frac{1-\alpha_{i^\star}}{\alpha_{i^\star}}\exp\left(-\frac{D_j}{2}k\right) + \exp(-C_j k),$$

where $C_j := \frac{D_j^2}{8\big(\nu_j^2 + b_j D_j/2\big)}$. Therefore, to make the mixture identification remainder $\leq \eta$, it suffices (up to absolute constants and polylog factors) to take

$$\begin{aligned}
k = \tilde{O}\Big(&\frac{\sigma^2_\varepsilon + \bar{V}}{\Delta^2_{\text{lin}\to\text{ser}}}\log\frac{1}{\eta} \\
&\vee \Big[\frac{(\sigma^2_\varepsilon+\bar{V})^2}{\Delta^4_{\text{lin}\to\text{ser}}\sigma^4_\varepsilon}\big(B_f^2(\sigma^2_\varepsilon+\bar{V}) + \bar{V}^2\big) \\
&\quad + \frac{(\sigma^2_\varepsilon+\bar{V})\bar{V}}{\Delta^2_{\text{lin}\to\text{ser}}\sigma^2_\varepsilon}\Big]\log\frac{1}{\eta}\Big).
\end{aligned} \qquad (4)$$

The first term is the dominant, interpretable signal-to-noise scaling:

$$k \asymp \frac{\sigma^2_\varepsilon + \bar{V}}{\boldsymbol{w}_\star^\top \Sigma_X \boldsymbol{w}_\star + b_\star^2}\log\frac{1}{\eta}.$$

**Pair B: true series (degree $\geq 2$) vs. wrong linear** Now the data come from $f^\star(\boldsymbol{x}) = \sum_{r=r_0}^{R_{\max}} a_r^\star g_r(\boldsymbol{x})$ with $\|\boldsymbol{a}^\star\|_2 \leq B_a$ and the wrong family is linear. Orthogonality gives $\Pi_{\text{lin}}f^\star \equiv 0$ (since $r_0 \geq 2$ and $\mathbb{E}[\boldsymbol{X}] = 0$), hence

$$\begin{aligned}
\Delta^2_{\text{ser}\to\text{lin}} &:= \big\|f^\star - \Pi_{\text{lin}}f^\star\big\|^2_{L^2(\mathcal{P}_X)} \\
&= \|f^\star\|^2_{L^2(\mathcal{P}_X)} \\
&= \sum_{r=r_0}^{R_{\max}}(a_r^\star)^2.
\end{aligned}$$

Exactly the same argument as above yields, for $t \geq t_0 = \tilde{O}\big(\frac{d_{\text{feat}}+R_{\max}-r_0+1}{\Delta^2_{\text{ser}\to\text{lin}}}\big)$, $D_j = \frac{\Delta^2_{\text{ser}\to\text{lin}}}{4(\sigma^2_\varepsilon+\bar{V})}$, $\nu_j^2 \leq \frac{8B_f^2(\sigma^2_\varepsilon+\bar{V})+\bar{V}^2}{\sigma^4_\varepsilon}$, and $b_j = \frac{2\bar{V}}{\sigma^2_\varepsilon}$ and the same $k$–order as in (4) with $\Delta_{\text{lin}\to\text{ser}}$ replaced by $\Delta_{\text{ser}\to\text{lin}}$.

**Remarks and extensions** All bounds above use only: (i) $|f| \leq B_f$ on the support of $\mathcal{P}_X$; (ii) the uniform predictive variance upper bound (2); and (iii) $L^2(\mathcal{P}_X)$–orthogonality for the two families considered. Using truncated conjugate priors guarantees (i) and keeps (ii) finite with the explicit $\bar{V}$ given. The sub-exponential constants remain valid for truncated conjugate posteriors because truncation can only decrease posterior covariances, hence decrease $|a_t|$ and $|b_t|$.

If $\{g_r\}$ is merely linearly independent (non-orthonormal), let $\Pi_{\text{ser}}$ be the $L^2(\mathcal{P}_X)$–projection onto $\text{span}\{g_r\}$. Then the formulas hold with $\Delta^2_{\text{lin}\to\text{ser}} = \big\|f^\star - \Pi_{\text{ser}}f^\star\big\|^2_{L^2(\mathcal{P}_X)}$, $\Delta^2_{\text{ser}\to\text{lin}} = \big\|f^\star - \Pi_{\text{lin}}f^\star\big\|^2_{L^2(\mathcal{P}_X)}$, and the same $(\nu_j, b_j)$ (with the same $\bar{V}$) because the mgf bound depended only on boundedness and Eq. (2).

## G. Technical Lemmas

**Lemma G.1** (Posterior variance is bounded by the true task's minimax risk). *Suppose the prompt-generating process is as described in Definition 2.1 and that Assumptions 2.4-2.5 hold. Fix a task-type index $i^\star \in \{1, \ldots, T\}$ and recall that $F_{i^\star} = \text{supp}(\mathcal{P}_{F_{i^\star}})$ is the corresponding function class (support of the true task type prior). For any $k \geq 1$,*

$$\begin{aligned}
&\mathbb{E}_{f\sim\mathcal{P}_{F_{i^\star}}}\mathbb{E}_{D^k\sim\mathcal{P}^{\otimes k}_{X,Y|f}}\mathbb{E}_{\boldsymbol{x}\sim\mathcal{P}_X}\Big[\text{Var}_{f\sim\mathcal{P}_{F_{i^\star}|D^k}}(f(\boldsymbol{x}))\Big] \\
&\leq \inf_M \sup_{f\in F_{i^\star}}\mathbb{E}_{P^k}\Big[\big(f(\boldsymbol{x}_{k+1}) - M(P^k)\big)^2\Big|f\Big],
\end{aligned}$$

*where the left-hand side is the conditional Posterior Variance average under the true task type and $M$ belongs to the bounded and measurable function space.*

This suggests that if the true task type is given, the Posterior Variance is smaller than the minimax $L_2$ prediction error.

**Lemma G.2** (Sequential covering bound). *Fix $k \in \mathbb{N}_+$. Let $\mathcal{U} \subset \mathbb{R}^{d_{\text{eff}}}$ and $\mathcal{C} \subset \mathbb{R}^{d_{\text{feat}}}$ be bounded with $\sup_{\boldsymbol{u}\in\mathcal{U}}\|\boldsymbol{u}\|_2 \leq R_U$ and $\text{diam}(\mathcal{C}) < \infty$. For any $\theta \in \Theta$ and $P^k = (\boldsymbol{u}_1, \ldots, \boldsymbol{u}_k, \boldsymbol{c}) \in \mathcal{U}^k \times \mathcal{C}$, consider the uniform-attention architecture*

$$M_\theta(P^k) = \rho_\theta\left(\frac{1}{k}\sum_{i=1}^k\phi_\theta(\boldsymbol{u}_i), \boldsymbol{c}\right),$$

*where the query $\boldsymbol{c}$ is shared across the $k$ context items within each $P^k$ (i.e., $\boldsymbol{c}$ does not depend on $i$ inside the mean $\frac{1}{k}\sum_{i=1}^k\phi_\theta(\boldsymbol{u}_i)$). Assume:*

*(i) $\phi_\theta : \mathcal{U} \to \Delta^{m-1}$ is $L_\phi$–Lipschitz, where $L_\phi := \text{Lip}(\phi_\theta) \leq \frac{2\sqrt{m}}{\tau}S(g_\theta)$ for our encoder with $\text{Renorm}_\tau$, and the ReLU component satisfies $S(g_\theta) \leq C_\phi m^{1/d_{\text{eff}}}$. Moreover, $(\phi_\theta)_j \in [0,1]$ and $\sum_{j=1}^m(\phi_\theta)_j \equiv 1$, and $\phi_\theta$ admits a realization with $\tilde{O}(m)$-weights and $O(\log m)$-layers. Put $B_\phi := \sup_j\|(\phi_\theta)_j\|_\infty \leq 1$.*

*(ii)* $\rho_\theta : \Delta^{m-1} \times \mathcal{C} \to \mathbb{R}$ *is a ReLU network with spectral product* $S(\rho_\theta) \leq C_\rho m^{1/2}$, *is jointly Lipschitz,*

$$|\rho_\theta(\boldsymbol{s}, \boldsymbol{c}) - \rho_\theta(\boldsymbol{s}', \boldsymbol{c}')| \leq L_s \|\boldsymbol{s} - \boldsymbol{s}'\|_2 + L_c \|\boldsymbol{c} - \boldsymbol{c}'\|_2,$$

*with* $L_s, L_c \leq C_\rho m^{1/2}$, *and its (clipped) output is bounded,* $|\rho_\theta| \leq B_M$.

Let $\mathcal{H} := \{P^k \mapsto M_\theta(P^k) - M_{\mathrm{Bayes}}(P^k) : \theta \in \Theta\}$ *be the centered class for any fixed target* $M_{\mathrm{Bayes}}$. *Denote by* $N_2^{\mathrm{seq}}(\delta, \cdot; z)$ *the sequential covering number under the* $\ell_2$ *sequential metric on a depth-k predictable tree* $z$. *Then, for all* $\delta \in (0, 2B_M]$,

$$\sup_z \log N_2^{\mathrm{seq}}(\delta, \mathcal{H}; z) \lesssim m \log\left(\frac{\sqrt{m}}{\delta}\right) + k \log\left(\frac{1}{\delta}\right).$$

**Lemma G.3** (Approximation error of the Bayes predictor by a uniform-attention Transformer). *Let* $\mathcal{U} \subset \mathbb{R}^{d_{\mathrm{eff}}}$ *and* $\mathcal{C} \subset \mathbb{R}^{d_{\mathrm{feat}}}$ *be non-empty compact sets with* $\mathrm{diam}(\mathcal{U}) \leq 1$. *For every* $k \in \mathbb{N}_+$, *consider a permutation-invariant map* $M_{\mathrm{Bayes}} : \mathcal{Z}^k \to \mathbb{R}$ *on* $\mathcal{Z} := \mathcal{U} \times \mathcal{C}$ *satisfying the Hölder condition*

$$|M_{\mathrm{Bayes}}(\boldsymbol{z}_{1:k}) - M_{\mathrm{Bayes}}(\boldsymbol{z}'_{1:k})| \leq L \frac{1}{k} \sum_{i=1}^k \|\boldsymbol{z}_i - \boldsymbol{z}'_i\|_2^\alpha,$$

*where* $\alpha \in (0, 1]$, $\boldsymbol{z}_i = (\boldsymbol{u}_i, \boldsymbol{c})$, *and* $\boldsymbol{z}'_i = (\boldsymbol{u}'_i, \boldsymbol{c}')$. *Then, for any* $\eta \in (0, e^{-1})$, *there exists an integer* $m \asymp \eta^{-d_{\mathrm{eff}}/\alpha}$ *and a* $C^\infty$ *partition of unity* $\phi = (\phi_1, \ldots, \phi_m) : \mathcal{U} \to [0, 1]^m$ *with* $\sum_{j=1}^m \phi_j \equiv 1$ *such that, writing* $s(\boldsymbol{u}_{1:k}) := \frac{1}{k} \sum_{i=1}^k \phi(\boldsymbol{u}_i) \in \Delta^{m-1}$, *one can construct a (clipped) ReLU decoder* $\rho_\theta : \Delta^{m-1} \times \mathcal{C} \to \mathbb{R}$ *so that* $\sup_{c \in \mathcal{C}} \sup_{\boldsymbol{u}_{1:k} \in \mathcal{U}^k} |M_{\mathrm{Bayes}}(\boldsymbol{u}_{1:k}, \boldsymbol{c}) - \rho_\theta(s(\boldsymbol{u}_{1:k}), \boldsymbol{c})| \leq C(d_{\mathrm{eff}}) L\eta$. *Furthermore,* $\rho_\theta$ *is uniformly Lipschitz and bounded with respect to* $(\boldsymbol{s}, \boldsymbol{c})$, *and the layer-wise spectral product can be controlled as follows:*

$$\left|\rho_\theta(\boldsymbol{s}, \boldsymbol{c}) - \rho_\theta(\boldsymbol{s}', \boldsymbol{c}')\right| \leq L_s \|\boldsymbol{s} - \boldsymbol{s}'\|_2 + L_c \|\boldsymbol{c} - \boldsymbol{c}'\|_2,$$

$$L_s \leq CL\sqrt{m}, \quad L_c \leq CLm^{(1-\alpha)/d_{\mathrm{eff}}} \leq CL\sqrt{m},$$

$$|\rho_\theta| \leq B_M \qquad S(\rho_\theta) \leq CL\sqrt{m}.$$

*In addition,* $\phi$ *can be uniformly approximated by a ReLU network with* $O(\log m)$-*layers and* $O(m \log m)$-*weights, and its implementation satisfies* $\sum_j (\phi_\theta)_j \equiv 1$, $(\phi_\theta)_j \in [0, 1]$, *with the spectral product satisfying* $S(\phi_\theta) \leq C_\phi m^{1/d_{\mathrm{eff}}}$.

This lemma guarantees that the uniform-attention Transformer we are analyzing has the capacity to adequately represent smooth Bayesian predictors. This yields a fixed-length, permutation-invariant representation independent of context length $p$ with provable approximation rates that feed directly into the sequential generalization analysis.

**Lemma G.4** (Oracle inequality for $R_{\mathrm{BG}}$). *Let* $\mathcal{D}_{\mathrm{train}} = \{\{(P_j^k, y_{j,k+1})\}_{k=1}^p\}_{j=1}^N$ *be draws from the prompt-generating process in Definition 2.1. Let* $M_{\hat{\theta}}$ *be the ERM* (1) *of the Transformer (Definition 2.2). Suppose Assumptions 2.4–2.5 hold. If* $\inf_{\theta \in \Theta} R_{\mathrm{BG}}(M_\theta) = O(\frac{1}{N}(\frac{m}{p} + 1))$,

$$\mathbb{E}R_{\mathrm{BG}}(M_{\hat{\theta}}) \lesssim \inf_{\theta \in \Theta} R_{\mathrm{BG}}(M_\theta) + \frac{m}{pN} \mathrm{polylog}(pN)$$

$$+ \frac{1}{N} \mathrm{polylog}(pN),$$

*where* $\mathrm{polylog}(pN)$ *denotes a factor that is a polynomial in* $\log pN$, *the expectation is taken with respect to* $\mathcal{D}_{\mathrm{train}}$, *and* $\mathcal{M} := \{M_\theta : \theta \in \Theta\}$.

The pretraining generalization term is $\tilde{O}(\frac{m}{pN} + \frac{1}{N})$. Since we view $m$ as an architecture design parameter and choose $m = m(pN)$, the hypothesis class $\mathcal{M}$ and its approximation error are $pN$-dependent. Increasing $m(pN)$ improves the approximation ability but also increases the estimation error, and the above bound balances these two effects. Note that the auxiliary condition in the above lemma is satisfied (up to constants) for the choice $m = m^\star$ by Lemma G.3.

# H. Proofs of the Main Results

*Proof of Proposition 3.1.* Let $R_k(M) := \mathbb{E}_{I,f,D^k,\boldsymbol{x}_{k+1}}\left[(f(\boldsymbol{x}_{k+1}) - M(P^k))^2\right]$, where the expectation is taken over the joint distribution of the task $I \sim \mathcal{P}_I$, the function $f \sim \mathcal{P}_{F_I}$, the context $D^k \sim \mathcal{P}_{X,Y|f}^{\otimes k}$, and the query $\boldsymbol{x}_{k+1} \sim \mathcal{P}_X$. Then, $R(M) = \frac{1}{p}\sum_{k=1}^p R_k(M)$.

For any $k$-context $D^k$ and query $\boldsymbol{x}_{k+1}$, define $M_{\mathrm{Bayes}}(P^k) = \mathbb{E}_{f\sim\mathcal{P}(f|D^k)}[f(\boldsymbol{x}_{k+1})]$. By simple algebra:

$$R_k(M)$$
$$= \mathbb{E}_{I,f,P^k}[(f(\boldsymbol{x}_{k+1}) - M(P^k))^2]$$
$$= \mathbb{E}_{I,f,P^k}[(f(\boldsymbol{x}_{k+1}) - M_{\mathrm{Bayes}}(P^k))^2]$$
$$+ \mathbb{E}_{I,f,P^k}[(M_{\mathrm{Bayes}}(P^k) - M(P^k))^2]$$
$$+ 2\mathbb{E}_{I,f,P^k}[(f(\boldsymbol{x}_{k+1}) - M_{\mathrm{Bayes}}(P^k))$$
$$(M_{\mathrm{Bayes}}(P^k) - M(P^k))]. \tag{5}$$

Let $\mathcal{G}'_k$ be the $\sigma$-algebra generated by $(D^k, \boldsymbol{x}_{k+1})$. Since $f$ is almost surely finite and $(M_{\mathrm{Bayes}}(P^k) - M(P^k))$ is $\mathcal{G}'_k$-measurable, by the tower property of conditional expectation:

$$\mathbb{E}_{I,f,P^k}[(f(\boldsymbol{x}_{k+1}) - M_{\mathrm{Bayes}}(P^k))(M_{\mathrm{Bayes}}(P^k) - M(P^k))]$$
$$= \mathbb{E}_{I,f,P^k}\Big[(M_{\mathrm{Bayes}}(P^k) - M(P^k))$$
$$\mathbb{E}_{I,f}[f(\boldsymbol{x}_{k+1}) - M_{\mathrm{Bayes}}(P^k) \mid \mathcal{G}'_k]\Big].$$

$M_{\mathrm{Bayes}}(P^k) = \mathbb{E}_{f\sim\mathcal{P}(f|D^k)}[f(\boldsymbol{x}_{k+1})]$ implies the inner expectation equals zero.

From (5) with vanishing cross-term, $R_k(M)$ is decomposed as $R_k(M) = R_{\mathrm{PV},k} + R_{\mathrm{BG},k}(M)$, where

$$
\begin{aligned}
R_{\mathrm{PV},k} &:= \mathbb{E}_{I,f,P^k}[(f(\boldsymbol{x}_{k+1}) - M_{\mathrm{Bayes}}(P^k))^2] \\
&= \mathbb{E}_{P^k}[\mathbb{E}_{f \sim \mathcal{P}(f|D^k)}(f(\boldsymbol{x}_{k+1}) - M_{\mathrm{Bayes}}(P^k))^2] \\
&= \mathbb{E}_{D^k, \boldsymbol{x}_{k+1}}[\mathrm{Var}_{f \sim \mathcal{P}(f|D^k)}(f(\boldsymbol{x}_{k+1}))],
\end{aligned}
$$

and

$$
\begin{aligned}
R_{\mathrm{BG},k}(M) &:= \mathbb{E}_{P^k}[\{M_{\mathrm{Bayes}}(P^k) - M(P^k)\}^2] \\
&= \mathbb{E}_{P^k}[\{\mathbb{E}_{f \sim \mathcal{P}(f|D^k)}[f(\boldsymbol{x}_{k+1})] - M(P^k)\}^2].
\end{aligned}
$$

Hence, $R(M) = \frac{1}{p}\sum_{k=1}^{p} R_{\mathrm{PV},k} + \frac{1}{p}\sum_{k=1}^{p} R_{\mathrm{BG},k}(M) = R_{\mathrm{PV}} + R_{\mathrm{BG}}(M)$.

$\square$

*Proof of Theorem 3.2.* **Step 0 (clipping via a high-probability event).** Let $t_\varepsilon := \sigma_\varepsilon \sqrt{2\log(4p/\delta)}$ for $\delta \in (0, e^{-1})$, and define

$$
\mathcal{E} := \left\{ \max_{1 \le i \le p+1} |\varepsilon_i| \le t_\varepsilon \right\}.
$$

By sub-Gaussian tails and a union bound, $\Pr(\mathcal{E}^c) \le \delta$. On $\mathcal{E}$, writing $\boldsymbol{z}_i := (\boldsymbol{x}_i, y_i, \boldsymbol{x}_{k+1})$, we have $\boldsymbol{z}_i \in B(0, R_{rad})$ with radius $R_{rad} := C(B_X + B_f + t_\varepsilon)$, hence $\boldsymbol{z}_i \in \mathcal{Z}_R := B(0, R_{rad}) \subset \mathbb{R}^{2d_{\mathrm{feat}}+1}$ (compact). Rescale $\tilde{\boldsymbol{z}} := \boldsymbol{z}/(2R_{rad})$ so that $\mathrm{diam}(\tilde{\mathcal{Z}}_R) \le 1$.

**Step 1 (approximation & aggregation noise on $\mathcal{E}$).** Apply Lemma G.3 with the shared variable $\boldsymbol{c} := \boldsymbol{x}_{k+1}$ and $\boldsymbol{z}_i = (\boldsymbol{x}_i, y_i, \boldsymbol{x}_{k+1})$. With grid scale $\eta \asymp m^{-1/d_{\mathrm{eff}}}$, on $\mathcal{E}$, squared error is

$$
C_1 (2R_{rad})^{2\alpha} \eta^{2\alpha}.
$$

Since $R_{rad} \lesssim \sqrt{\log(p/\delta)}$, the factor $(2R_{rad})^{2\alpha}$ is polylogarithmic and is absorbed into $\tilde{O}(\cdot)$. Choosing $\eta \asymp m^{-1/d_{\mathrm{eff}}}$ gives $m^{-2\alpha/d_{\mathrm{eff}}}$ up to polylogarithmic factors.

**Step 2 (estimation error and combination).** From Lemma G.4, the estimation term is $\tilde{O}\left(\frac{m}{pN} + \frac{1}{N}\right)$. Combining with Step 1 gives

$$
m^{-\frac{2\alpha}{d_{\mathrm{eff}}}} + \frac{m}{pN} + \frac{1}{N}.
$$

Optimizing over $m$ yields the displayed rate (polylog factors absorbed into $\tilde{O}$).

**Step 3 (contribution of $\mathcal{E}^c$).** As in Step 7 of Lemma G.4's proof, using $(B_f + B_M)^2 + \sigma_\varepsilon^2$ as an envelope and sub-Gaussian tails, the contribution on $\mathcal{E}^c$ is $O(\delta + \delta \log(p/\delta))$. With $\delta := (pN)^{-2}$, this is negligible compared to the main terms.

$\square$

*Proof of Theorem 3.3.* Recall that $D^k = (\boldsymbol{x}_1, y_1, \ldots, \boldsymbol{x}_k, y_k)$. By the chain rule and the definition of $Z_{j,t}$,

$$
\begin{aligned}
\frac{p_j(D^k)}{p_{i^\star}(D^k)} &= \prod_{t=1}^{k} \frac{p_j(\boldsymbol{x}_t, y_t \mid D^{t-1})}{p_{i^\star}(\boldsymbol{x}_t, y_t \mid D^{t-1})} \quad (6) \\
&= \prod_{t=1}^{k} \frac{p_j(y_t \mid \boldsymbol{x}_t, D^{t-1})}{p_{i^\star}(y_t \mid \boldsymbol{x}_t, D^{t-1})} \\
&= \exp\left(\sum_{t=1}^{k} Z_{j,t}\right).
\end{aligned}
$$

Write $\pi_i(D^k) := \Pr(I = i \mid D^k)$ and $\mu_i(\boldsymbol{x}) := \mathbb{E}[f(\boldsymbol{x}) \mid I = i, D^k]$. By the law of total variance conditioning on $I$,

$$
\mathrm{Var}(f(\boldsymbol{x}) \mid D^k) = \underbrace{\mathbb{E}_{I|D^k}[\mathrm{Var}(f(\boldsymbol{x}) \mid I, D^k)]}_{(A)} \\
+ \underbrace{\mathrm{Var}_{I \sim \mathcal{P}_{I|D^k}}(\mu_I(\boldsymbol{x}))}_{(B)}. \quad (7)
$$

We compare the right-hand side with $\mathrm{Var}(f(\boldsymbol{x}) \mid I = i^\star, D^k)$.

**Step 1 (term (A)).** Using $|f(\boldsymbol{x})| \le B_f$,

$$
\begin{aligned}
&\left| \mathbb{E}_{I|D^k}[\mathrm{Var}(f(\boldsymbol{x}) \mid I, D^k)] - \mathrm{Var}(f(\boldsymbol{x}) \mid I = i^\star, D^k) \right| \\
&\le \sum_{j \ne i^\star} \pi_j(D^k) |\mathrm{Var}_j - \mathrm{Var}_{i^\star}| \\
&\le B_f^2 \sum_{j \ne i^\star} \pi_j(D^k),
\end{aligned}
$$

where $\mathrm{Var}_j := \mathrm{Var}(f(\boldsymbol{x}) \mid I = j, D^k) \le B_f^2$.

**Step 2 (term (B)).** For any $\mathcal{G}'_k$-measurable scalar $a$, $\mathrm{Var}_{I \sim \mathcal{P}_{I|D^k}}(\mu_I) \le \mathbb{E}_{I|D^k}(\mu_I - a)^2$. Choosing $a = \mu_{i^\star}(\boldsymbol{x})$ and using $|\mu_i(\boldsymbol{x})| \le B_f$,

$$
\begin{aligned}
\mathrm{Var}_{I \sim \mathcal{P}_{I|D^k}}(\mu_I(\boldsymbol{x})) &\le \sum_j \pi_j(D^k)(\mu_j(\boldsymbol{x}) - \mu_{i^\star}(\boldsymbol{x}))^2 \\
&\le 4B_f^2 \sum_{j \ne i^\star} \pi_j(D^k).
\end{aligned}
$$

Combining the two steps with (7), we obtain $\mathrm{Var}(f(\boldsymbol{x}) \mid D^k) \le \mathrm{Var}(f(\boldsymbol{x}) \mid I = i^\star, D^k) + 5B_f^2 \sum_{j \ne i^\star} \pi_j(D^k)$. Taking $\mathbb{E}_{\boldsymbol{x} \sim \mathcal{P}_X}$ and then $\mathbb{E}_{D^k|I=i^\star}$ yields

$$
\begin{aligned}
&\mathbb{E}_{D^k, \boldsymbol{x}|I=i^\star}[\mathrm{Var}_{f|D^k}\{f(\boldsymbol{x})\}] \\
&\le \mathbb{E}_{D^k, \boldsymbol{x}|I=i^\star}\left[\mathrm{Var}(f(\boldsymbol{x}) \mid I = i^\star, D^k)\right] \\
&\quad + 5B_f^2 \mathbb{E}_{D^k|I=i^\star}[1 - \pi_{i^\star}(D^k)]. \quad (8)
\end{aligned}
$$

**Step 3 (posterior concentration of the task index).** Let $S_{j,k} := \sum_{t=1}^{k} Z_{j,t}$ and $\lambda_{j,k} := e^{S_{j,k}}$. By the assumption, $\{Z_{j,t} + D_j\}$ are conditionally sub-exponential supermartingale differences. Applying a Bernstein-type supermartingale inequality (Theorem 2.6 in Fan et al., 2015), for each $j \neq i^\star$,

$$\Pr\left(S_{j,k} + kD_j \geq \tfrac{1}{2}kD_j \mid I = i^\star\right) \leq e^{-C_j k},$$

where $C_j := \frac{D_j^2}{8(\nu_j^2 + b_j D_j/2)}$. Hence, by a union bound, there is an event $\mathcal{E}_k := \left\{\lambda_{j,k} \leq e^{-D_j k/2} \forall j \neq i^\star\right\}$ with $\Pr(\mathcal{E}_k) \geq 1 - (T-1)e^{-Ck}$, where $C := \min_{j \neq i^\star} C_j$. On $\mathcal{E}_k$, using (6),

$$S_k := \sum_{j \neq i^\star} \frac{\alpha_j}{\alpha_{i^\star}} \lambda_{j,k} \leq \frac{1 - \alpha_{i^\star}}{\alpha_{i^\star}} e^{-D_{\min} k/2},$$

where $\pi_{i^\star}(D^k) = \frac{1}{1+S_k} \geq 1 - S_k$. Hence $1 - \pi_{i^\star}(D^k) \leq S_k$ on $\mathcal{E}_k$, while trivially $1 - \pi_{i^\star}(D^k) \leq 1$ on $\mathcal{E}_k^{\mathsf{c}}$. Therefore

$$\mathbb{E}_{D^k|i^\star}\left[1 - \pi_{i^\star}(D^k)\right] \leq \frac{1 - \alpha_{i^\star}}{\alpha_{i^\star}} e^{-D_{\min} k/2} + (T-1)e^{-Ck}.$$

**Step 4 (conclusion).** Plug the last inequality into (8) to obtain the displayed bound for $\mathbb{E}_{D^k, \boldsymbol{x}|I=i^\star}\left[\mathrm{Var}_{f|D^k}\{f(\boldsymbol{x})\}\right]$. Finally, apply Lemma G.1 to bound $\mathbb{E}_{D^k, \boldsymbol{x}|I=i^\star}\left[\mathrm{Var}\left(f(\boldsymbol{x}) \mid I = i^\star, D^k\right)\right]$ by $\inf_M \sup_{f \in F_{i^\star}} \mathbb{E}_{P^k}\left[\left(f(\boldsymbol{x}_{k+1}) - M(P^k)\right)^2 \mid f\right]$. $\square$

# I. Proofs of the Technical Lemmas

*Proof of Lemma G.1.* Define the MSE at step $k$ under $f$, $r_k(M, f) := \mathbb{E}_{P^k}\left[\left(f(\boldsymbol{x}_{k+1}) - M(P^k)\right)^2 \mid f\right]$, and the minimax risk at step $k$ for the true task type $R_k^\star(F_{i^\star}) := \inf_M \sup_{f \in F_{i^\star}} r_k(M, f)$. For any fixed $M$ and any measure $\Pi$ supported on $F_{i^\star}$,

$$\sup_{f \in F_{i^\star}} r_k(M, f) \geq \int r_k(M, f) \mathrm{d}\Pi(f).$$

Taking $\Pi = \mathcal{P}_{F_{i^\star}}$ and then infimum over $M$,

$$R_k^\star(F_{i^\star}) \geq \inf_M \int r_k(M, f) \mathrm{d}\mathcal{P}_{F_{i^\star}}(f).$$

By Tonelli's theorem and the tower property, $\int r_k(M, f) \mathrm{d}\mathcal{P}_{F_{i^\star}}(f) = \mathbb{E}_{D^k, \boldsymbol{x}_{k+1}|I=i^\star}\left[\mathbb{E}_{f \sim \mathcal{P}_{f|I=i^\star, D^k}}\left[\left(f(\boldsymbol{x}_{k+1}) - M(P^k)\right)^2\right]\right]$. Since $M(P^k)$ is $\mathcal{G}_k'$-measurable, the inner expectation is minimized pointwise (for each realized $D^k, \boldsymbol{x}_{k+1}$) by the posterior mean $\mathbb{E}\left[f(\boldsymbol{x}_{k+1}) \mid I = i^\star, D^k, \boldsymbol{x}_{k+1}\right]$, and its minimum value is $\mathrm{Var}\left(f(\boldsymbol{x}_{k+1}) \mid I = i^\star, D^k\right)$. Therefore $\inf_M \int r_k(M, f) \mathrm{d}\mathcal{P}_{F_{i^\star}}(f) = \mathbb{E}_{f \sim \mathcal{P}_{F_{i^\star}}, D^k \sim \mathcal{P}_{X,Y|f}^{\otimes k}, \boldsymbol{x}_{k+1} \sim \mathcal{P}_X}\left[\mathrm{Var}_{f \sim \mathcal{P}_{F_{i^\star}|D^k}}(f(\boldsymbol{x}_{k+1}))\right]$ holds. $\square$

*Proof of Lemma G.2.* Note that by the definition of the decoder network and the clipping operation, the decoder satisfies the uniformly Lipschitz condition in both arguments; biases do not affect Lipschitz constants, ReLU and clipping are 1-Lipschitz, and hence the constants can be controlled by the spectral product of the decoder.

*Step 1: Contraction via triangle inequality.* Let $\mathcal{S} := \{P^k \mapsto \frac{1}{k} \sum_{i=1}^{k} \phi_\theta(\boldsymbol{u}_i)\}$ and $\mathcal{R} := \{(\boldsymbol{s}, \boldsymbol{c}) \mapsto \rho_\theta(\boldsymbol{s}, \boldsymbol{c}) : (\boldsymbol{s}, \boldsymbol{c}) \in \Delta^{m-1} \times \mathcal{C}\}$. For any two predictors $\rho_\theta \circ S_\theta$ and $\rho_{\theta'} \circ S_{\theta'}$ evaluated along a predictable tree $z$, the $(L_s, L_c)$–Lipschitz property of $\rho_\theta$ in $\boldsymbol{s}$ and the triangle inequality give

$$\begin{aligned}
&\left|\rho_\theta(S_\theta(P^t), \boldsymbol{c}_t) - \rho_{\theta'}(S_{\theta'}(P^t), \boldsymbol{c}_t)\right| \\
&\leq L_s \left\|S_\theta(P^t) - S_{\theta'}(P^t)\right\|_2 \\
&\quad + \left|\rho_\theta(S_{\theta'}(P^t), \boldsymbol{c}_t) - \rho_{\theta'}(S_{\theta'}(P^t), \boldsymbol{c}_t)\right|.
\end{aligned}$$

Consequently, a $(\delta/(2L_s))$–cover of the pooled-feature class $\mathcal{S}$ together with a $(\delta/2)$–cover of the decoder outputs $\mathcal{R}$ produces a $\delta$–cover of the composite class $\{\rho_\theta \circ S_\theta\}$ under the $\ell_2$ sequential metric. Equivalently,

$$\begin{aligned}
&\sup_z \log N_2^{\mathrm{seq}}\left(\delta, \{\rho_\theta \circ S_\theta\}; z\right) \\
&\leq \sup_z \log N_2^{\mathrm{seq}}\left(\tfrac{\delta}{2L_s}, \mathcal{S}; z\right) + \sup_z \log N_2^{\mathrm{seq}}\left(\tfrac{\delta}{2}, \mathcal{R}; z\right).
\end{aligned}$$

This single reduction step subsumes the earlier contraction and triangle-inequality arguments and will be followed by separate bounds for $\mathcal{S}$ (Step 2) and $\mathcal{R}$ (Step 3).

*Step 2: Cover of the pooled features $\mathcal{S}$.* Let $\Phi = \{\phi_\theta : \theta \in \Theta\}$. For any $\theta, \theta' \in \Theta$ and any prompt $P^k$,

$$\begin{aligned}
\|S_\theta(P^k) - S_{\theta'}(P^k)\|_2 &= \left\|\frac{1}{k} \sum_{i=1}^{k} (\phi_\theta(\boldsymbol{u}_i) - \phi_{\theta'}(\boldsymbol{u}_i))\right\|_2 \\
&\leq \sup_{\boldsymbol{u} \in \mathcal{U}} \|\phi_\theta(\boldsymbol{u}) - \phi_{\theta'}(\boldsymbol{u})\|_2.
\end{aligned}$$

Fix $\eta \in (0, 1)$ and set $r := \eta/(4L_\phi)$, where $L_\phi := \mathrm{Lip}(\phi_\theta)$. Take an $r$-net $\mathcal{N} \subset \mathcal{U}$ of input space of $\phi_\theta$ with

$$\begin{aligned}
|\mathcal{N}| &\leq C(d_{\mathrm{eff}}) \left(\frac{\mathrm{diam}(\mathcal{U})}{r}\right)^{d_{\mathrm{eff}}} \\
&= C(d_{\mathrm{eff}}) \left(\frac{4L_\phi \mathrm{diam}(\mathcal{U})}{\eta}\right)^{d_{\mathrm{eff}}}.
\end{aligned}$$

By triangle inequality and Lipschitzness, for every $\boldsymbol{u} \in \mathcal{U}$, there exists $\boldsymbol{u}' \in \mathcal{N}$ such that

$$\begin{aligned}
\|\phi_\theta(\boldsymbol{u}) - \phi_{\theta'}(\boldsymbol{u})\|_2 &\leq \|\phi_\theta(\boldsymbol{u}') - \phi_{\theta'}(\boldsymbol{u}')\|_2 + 2L_\phi r \\
&\leq \|\phi_\theta(\boldsymbol{u}') - \phi_{\theta'}(\boldsymbol{u}')\|_2 + \eta/2.
\end{aligned}$$

Hence a cover of $\{\phi_\theta(\cdot)\}$ on $\mathcal{N}$ at scale $\eta/2$ yields a uniform cover on $\mathcal{U}$ at scale $\eta$.

Note that

$$\log N_{\infty,2}(\eta, \Phi; \mathcal{N})$$
$$\leq \sum_{j=1}^{m} \log N_{\infty}\left(\frac{\eta}{\sqrt{m}}, \Phi_j; \mathcal{N}\right)$$
$$\leq \sum_{j=1}^{m} \text{Pdim}(\Phi_j) \log \frac{C|\mathcal{N}|\sqrt{m}}{\eta}.$$

From Anthony & Bartlett (1999); Bartlett et al. (2019), using $\text{Pdim}(\Phi) = \tilde{O}(m)$ for the coordinate-wise $[0,1]$-bounded ReLU features, the finite-set (size $|\mathcal{N}|$) covering bound gives

$$\log N_{\infty,2}\left(\frac{\eta}{2}, \Phi; \mathcal{N}\right)$$
$$\lesssim \text{Pdim}(\Phi)\left[\log\left(\frac{C\sqrt{m}}{\eta}\right) + d_{\text{eff}}\log\left(\frac{C' L_\phi \text{diam}(\mathcal{U})}{\eta}\right)\right]$$
$$\lesssim m\left[\log\left(\frac{C\sqrt{m}}{\eta}\right) + d_{\text{eff}}\log\left(\frac{C' L_\phi \text{diam}(\mathcal{U})}{\eta}\right)\right].$$

Substituting $\eta = \delta/(2L_s)$ from Step 1 yields the sequential bound

$$\sup_z \log N_2^{\text{seq}}\left(\frac{\delta}{2L_s}, \mathcal{S}; z\right)$$
$$\lesssim m\left[\log\left(\frac{\tilde{C} L_s \sqrt{m}}{\delta}\right) + d_{\text{eff}}\log\left(\frac{\tilde{C}' L_\phi \text{diam}(\mathcal{U}) L_s}{\delta}\right)\right],$$

uniformly in $z$.

*Step 3: Uniform cover of the decoder $\mathcal{R}$.* Fix a predictable input tree $z = \{(s_t(\xi_{1:t-1}), c_t(\xi_{1:t-1}))\}_{t \leq k}$ with nodes in $\Delta^{m-1} \times \mathcal{C}$. Fix $\delta \in (0, 2B_M]$ and build a uniform grid on the output range $\mathcal{G} := \{-B_M, -B_M + \delta, -B_M + 2\delta, \ldots, -B_M + J\delta\}$, $J := \lceil \frac{2B_M}{\delta} \rceil$, so that for any $y \in [-B_M, B_M]$ there exists $q(y) \in \mathcal{G}$ with $|y - q(y)| \leq \delta/2$. Now consider the family $\mathcal{V}$ of depth-wise constant predictable trees $v = \{v_t\}_{t \leq k}$ defined by choosing, independently for each depth $t$, a grid value $g_t \in \mathcal{G}$ and setting $v_t(\cdot) \equiv g_t$ (constant on all nodes at depth $t$). Then $|\mathcal{V}| = |\mathcal{G}|^k = (J+1)^k$.

Fix any decoder $\rho_\theta \in \mathcal{R}$ and any path $\xi \in \{\pm 1\}^k$. Along this path, we observe the length-$k$ sequence of decoder outputs $y_t := \rho_\theta(s_t(\xi_{1:t-1}), c_t(\xi_{1:t-1})) \in [-B_M, B_M]$. Define the depth-wise grid sequence $g_t := q(y_t) \in \mathcal{G}$ and take the corresponding $v^\star \in \mathcal{V}$ with $v_t^\star(\cdot) \equiv g_t$. Then, along the path $\xi$,

$$\frac{1}{k}\sum_{t=1}^{k}(v_t^\star(\xi_{1:t-1}) - y_t)^2 \leq \frac{1}{k}\sum_{t=1}^{k}\left(\frac{\delta}{2}\right)^2 = \left(\frac{\delta}{2}\right)^2,$$

that is, $d_{2,\xi}(\rho_\theta \circ z, v^\star; z) \leq \delta/2$. Since this holds for every $\rho_\theta$ and every path $\xi$, the set $\mathcal{V}$ is a sequential $(\delta/2)$–cover of $\mathcal{R}$ on $z$. Therefore,

$$N_2^{\text{seq}}\left(\frac{\delta}{2}, \mathcal{R}; z\right) \leq |\mathcal{V}| = (J+1)^k \leq \left(\frac{2B_M}{\delta} + 2\right)^k.$$

Taking logarithms yields

$$\sup_z \log N_2^{\text{seq}}\left(\frac{\delta}{2}, \mathcal{R}; z\right) \leq k\log\left(\frac{2B_M}{\delta} + 2\right)$$
$$\lesssim k\log\left(\frac{CB_M}{\delta}\right).$$

$\square$

*Proof of Lemma G.3.* We will write $C, C(d), \ldots$ for positive constants depending only on displayed arguments. Note that, w.r.t. $\ell_2$, the renormalization layer with parameter $\tau$ has Lipschitz constant $L_{\text{renorm}} \leq \frac{2\sqrt{m}}{\tau}$. Since ReLU is 1-Lipschitz and biases do not affect Lipschitz constants, the global Lipschitz modulus satisfies $\text{Lip}(\mathcal{T}_\theta) \leq S(\mathcal{T}_\theta)$ for ReLU network $\mathcal{T}_\theta$.

**Step 1 (feature map: soft histogram).** Fix

$$\delta := \left(\frac{\eta}{8\sqrt{d_{\text{eff}}}}\right)^{1/\alpha} \in (0, 1), \qquad r := \delta/4.$$

Let $U \supset \mathcal{U}$ be an axis-aligned cube with $\text{dist}(\mathcal{U}, \partial U) \geq r$, where $\text{dist}(\mathcal{U}, \partial U) := \inf\{\|\boldsymbol{u} - \boldsymbol{u}'\| : \boldsymbol{u} \in \mathcal{U}, \boldsymbol{u}' \in \partial U\}$ denotes the Euclidean distance between $\mathcal{U}$ and the boundary of $U$. Partition $U$ into a regular grid of closed cubes $\{Q_j\}_{j=1}^{m}$ of side length $\delta$, so that $m \asymp \delta^{-d_{\text{eff}}}$; denote by $\boldsymbol{q}_j$ the center of $Q_j$ and set the representative point

$$\boldsymbol{r}_j \in \arg\min_{\boldsymbol{u} \in \mathcal{U}} \|\boldsymbol{u} - \boldsymbol{q}_j\|_2.$$

Let $\kappa \in C_c^\infty(\mathbb{R}^{d_{\text{eff}}})$ be a nonnegative and radially symmetric mollifier with $\int \kappa = 1$ and $\text{supp}\,\kappa \subset B(0,1)$. Put $\kappa_r(\boldsymbol{x}) := r^{-d_{\text{eff}}}\kappa(\boldsymbol{x}/r)$ and define

$$\phi_j(\boldsymbol{x}) := (\mathbf{1}_{Q_j} * \kappa_r)(\boldsymbol{x}).$$

Then $\text{supp}\,\phi_j \subset Q_j^+ := \{\boldsymbol{q} : \text{dist}(\boldsymbol{q}, Q_j) \leq r\}$. Since the pairwise intersections of the grid cells have Lebesgue measure zero, we have $\sum_j \mathbf{1}_{Q_j} = \mathbf{1}_U$ almost everywhere, and because $B(\boldsymbol{x}, r) \subset U$ for all $\boldsymbol{x} \in \mathcal{U}$, convolution with the unit-mass mollifier ignores these measure-zero discrepancies, yielding $\sum_j \phi_j(\boldsymbol{x}) = (\sum_j \mathbf{1}_{Q_j}) * \kappa_r(\boldsymbol{x}) = \mathbf{1}_U * \kappa_r(\boldsymbol{x}) = 1$ pointwise on $\mathcal{U}$. Also, by Young's inequality, $\|\nabla\phi_j\|_\infty \leq \|\mathbf{1}_{Q_j}\|_\infty \|\nabla\kappa_r\|_1 = \|\nabla\kappa\|_1 r^{-1}$. Since $r = \delta/4$, we get $\|\nabla\phi_j\|_\infty \leq (4\|\nabla\kappa\|_1)\delta^{-1} =: C\delta^{-1}$, uniformly in $j$. For $\boldsymbol{u}_{1:k} \in \mathcal{U}^k$, define the soft histogram

$$s_j := \frac{1}{k}\sum_{i=1}^{k}\phi_j(\boldsymbol{u}_i), \qquad \boldsymbol{s} = (s_1, \ldots, s_m) \in \Delta^{m-1}.$$

**Step 2 (decoder construction).** For each fixed $\boldsymbol{c}$, define the ground cost on indices by

$$c^{(u)}(j, \ell) := \|\boldsymbol{r}_j - \boldsymbol{r}_\ell\|_2^\alpha, \qquad 0 < \alpha \leq 1,$$

and let $W_\alpha^{(u)}$ be the discrete 1-Wasserstein distance on the simplex $\Delta^{m-1} = \{s \in [0,1]^m : \sum_j s_j = 1\}$ with cost $c^{(u)}$:

$$W_\alpha^{(u)}(s,t) := \min_{\pi \geq 0} \sum_{j,\ell} c^{(u)}(j,\ell) \pi_{j\ell}$$

$$\text{s.t.} \quad \sum_\ell \pi_{j\ell} = s_j, \quad \sum_j \pi_{j\ell} = t_\ell.$$

where $s, t \in \Delta^{m-1}$. Note that $c^{(u)}$ is a metric since $0 < \alpha \leq 1$. Let $\Delta_k := \{\frac{n}{k} : n \in \{0, \ldots, k\}^m, \sum_j n_j = k\}$. For $v = n/k \in \Delta_k$, define

$$\rho_c(v)$$
$$:= M_{\text{Bayes}}(\underbrace{(r_1, c), \ldots, (r_1, c)}_{n_1}, \ldots, \underbrace{(r_m, c), \ldots, (r_m, c)}_{n_m}).$$

This is well-defined by permutation invariance of $M_{\text{Bayes}}$.

Let $s = n/k$ and $t = n'/k$ be points of $\Delta_k$. Construct an integer matrix $A = (A_{j\ell})$ with row sums $n$ and column sums $n'$ (e.g., by the Northwest corner rule (Peyré & Cuturi, 2019)), and set $\pi := A/k$. Then $\pi \in \Pi(s,t)$ is a feasible transport plan. Enumerating the $k$ pairs so that $(r_{j(i)}, r_{\ell(i)})$ appears exactly $A_{j\ell}$ times, the Hölder condition yields

$$|\rho_c(s) - \rho_c(t)| \leq \frac{L}{k} \sum_{i=1}^k \|r_{j(i)} - r_{\ell(i)}\|_2^\alpha$$
$$= L \sum_{j,\ell} c^{(u)}(j,\ell) \frac{A_{j\ell}}{k}$$
$$= L \sum_{j,\ell} c^{(u)}(j,\ell) \pi_{j\ell},$$

where $c^{(u)}(j,\ell) := \|r_j - r_\ell\|_2^\alpha$. Since this bound holds for $\pi^* \in \Pi(s,t)$,

$$|\rho_c(s) - \rho_c(t)| \leq L \, W_\alpha^{(u)}(s,t), \tag{9}$$

which proves the $L$-Lipschitz property on $\Delta_k$.

Extend to all $s \in \Delta^{m-1}$ by the McShane-type formula

$$\rho_c^\star(s) := \inf_{v \in \Delta_k} \left\{ \rho_c(v) + L W_\alpha^{(u)}(s,v) \right\}, \tag{10}$$

which satisfies $\rho_c^\star(v) = \rho_c(v)$ for $v \in \Delta_k$ and, by the inequality (9), the Lipschitz property

$$|\rho_c^\star(s) - \rho_c^\star(t)| \leq L W_\alpha^{(u)}(s,t) \qquad (\forall s, t).$$

By this construction, $\rho_c^\star(v) = \rho_c(v)$ holds. Indeed, for $v \in \Delta_k$, taking $t = v$ in (10) gives $\rho_c^\star(v) \leq \rho_c(v)$. Conversely, the inequality (9) implies $\rho_c(v) \leq \rho_c(t) + L W_\alpha^{(u)}(t,v)$ for every $t \in \Delta_k$, hence $\rho_c(v) \leq \inf_t \{\rho_c(t) + L W_\alpha^{(u)}(v,t)\} = \rho_c^\star(v)$. Therefore $\rho_c^\star(v) = \rho_c(v)$.

We next show its $L$-Lipschitzness. For any $s, t$ and any $v \in \Delta_k$, the triangle inequality yields $W_\alpha^{(u)}(s,v) \leq W_\alpha^{(u)}(s,t) + W_\alpha^{(u)}(t,v)$. Taking infima over $v$, $\rho_c^\star(s) \leq \rho_c^\star(t) + L W_\alpha^{(u)}(s,t)$ and $\rho_c^\star(t) \leq \rho_c^\star(s) + L W_\alpha^{(u)}(s,t)$, so $|\rho_c^\star(s) - \rho_c^\star(t)| \leq L W_\alpha^{(u)}(s,t)$.

We also note its piecewise linearity. By the Kantorovich–Rubinstein dual (Peyré & Cuturi, 2019) on a finite space,

$$W_\alpha^{(u)}(s,v) = \sup_{\varphi \in \mathbb{R}^m : |\varphi_j - \varphi_\ell| \leq c^{(u)}(j,\ell)} \langle \varphi, s - v \rangle,$$

so $s \mapsto \rho_c^\star(s)$ is the lower envelope of finitely many support functions and thus piecewise linear on $\Delta^{m-1}$.

**Step 3 (error decomposition and bounds).** Adopt a half-open tie-breaking so that each $u_i$ belongs to a unique cell $Q_{j(i)}$. Let the hard histogram be $h := \frac{1}{k}(n_1^{\text{hard}}, \ldots, n_m^{\text{hard}})$ with $n_j^{\text{hard}} := \#\{i : u_i \in Q_j\}$. Then, with $z_i = (u_i, c)$ and using the Hölder condition while keeping $c$ fixed,

$$|M_{\text{Bayes}}(u_{1:k}, c) - \rho_\theta(s, c)|$$
$$\leq \underbrace{\left| M_{\text{Bayes}}(u_{1:k}, c) - M_{\text{Bayes}}\left((r_{j(1)}, c), \ldots, (r_{j(k)}, c)\right) \right|}_{\text{quantization in } u}$$
$$+ \underbrace{|\rho_c^\star(h) - \rho_c^\star(s)|}_{\text{hard-to-soft transport}} + \underbrace{|\rho_c^\star(s) - \rho_\theta(s, c)|}_{\text{network approximation}}.$$

*Quantization*: $\|u_i - r_{j(i)}\|_2 \leq \sqrt{d_{\text{eff}}} \delta$, the Hölder condition gives

$$\left| M_{\text{Bayes}}(u_{1:k}, c) - M_{\text{Bayes}}\left((r_{j(1)}, c), \ldots, (r_{j(k)}, c)\right) \right|$$
$$\leq \frac{L}{k} \sum_{i=1}^k \|u_i - r_{j(i)}\|_2^\alpha$$
$$\leq C(d_{\text{eff}}) L \delta^\alpha.$$

Moreover, $M_{\text{Bayes}}\left((r_{j(1)}, c), \ldots, (r_{j(k)}, c)\right) = \rho_c(h) = \rho_c^\star(h)$.

*Transport*: Define a coupling $\pi$ between $h$ and $s$ by moving, for each $i$, the mass $1/k$ placed at $r_{j(i)}$ to the mixture $\sum_{j=1}^m \phi_j(u_i) \delta_{r_j}$:

$$\pi_{j(i) \to j}^{(i)} := \frac{1}{k} \phi_j(u_i), \qquad \pi := \sum_{i=1}^k \sum_{j=1}^m \pi_{j(i) \to j}^{(i)}.$$

Because $\sum_j \phi_j \equiv 1$, $\pi$ has marginals $h$ and $s$, hence is feasible for $W_\alpha^{(u)}$. If $\phi_j(u_i) > 0$ then $u_i \in Q_j^+$, and by the triangle inequality together with Step 1,

$$\|r_{j(i)} - r_j\|_2 \leq \|r_{j(i)} - u_i\|_2 + \|u_i - r_j\|_2 \leq C(d_{\text{eff}}) \delta.$$

Therefore, with $W_\alpha^{(u)}$,

$$W_\alpha^{(u)}(h, s) \leq \sum_{i=1}^k \sum_{j=1}^m \pi_{j(i) \to j}^{(i)} \|r_{j(i)} - r_j\|_2^\alpha \leq C(d_{\text{eff}}) \delta^\alpha,$$

and since $\rho_{\boldsymbol{c}}^{\star}$ is $L$-Lipschitz w.r.t. $W_{\alpha}^{(u)}$,

$$|\rho_{\boldsymbol{c}}^{\star}(\boldsymbol{h}) - \rho_{\boldsymbol{c}}^{\star}(\boldsymbol{s})| \leq L W_{\alpha}^{(u)}(\boldsymbol{h}, \boldsymbol{s}) \leq C(d_{\mathrm{eff}}) L \delta^{\alpha}.$$

Combining the three bounds and using $\mathrm{diam}(\mathcal{U}) \leq 1$ (so that $c^{(u)}(j, \ell) \leq 1$ and $W_{\alpha}^{(u)} \leq \mathrm{TV} = \frac{1}{2}\|\cdot\|_1$), we obtain

$$|M_{\mathrm{Bayes}}(\boldsymbol{u}_{1:k}, \boldsymbol{c}) - \rho_{\theta}(\boldsymbol{s}, \boldsymbol{c})|$$
$$\leq C(d_{\mathrm{eff}}) L \delta^{\alpha} + |\rho_{\boldsymbol{c}}^{\star}(\boldsymbol{s}) - \rho_{\theta}(\boldsymbol{s}, \boldsymbol{c})|.$$

Finally choose $\rho_{\theta}$ so that $\sup_{(\boldsymbol{s}, \boldsymbol{c})} |\rho_{\boldsymbol{c}}^{\star}(\boldsymbol{s}) - \rho_{\theta}(\boldsymbol{s}, \boldsymbol{c})| \leq CL\delta^{\alpha}$ (Step 4(iii)). Then

$$\sup_{\boldsymbol{c}} \sup_{\boldsymbol{u}_{1:k} \in \mathcal{U}^k} |M_{\mathrm{Bayes}}(\boldsymbol{u}_{1:k}, \boldsymbol{c}) - \rho_{\theta}(\boldsymbol{s}, \boldsymbol{c})| \leq C(d_{\mathrm{eff}}) L \delta^{\alpha}.$$

Choosing $\delta \asymp \eta^{1/\alpha}$ and $m \asymp \delta^{-d_{\mathrm{eff}}}$ yields the claimed bound $C(d_{\mathrm{eff}}) L \eta$.

**Step 4 (Neural implementation).** We first consider the joint regularity of $(\boldsymbol{s}, \boldsymbol{c}) \mapsto \rho_{\boldsymbol{c}}^{\star}(\boldsymbol{s})$ on the compact domain $\Delta^{m-1} \times \mathcal{C}$.

*(i) Joint Lipschitz in $(\boldsymbol{s}, \boldsymbol{c})$.* By Step 2, for each fixed $\boldsymbol{c}$ and all $\boldsymbol{s}, \boldsymbol{s}' \in \Delta^{m-1}$,

$$|\rho_{\boldsymbol{c}}^{\star}(\boldsymbol{s}) - \rho_{\boldsymbol{c}}^{\star}(\boldsymbol{s}')| \leq L W_{\alpha}^{(u)}(\boldsymbol{s}, \boldsymbol{s}').$$

On the simplex, we have $W_{\alpha}^{(u)}(\boldsymbol{s}, \boldsymbol{s}') \leq \frac{\mathrm{diam}(\mathcal{U})^{\alpha}}{2}\|\boldsymbol{s} - \boldsymbol{s}'\|_1 \leq \frac{\mathrm{diam}(\mathcal{U})^{\alpha}}{2}\sqrt{m}\|\boldsymbol{s} - \boldsymbol{s}'\|_2$: it follows from the trivial plan that transports the total variation mass across at most $\mathrm{diam}(\mathcal{U})^{\alpha}$. Since $\mathrm{diam}(\mathcal{U}) \leq 1$,

$$|\rho_{\boldsymbol{c}}^{\star}(\boldsymbol{s}) - \rho_{\boldsymbol{c}}^{\star}(\boldsymbol{s}')| \leq CL\sqrt{m}\|\boldsymbol{s} - \boldsymbol{s}'\|_2.$$

Next, fix $\boldsymbol{s}$ and vary $\boldsymbol{c}, \boldsymbol{c}'$. From the Hölder assumption on $M_{\mathrm{Bayes}}$ applied to $\boldsymbol{z}_i = (\boldsymbol{r}_{j(i)}, \boldsymbol{c})$ and $\boldsymbol{z}_i' = (\boldsymbol{r}_{j(i)}, \boldsymbol{c}')$ we obtain $|\rho_{\boldsymbol{c}}(\boldsymbol{v}) - \rho_{\boldsymbol{c}'}(\boldsymbol{v})| \leq L\|\boldsymbol{c} - \boldsymbol{c}'\|_2^{\alpha}$ for all $\boldsymbol{v} \in \Delta_k$. By the McShane envelope (10), $(\boldsymbol{s}, \boldsymbol{c}) \mapsto \rho_{\boldsymbol{c}}^{\star}(\boldsymbol{s})$ is $\alpha$-Hölder in $\boldsymbol{c}$:$|\rho_{\boldsymbol{c}}^{\star}(\boldsymbol{s}) - \rho_{\boldsymbol{c}'}^{\star}(\boldsymbol{s})| \leq L\|\boldsymbol{c} - \boldsymbol{c}'\|_2^{\alpha}$. To meet the size of networks in Definition 2.2, we first apply a McShane-type $\alpha$-Hölder extension to the whole space $\mathbb{R}^{d_{\mathrm{feat}}}$, and then convolve only in the $\boldsymbol{c}$-direction with a standard mollifier (Appendix C.5 in Evans, 2010) $\eta_h$. This yields, for any $(\boldsymbol{s}, \boldsymbol{c})$, $|\rho_{\boldsymbol{c}}^{\star}(\boldsymbol{s}) - \rho_{\boldsymbol{c}}^{\sharp}(\boldsymbol{s})| \leq |\int (\rho_{\boldsymbol{c}}^{\star}(\boldsymbol{s}) - \rho_{\boldsymbol{c}-h\boldsymbol{z}}^{\star}(\boldsymbol{s})) \eta(\boldsymbol{z}) \, d\boldsymbol{z}| \leq \int L\|h\boldsymbol{z}\|^{\alpha} \eta(\boldsymbol{z}) \, d\boldsymbol{z} \leq C_{\eta} L h^{\alpha}$, and $\mathrm{Lip}_{\boldsymbol{c}}(\rho^{\sharp}) \lesssim h^{\alpha-1}$ uniformly in $(\boldsymbol{s}, \boldsymbol{c})$. In what follows we approximate $\rho^{\sharp}$ by a ReLU network and keep the same notation $\rho_{\theta}$.

*(ii) ReLU approximation of the feature map.* As in the current proof, each $\boldsymbol{u} \mapsto \phi_j(\boldsymbol{u})$ is $C^{\infty}$ on $[0, 1]^{d_{\mathrm{eff}}}$ with $\|\nabla \phi_j\|_{\infty} \lesssim \delta^{-1}$, hence by ReLU approximation (Yarotsky, 2017) there exists a ReLU network of depth $O(\log(1/\eta_{\phi}))$ and size $O(m \log(1/\eta_{\phi}))$ that uniformly approximates $\phi = (\phi_1, \ldots, \phi_m)$ on $\mathcal{U}$ with error $\eta_{\phi} \in (0, e^{-1})$. Additionally, we set spectral product

$$S(\phi_{\theta}) \asymp \delta^{-1} = m^{1/d_{\mathrm{eff}}},$$

matching size of the Transformer in Definition 2.2. After applying the fixed renormalization layer $\mathrm{Renorm}_{\tau} : \mathbb{R}^m \to \Delta^{m-1}$, the features are simplex-valued.

*(iii) ReLU approximation of the decoder.* On the compact set $\Delta^{m-1} \times \mathcal{C}$, the map $(\boldsymbol{s}, \boldsymbol{c}) \mapsto \rho_{\boldsymbol{c}}^{\sharp}(\boldsymbol{s})$ is jointly Lipschitz with moduli $(L_s, L_c)$ from (i), and for each fixed $\boldsymbol{c}$ it is piecewise-linear in $\boldsymbol{s}$ (lower envelope of affine forms by the KR dual). Therefore, by standard approximation results for Lipschitz targets on a compact domain (Yarotsky, 2017), there exists a ReLU network $\rho_{\theta} : \Delta^{m-1} \times \mathcal{C} \to \mathbb{R}$ such that

$$\sup_{(\boldsymbol{s}, \boldsymbol{c})} |\rho_{\boldsymbol{c}}^{\sharp}(\boldsymbol{s}) - \rho_{\theta}(\boldsymbol{s}, \boldsymbol{c})| \leq CL\delta^{\alpha}.$$

Moreover, by spectral normalization of the linear layers, we can enforce $\mathrm{Lip}_s(\rho_{\theta}) \leq cL_s = cCL\sqrt{m}$ and $\mathrm{Lip}_{\boldsymbol{c}}(\rho_{\theta}) \leq cL_c = cL\delta^{\alpha-1}$, so the decoder's spectral product can be taken as

$$S(\rho_{\theta}) = O\big(L\sqrt{m} + L\delta^{\alpha-1}\big)$$

under the $\ell_2$-metric used. Note that $\delta^{\alpha-1} = O(m^{(1-\alpha)/d_{\mathrm{eff}}}) = O(\sqrt{m})$ as $d_{\mathrm{eff}} \geq 2$. Note that the number of parameters of the decoder does not affect the upper bound of the predictive risk in Theorem 3.2. Instead, we evaluate the complexity regarding the decoder by counting the number of $\delta$-cubes to cover the space of length-$k$ sequences (see proof of Lemma G.2, Step 3).

Finally, combining (ii)–(iii) with Step 3 and taking $\eta_{\phi} = 1/m$, we obtain

$$\sup_{\boldsymbol{c}} \sup_{\boldsymbol{u}_{1:k} \in \mathcal{U}^k} \left| M_{\mathrm{Bayes}}(\boldsymbol{u}_{1:k}, \boldsymbol{c}) - \rho_{\theta}\Big(\frac{1}{k}\sum_{i=1}^{k}\phi(\boldsymbol{u}_i), \boldsymbol{c}\Big) \right|$$
$$\leq C(d_{\mathrm{eff}}) L \delta^{\alpha}.$$

Choosing $\delta \asymp \eta^{1/\alpha}$ and $m \asymp \delta^{-d_{\mathrm{eff}}}$ yields the lemma. $\square$

*Proof of Lemma G.4.* Recall that we work on standard Borel measurable spaces (Borel $\sigma$-fields of Polish spaces), so regular conditional probabilities exist; see, e.g., Durrett (2019). Consequently, there exists a (measurable) version of the posterior probability kernel $D^k \mapsto \mathrm{Pr}(f \in \cdot \mid D^k)$, unique $\mathrm{Pr}$-a.s.; we fix one such version (extending it arbitrarily on a $\mathrm{Pr}$-null set) once and for all. All conditioning statements below, including $\mathbb{E}[f(\boldsymbol{x}_{k+1}) \mid D^k]$ and $\mathrm{Var}(f(\boldsymbol{x}_{k+1}) \mid D^k)$, are taken with respect to this fixed version.

A technical point concerns the measurability of suprema over the parameter space $\Theta$, which is required for expectations to be well-defined. Note that under our assumptions, the parameter space $\Theta$ is separable and, for any fixed sample, $\theta \mapsto (y - M_{\theta}(P))^2$ is continuous, so the relevant random suprema are measurable.

**Step 1 (Reduction via a centered, Bayes-offset objective).**
For each block $j$, write

$$\Lambda_j(\theta) := \frac{1}{p} \sum_{k=1}^{p} \left(y_{j,k+1} - M_\theta(P_j^k)\right)^2$$
$$= A_j(\theta) + B_j(\theta) + C_j,$$

where

$$A_j(\theta) := \frac{1}{p} \sum_{k=1}^{p} \left(M_{\mathrm{Bayes}}(P_j^k) - M_\theta(P_j^k)\right)^2,$$

$$B_j(\theta) := \frac{2}{p} \sum_{k=1}^{p} \left(y_{j,k+1} - M_{\mathrm{Bayes}}(P_j^k)\right)$$
$$\times \left(M_{\mathrm{Bayes}}(P_j^k) - M_\theta(P_j^k)\right),$$

and $C_j := \frac{1}{p} \sum_{k=1}^{p} \left(y_{j,k+1} - M_{\mathrm{Bayes}}(P_j^k)\right)^2$, which does not depend on $\theta$. Define the centered (Bayes-offset) empirical objective

$$\widehat{\mathcal{R}}(\theta) := \frac{1}{N} \sum_{j=1}^{N} \widetilde{\Lambda}_j(\theta), \quad \widetilde{\Lambda}_j(\theta) := A_j(\theta) + B_j(\theta).$$

Then $\arg\min_\theta \frac{1}{N} \sum_j \Lambda_j(\theta) = \arg\min_\theta \widehat{\mathcal{R}}(\theta)$, i.e., the ERM $\hat\theta$ is unchanged by the offset. Define the population counterpart $\mathcal{R}(\theta) := \mathbb{E}[\widetilde{\Lambda}_j(\theta)]$; using $\mathbb{E}[y - M_{\mathrm{Bayes}}(P) \mid P] = 0$,

$$\mathcal{R}(\theta) = \mathbb{E}\left[(M_{\mathrm{Bayes}}(P) - M_\theta(P))^2\right] = R_{\mathrm{BG}}(M_\theta).$$

Let $\theta^\star \in \arg\min_\theta \mathcal{R}(\theta)$. Then

$$R_{\mathrm{BG}}(M_{\hat\theta}) - R_{\mathrm{BG}}(M_{\theta^\star})$$
$$= \mathcal{R}(\hat\theta) - \mathcal{R}(\theta^\star)$$
$$= \mathcal{R}(\hat\theta) - \widehat{\mathcal{R}}(\hat\theta) + \widehat{\mathcal{R}}(\hat\theta) - \widehat{\mathcal{R}}(\theta^\star) + \widehat{\mathcal{R}}(\theta^\star) - \mathcal{R}(\theta^\star)$$
$$\leq \mathcal{R}(\hat\theta) - \widehat{\mathcal{R}}(\hat\theta) + \widehat{\mathcal{R}}(\theta^\star) - \mathcal{R}(\theta^\star) \quad (11)$$

and hence $\mathbb{E}[R_{\mathrm{BG}}(M_{\hat\theta}) - R_{\mathrm{BG}}(M_{\theta^\star})] \leq \mathbb{E}[\mathcal{R}(\hat\theta) - \widehat{\mathcal{R}}(\hat\theta)] \leq \mathbb{E}\left[\sup_\theta |\mathcal{R}(\theta) - \widehat{\mathcal{R}}(\theta)|\right]$.

**Step 2 (Localization at worst–path sequential radius).**
Let $h_\theta := M_\theta - M_{\mathrm{Bayes}}$. Fix a $\mathcal{Z}$–valued predictable tree $Z = (Z_k)_{k=1}^{p}$ of depth $p$ that is decoupled tangent to the prompt process, in the sense that for each depth $k$ and each past $\xi_{1:k-1} \in \{\pm 1\}^{k-1}$, the conditional distribution of $Z_k(\xi_{1:k-1})$ equals the conditional distribution of $P^k$ given $D^{k-1}$ (de la Peña & Giné, 1999; Rakhlin et al., 2015); namely $Z_k(\xi_{1:k-1}) \mid D^{k-1} \overset{d}{=} P^k \mid D^{k-1}$. Conditioning on a realization $Z = z$, we refer to such $z$ as a data-containing tangent tree. For any realization $z$ of $Z$, define the worst–path sequential $\ell_2$ radius on $z$ by

$$\|h\|_{\mathrm{seq},2;z} := \left\{ \sup_{\xi \in \{\pm 1\}^p} \frac{1}{p} \sum_{k=1}^{p} h\left(z_k(\xi_{1:k-1})\right)^2 \right\}^{1/2}.$$

For $r > 0$, we localize by the uniform worst–path radius

$$\mathcal{H}(r) := \left\{ h_\theta = M_\theta - M_{\mathrm{Bayes}} : \sup_z \|h_\theta\|_{\mathrm{seq},2;z} \leq r \right\}.$$

Then, for any $\theta$ such that $h_\theta \in \mathcal{H}(r)$, since $h_\theta$ is a bounded measurable function, $R_{\mathrm{BG}}(M_\theta) = \frac{1}{p} \sum_{k=1}^{p} \mathbb{E}_{P^k}\left[h_\theta\left(P^k\right)^2\right] = \mathbb{E}_{Z,\xi}\left[\frac{1}{p} \sum_{k=1}^{p} h_\theta\left(Z_k(\xi_{1:k-1})\right)^2\right] \leq \sup_z \|h_\theta\|_{\mathrm{seq},2;z}^2 \leq r^2$. Hence, $h_\theta \in \mathcal{H}(r)$ implies $R_{\mathrm{BG}}(M_\theta) \leq r^2$.

**Step 3 (High–probability envelope for the squared loss).**
Let $\delta := (pN)^{-2}$ and define the event

$$\mathcal{E} := \left\{ \max_{j \in [N], k \in [p]} |\varepsilon_{j,k+1}| \leq t_\delta \right\},$$

where $t_\delta := \sigma_\varepsilon \sqrt{2 \log\left(\frac{2pN}{\delta}\right)}$. By the sub-Gaussian tail bound and a union bound, $\Pr(\mathcal{E}^c) \leq \delta$. On $\mathcal{E}$, for every $(j,k)$ and every $\theta \in \Theta$ we have $|y_{j,k+1} - M_\theta(P_j^k)| \leq |f(\boldsymbol{x}_{j,k+1})| + |\varepsilon_{j,k+1}| + |M_\theta(P_j^k)| \leq B_f + t_\delta + B_M =: \widetilde{B}$, hence, using $\delta = (pN)^{-2}$,

$$\widetilde{B} = B_f + B_M + \sigma_\varepsilon \sqrt{2 \log\left(\frac{2pN}{\delta}\right)}$$
$$\leq B_f + B_M + \sigma_\varepsilon \sqrt{6 \log(2pN)}.$$

We first carry out the analysis on $\mathcal{E}$ (where the above envelope holds) and add a negligible $O(\delta)$ contribution to expectations in Step 7.

**Step 4 (Block symmetrization for the centered objective).**
We work directly with the centered blocks $\widetilde{\Lambda}_j(\theta) = A_j(\theta) + B_j(\theta)$ and their mean:

$$\sup_{\theta \in \Theta} \left|(\widehat{\mathcal{R}} - \mathcal{R})(\theta)\right| = \sup_{\theta \in \Theta} \left| \frac{1}{N} \sum_{j=1}^{N} \widetilde{\Lambda}_j(\theta) - \mathbb{E}\widetilde{\Lambda}_j(\theta) \right|.$$

Since $\widetilde{\Lambda}_1, \ldots, \widetilde{\Lambda}_N$ are i.i.d., standard symmetrization with Rademacher variables $(\epsilon_j)_{j=1}^{N}$, Cauchy–Schwarz inequality and Jensen inequality give $\mathbb{E}[\sup_\theta |(\widehat{\mathcal{R}} - \mathcal{R})(\theta)|] \leq \frac{C}{\sqrt{N}} (\mathbb{E} \sup_\theta \widetilde{\Lambda}_1(\theta)^2)^{1/2}$.

We decompose $\widetilde{\Lambda}_1(\theta) = A_1(\theta) + B_1(\theta)$. From the definition of $\mathcal{H}(r)$, $\mathbb{E}[\sup_{\theta \in \mathcal{H}(r)} A_1^2(\theta)]^{1/2} \leq r^2$. We then analyze $B_1^2(\theta) = \{\frac{2}{p} \sum_{k=1}^{p} (y_{1,k+1} - M_{\mathrm{Bayes}}(P_1^k))(M_{\mathrm{Bayes}}(P_1^k) - M_\theta(P_1^k))\}^2$. Note that $|M_{\mathrm{Bayes}}(P_1^k) - M_\theta(P_1^k)| \leq B_f + B_M$ and $B_1$ is constructed by a martingale difference sequence with filtration $\mathcal{G}_k'$. Since $\mathbb{E}[X^2] = 2 \int_0^\infty t \Pr(|X| > t) dt \leq 2 \int_{t_0}^\infty t \Pr(|X| > t) dt + 2 \int_0^{t_0} t dt$, evaluation of the tail probability from

Lemma 8 in (Rakhlin et al., 2015) yields

$$\mathbb{E}\left[\sup_\theta |B_1(\theta)|^2\right]^{1/2} \lesssim \widetilde{B} \log^3 p \, \mathfrak{R}_p^{\text{seq}}\left(\mathcal{H}(r)\right),$$

with the depth-$p$ sequential Rademacher complexity $\mathfrak{R}_p^{\text{seq}}(\mathcal{F}) := \sup_z \mathbb{E}_\xi[\sup_{f\in\mathcal{F}} \frac{1}{p}\sum_{t=1}^p \xi_t f(z_t(\xi_{1:t-1}))]$. Therefore

$$\mathbb{E}\sup_{\theta\in\mathcal{H}(r)}\left|(\widehat{\mathcal{R}} - \mathcal{R})(\theta)\right| \lesssim \frac{1}{\sqrt{N}}\left\{\log^3 p \, \mathfrak{R}_p^{\text{seq}}(\mathcal{H}(r)) + r^2\right\}.$$

**Step 5 (Sequential Dudley bound).** The sequential Dudley integral bound (Block et al., 2021, Corollary 10) gives, for an absolute constant $C > 0$,

$$\mathfrak{R}_p^{\text{seq}}\left(\mathcal{H}(r)\right)$$
$$\leq C \inf_{\alpha>0}\left\{\alpha + \frac{1}{\sqrt{p}}\int_\alpha^{D_r}\sup_z \sqrt{\log N'\left(\delta, \mathcal{H}(r); z\right)}d\delta\right\},$$

where $D_r$ denotes $\text{diam}(\mathcal{H}(r))$ and $N'$ denotes the fractional covering number (Block et al., 2021). Note that since every $h\in\mathcal{H}(r)$ satisfies $\|h\|_{\text{seq},2;z}\leq r$, the diameter under the path $\ell_2$ metric is at most $2r$, so the upper limit can be replaced by $2r$, from Lemma 7 in Block et al. (2021),

$$\mathfrak{R}_p^{\text{seq}}\left(\mathcal{H}(r)\right) \tag{12}$$
$$\leq C\inf_{\alpha>0}\left\{\alpha + \frac{1}{\sqrt{p}}\int_\alpha^{2r}\sup_z\sqrt{\log N_2^{\text{seq}}\left(\delta, \mathcal{H}(r); z\right)}d\delta\right\}.$$

From Lemma G.2, for universal constants $C_0, C_1 > 0$ and all $\delta \in (0, 2r]$, $\sup_z \log N_2^{\text{seq}}(\delta, \mathcal{H}(r); z) \leq C_0 m \log\left(\frac{\sqrt{m}}{\delta}\right) + C_1 p \log\left(\frac{1}{\delta}\right)$. Plugging this into (12) and optimizing over $\alpha$ absorbs polylogarithmic factors to give the succinct bound

$$\mathfrak{R}_p^{\text{seq}}\left(\mathcal{H}(r)\right) \lesssim r\frac{\sqrt{m+p}}{\sqrt{p}}\sqrt{\log\left(\frac{m}{r}\right)}.$$

**Step 6 (Self–bounding fixed point).**
Let $\Delta_\theta(P^k) := M_\theta(P^k) - M_{\text{Bayes}}(P^k)$, $\ell_\theta(P^k, y_{k+1}) := \{y_{k+1} - M_\theta(P^k)\}^2$, $\ell^{\text{Bayes}}(P^k, y_{k+1}) := \{y_{k+1} - M_{\text{Bayes}}(P^k)\}^2$. Then $R_{\text{BG}}(M_\theta) = \frac{1}{p}\sum_{k=1}^p \mathbb{E}[\Delta_\theta(P^k)^2]$ and

$$\ell_\theta - \ell^{\text{Bayes}} = \Delta_\theta^2 - 2\Delta_\theta\{y - M_{\text{Bayes}}(P^k)\}.$$

Hence $\mathbb{E}[\ell_\theta - \ell^{\text{Bayes}} \mid P^k] = \Delta_\theta^2$, and using $|\Delta_\theta| \leq B_M + B_f$ and $\mathbb{E}\{y - M_{\text{Bayes}}(P^k)\}^2 \leq C(B_f, B_M, \sigma_\varepsilon)$,

$$\mathbb{E}\left[(\ell_\theta - \ell^{\text{Bayes}})^2\right] \leq C_0 \mathbb{E}[\Delta_\theta^2] = C_0 R_{\text{BG}}(M_\theta),$$

that is, a Bernstein condition with exponent 1 for the excess loss holds.

For $r > 0$, set

$$\Theta(r) := \left\{\theta\in\Theta: R_{\text{BG}}(M_\theta) - \inf_{\vartheta\in\Theta}R_{\text{BG}}(M_\vartheta) \leq r\right\}.$$

By the standard symmetrization, we have $\mathbb{E}\left[\sup_{\theta\in\Theta(r)}\left|(\widehat{\mathcal{R}} - \mathcal{R})(\theta) - (\widehat{\mathcal{R}} - \mathcal{R})(\theta^\star)\right|\right] \lesssim \frac{1}{\sqrt{N}}\{\mathfrak{R}_p^{\text{seq}}(\mathcal{H}(r)) + \sqrt{r + R_{\text{BG}}(M_{\theta^\star})}\}$. Then, from Lemma 4 and Corollary 10 in Block et al. (2021), there exists a constant $c > 0$ (range rescaling absorbed into $c$) such that $\mathbb{E}\left[\sup_{\theta\in\Theta(r)}\left|(\widehat{\mathcal{R}} - \mathcal{R})(\theta) - (\widehat{\mathcal{R}} - \mathcal{R})(\theta^\star)\right|\right] \lesssim \sqrt{\frac{r + R_{\text{BG}}(M_{\theta^\star})}{N}}\left(1 + \sqrt{\log\frac{c_1}{r+R_{\text{BG}}(M_{\theta^\star})}} + \sqrt{\frac{m}{p}}\sqrt{\log\frac{c_2\sqrt{m}}{r+R_{\text{BG}}(M_{\theta^\star})}}\right)$.

By the basic inequality in (11), if $\mathbb{E}[\sup_{\theta\in\Theta(r)}|(\widehat{\mathcal{R}} - \mathcal{R})(\theta) - (\widehat{\mathcal{R}} - \mathcal{R})(\theta^\star)|] \leq \frac{r}{8} + cR_{\text{BG}}(M_{\theta^\star})$ with $c = o(1)$, then the ERM satisfies $R_{\text{BG}}(M_{\hat\theta}) - (1+c)R_{\text{BG}}(M_{\theta^\star}) \leq r/2$. Let the critical radius $r_\star$ be the smallest $r > 0$ solving $\frac{r}{8} \asymp \sqrt{\frac{r}{N}}(\sqrt{\frac{m}{p}} + 1)$. Hence $r_\star \asymp \frac{1}{N}(\frac{m}{p} + 1)$. Then, the ERM obeys $\mathbb{E}[R_{\text{BG}}(M_{\hat\theta})] \lesssim \inf_{\theta\in\Theta}R_{\text{BG}}(M_\theta) + \frac{1}{N}(\frac{m}{p} + 1)$.

**Step 7 (Control on $\mathcal{E}^c$).** Recall $\mathcal{E} := \{\max_{j,k}|\varepsilon_{j,k+1}| \leq t_\delta\}$ with $t_\delta := \sigma_\varepsilon\sqrt{2\log(2pN/\delta)}$. By sub-Gaussian tails and a union bound,

$$\Pr(\mathcal{E}^c) = \Pr\left(\exists(j,k): |\varepsilon_{j,k+1}| > t_\delta\right)$$
$$\leq 2pN\exp\left(-\frac{t_\delta^2}{2\sigma_\varepsilon^2}\right)$$
$$\leq \delta.$$

Let $\tilde{T} := \sup_{\theta\in\Theta}\left|(\widehat{\mathcal{R}} - \mathcal{R})(\theta)\right|$ and note that, by the identity $(y - M_\theta)^2 - (y - M_{\text{Bayes}})^2 = (M_{\text{Bayes}} - M_\theta)\{2y - M_\theta - M_{\text{Bayes}}\}$, Assumptions 2.4 and 2.5 imply a quadratic envelope of the form

$$\tilde{T} \leq C\left\{(B_f+B_M)^2 + \frac{1}{pN}\sum_{j,k}\varepsilon_{j,k+1}^2 + \mathbb{E}\varepsilon^2\right\}.$$

for some constant $C > 0$ (where $C, C', \dots$ below are universal constants). Thus,

$$\tilde{T} \leq C\left\{(B_f+B_M)^2 + \frac{1}{pN}\sum_{j,k}\varepsilon_{j,k+1}^2 + \sigma_\varepsilon^2\right\}. \tag{13}$$

To bound the expectation of $\tilde{T}$ on $\mathcal{E}^c$, we bound the tail of the second moment of each $\varepsilon$. For any $t > 0$, from the

sub-Gaussian ($\psi_2$) tail probability,

$$
\begin{aligned}
\mathbb{E}\left[\varepsilon^2 \mathbf{1}_{\{|\varepsilon|>t\}}\right] &= \int_t^\infty 2x \Pr(|\varepsilon|>x)\mathrm{d}x \\
&\leq 2 \int_t^\infty 2x \exp\left(-\frac{x^2}{2\sigma_\varepsilon^2}\right)\mathrm{d}x \\
&\leq 4\sigma_\varepsilon^2 \exp\left(-\frac{t^2}{2\sigma_\varepsilon^2}\right).
\end{aligned}
$$

Substituting $t = t_\delta$ yields $\mathbb{E}[\varepsilon^2 \mathbf{1}_{\{|\varepsilon|>t_\delta\}}] \leq 2\sigma_\varepsilon^2 \delta/(pN)$. Furthermore, by the decomposition $\varepsilon_{j,k+1}^2 \mathbf{1}_{\mathcal{E}^c} \leq \varepsilon_{j,k+1}^2 \mathbf{1}_{\{|\varepsilon_{j,k+1}|>t_\delta\}} + t_\delta^2 \mathbf{1}_{\mathcal{E}^c}$, it follows that

$$
\begin{aligned}
&\frac{1}{pN} \sum_{j,k} \mathbb{E}\left[\varepsilon_{j,k+1}^2 \mathbf{1}_{\mathcal{E}^c}\right] \\
&\leq \underbrace{\frac{1}{pN} \sum_{j,k} \mathbb{E}\left[\varepsilon_{j,k+1}^2 \mathbf{1}_{\{|\varepsilon_{j,k+1}|>t_\delta\}}\right]}_{\leq 2\sigma_\varepsilon^2 \delta/(pN)} + t_\delta^2 \Pr(\mathcal{E}^c). \quad (14)
\end{aligned}
$$

Combining (13) and (14), we get $\mathbb{E}\left[\tilde{T}\mathbf{1}_{\mathcal{E}^c}\right] \leq C\left\{(B_f+B_M)^2 + \sigma_\varepsilon^2\right\} \Pr(\mathcal{E}^c) + C'\left\{\sigma_\varepsilon^2 \delta + t_\delta^2 \Pr(\mathcal{E}^c)\right\}$. Substituting $t_\delta^2 = 2\sigma_\varepsilon^2 \log \frac{2pN}{\delta}$ and $\Pr(\mathcal{E}^c) \leq \delta$, we have $\mathbb{E}\left[\tilde{T}\mathbf{1}_{\mathcal{E}^c}\right] \leq C(B_f+B_M)^2\delta + C'\sigma_\varepsilon^2 \delta \log \frac{2pN}{\delta} + C''\sigma_\varepsilon^2\delta$. Finally, by using $\delta = (pN)^{-2}$, the right-hand side becomes $O\left(\sigma_\varepsilon^2(\log pN)/(pN)^2\right)$, which is negligible compared to the main term from Step 5. $\qquad\square$

