# OpenReview forum: "In-Context Learning Is Provably Bayesian Inference: A Generalization Theory for Meta-Learning"
_ICML.cc/2026/Conference — ICML 2026 regular_

### Official Review · Reviewer_xSVg · 2026-03-09

**Soundness:** 3
**Presentation:** 3
**Significance:** 3
**Originality:** 3
**Overall Recommendation:** 4
**Confidence:** 3

**Summary:**

This paper develops a finite-sample statistical theory within a meta-learning framework for in-context learning (ICL), by considering mixtures of diverse task types. The authors use a Bayes risk identity to separate the total ICL risk into two components: Bayes Gap and Posterior Variance, which are orthogonal to each other. By considering a uniform-attention Transformer, the authors derive an upper bound for this gap, which clarifies the dependence on the number of pretraining prompts and their context length. The authors derive that the Posterior Variance is determined solely by the difficulty of the true underlying task, while the uncertainty arising from the task mixture vanishes exponentially fast with only a few in-context examples. Finally, these results support that the uniform-attention Transformer selects the optimal meta-algorithm during pretraining and rapidly converges to the optimal algorithm for the true task at test time.

**Compliance With Llm Reviewing Policy:**

Affirmed.

**Final Justification:**

The writing of the paper is clear, and the derived theoretical results are solid. Although the theoretical framework has limitations, the derived results are somewhat significant in the area of ICL theory.

**Key Questions For Authors:**

See above

**Limitations:**

See above.

**Strengths And Weaknesses:**

The writing of the paper is clear, and the derived theoretical results are solid. The results are also significant in the area of ICL theory. The main weaknesses are regarding the assumptions of the system settings. Specifically,

1. On the right part of Line 155, the authors assume the decoder is uniformly Lipschitz. Given the definition of the decoder network, is there any justification for this assumption? I expect there are some theoretical results to derive this result under milder assumptions on W or b.

2. Another major concern is the uniform-attention Transformer. How to generalize this to more practical transformers?

---

> ### Author Rebuttal · Authors · 2026-03-31
>
> We thank you for the positive assessment of the paper’s technical quality and significance. We address the two questions below.
>
> ## Q1 (uniformly Lipschitz decoder assumption)
>
> Thank you very much for pointing this out. This can indeed be justified, and it is not intended as an additional assumption. For the ReLU decoder $\rho_{\theta}(z,c)=\rm{clip} (h_{\theta}(z,c))$, biases do not affect Lipschitz constants, ReLU is 1-Lipschitz, and clipping is also 1-Lipschitz. Hence the decoder Lipschitz constant is controlled by the spectral product already assumed in Definition 2.2:
> $$ \mathrm{Lip}(\rho_\theta)\le \mathrm{Lip}(h_\theta)\le S(\rho_\theta).$$
> Then, $|\rho_{\theta}(z,c) - \rho_{\theta}(z',c')|\le S(\rho_{\theta})\lVert (z, c)-(z', c')\rVert  \le S(\rho_{\theta}) (\lVert z-z' \rVert + \lVert c-c' \rVert),$
> so one can take $L_s=L_c = S(\rho_\theta)$. Thus, no extra restriction on the bias terms is needed; the displayed Lipschitz condition is simply an explicit unpacking of the spectral-product control already imposed on the decoder. We will revise the sentence as follows:
>
> > Note that by the definition of the decoder network and the clipping operation, the decoder $\rho_\theta$ satisfies the uniformly Lipschitz condition in both arguments: $|\rho_\theta(z,c) - \rho_\theta(z',c')| \le L_s\\|z - z'\\| + L_c\\|c - c'\\|$, where we can take $L_s = L_c  = S(\rho_\theta)$.
>
> ## Q2 (more practical Transformers)
>
> We agree that our current theory is not yet for arbitrary Transformers, and we will clarify this scope:
>
> **Model-agnostic:** Prop. 3.1, Thm. 3.3, and the Appendix C symmetrization / permutation-invariance results depend only on the exchangeable prompt model, not on the architecture.
>
> **Architecture-specific:** Thm. 3.2 is the only theorem tied to the uniform-attention class, since its proof uses a mean-pooled empirical-summary approximation of the Bayes predictor.
>
> **Intermediate:** Thm. 3.4 depends on the model only through regularity terms (e.g., prompt-Lipschitz constants), so it can in principle extend to other architectures once analogous bounds are established.
>
> Hence, a natural route to more practical Transformers is to keep Prop. 3.1 / Thm. 3.3 unchanged and replace only the approximation step in Thm. 3.2. Technically, this would require: (i) stability/Lipschitz bounds for multi-head softmax attention from operator-norm control of the Q/K/V/O maps, (ii) an approximation theorem showing that self-attention can represent the relevant context functionals, and then (iii) plugging these into the same sequential-learning-theoretic machinery [1,2]. We view this as an important direction for future work. We will make this explicit in the revised manuscript and briefly describe the technical route outlined above.
>
> ---
>
> We are grateful for the helpful feedback, and we would be happy to clarify any remaining concerns during the discussion period.
>
> ## References
>
> [1] Rakhlin, A., Sridharan, K., and Tewari, A. Online Learning via Sequential Complexities. *Journal of Machine Learning Research* 16:155–186, 2015.
>
> [2] Rakhlin, A., Sridharan, K., and Tewari, A. Sequential Complexities and Uniform Martingale Laws of Large Numbers. *Probability Theory and Related Fields* 161:111–153, 2015.

---

> > ### Author Rebuttal · Reviewer_xSVg · 2026-03-31
> >
> > I thank the reviewer for the detailed response. I will keep my current score.

---

> > > ### Author Response · Authors · 2026-04-04
> > >
> > > We thank you for your time and the discussion.

---

### Official Review · Reviewer_UrpP · 2026-03-12

**Soundness:** 4
**Presentation:** 3
**Significance:** 4
**Originality:** 4
**Overall Recommendation:** 5
**Confidence:** 3

**Summary:**

In-context learning (ICL)  is  a  phenomen of LLMs, where LLM (GPT3)  has been found to be able to adapt to new tasks from a  small number of training samples and without parameter
The submission investigates the ICL   by considering the ICL risk as the Bayes risk. The authors prove that the ICL risk (as formally defined by R(M) in the paper in section 2.3., M is a predictor of the next data point)  can be decomposed  into a sum of two terms:

R(M)= Bayes Gap + Posterior Variance

The Bayes Gap measures how closely the predictor model M, evaluated at a length k partial prompt, approximates the optimal Bayes predictor. Posterior Variance is independent of M and is irreducible. There are explicit upper bounds on the expectations of the empirical Bayes Gap  and of the  Posterior Variance.

The Bayes gap is studied empirically using a GPT2 model.

**Compliance With Llm Reviewing Policy:**

Affirmed.

**Key Questions For Authors:**

Q1:  This is perhaps more of a  comment:  There is quite a lot of terminology  that is not familiar for this reviewer. This  shows, of course,  a lack erudition. But is it necessary to  talk about 'prompts', 'task families', length p complete and partial  prompts ?  Are these standard in the literature on LLM or the authors' own language?   E.g.,   length p complete prompt is in statistical literature  simply a training set and a new sample.

Q2:  On p.6  after theorem 3.3  the authors state that   "This theorem  ... justifies the empirical observation that  ICL can quickly adapt to the specific task".  Again, maybe a limitation of  the reviewer, but this   does not seem quite pbvious. Also, what are the parameter up-dates invloved?

Q3: It is confusing/misleading that the exrensive set of assumptions needed for Theorem 3.3 is  presented  in an appendix  on p. 22. can this be revised?

**Limitations:**

Yes

**Strengths And Weaknesses:**

The submission is technically sound  and the results  claims well supported by theoretical analysis and  experimental results.   The paper includes theoretical results, and the proofs seem to be correct .  However, the assumption  of Hölder continuity of the Bayes  pedictor may not be valid in many models? Of course, the authors  provide in section E  good and elementary  examples of     Hölder continuity of theBayes  pedictor , but  for this additional technical assumptions are needed. Also,  the decoder of the transformer is assumed to  be uniformly  Lipschitz.

 The submission is  basically  clearly written and well structured.  The paragraph  at the bottom of right  column of page 3 starting with
"Averaging  simplex-valued ..."  somehow does not seem  properly placed.

 The paper addresses an important  and  relevant topic.   It advances understanding   in machine learning.  The decomposition of the ICL risk is intuitively natural.

 This  work provides new insights  and  deepens understanding of LLMs.

---

> ### Author Rebuttal · Authors · 2026-03-31
>
> We thank you for the careful reading and positive assessments of the paper’s importance and originality.
>
> ## Q1 (terminology)
>
> Thank you for pointing this out. Some of our terminology can be translated more directly into standard statistical language. We used the prompt-based wording to stay close to the ICL or LLM literature, where “prompt” and “in-context examples” are standard, while terms such as “complete prompt,” “partial prompt,” and “task family” are our labels for the corresponding statistical objects.
>
> In the final version, we will add short parenthetical clarifications or a brief footnote, e.g., complete prompt (= dataset together with a new input), partial prompt (= prefix of observed dataset together with the next input), and task family (= model/function class under a prior).
>
> ## Q2 (meaning of “adapt to the specific task”).
>
> We thank you for raising this question. This sentence might be too compressed. By “adapt,” we do **not** mean test-time parameter updates—there are none in our setting; the model parameters remain fixed after pretraining. Rather, Thm. 3.3 shows that the posterior over the latent task type internally concentrates fast on the true task family as the context length $k$ grows. Hence, the extra error caused by task-mixture ambiguity decays exponentially with context. This posterior-concentration / task-identification effect is the precise sense in which we meant that “ICL can quickly adapt to the specific task.” Combined with Thm. 3.2, this suggests that **even with its parameters fixed, the Transformer behaves as if it were internally constructing the Bayes posterior over task types, and its prediction progressively approaches the Bayes predictor for the true task family.**
>
> ## Q3 (Appendix F and Thm. 3.3).
>
> We agree that the presentation can be clearer here. To clarify, Appendix F on p. 22 does **not** introduce additional assumptions required by Thm. 3.3. The theorem in the main text already stands under its own abstract sequential conditions (see line 352 for an explanation). Appendix F is only a concrete illustrative instantiation, where we specialize Thm. 3.3 to the linear-vs-series regression mixture and verify its abstract conditions under transparent sufficient conditions. In the final version, we will make this explicit by adding a sentence such as:
>
> > “This appendix is illustrative rather than required for Thm. 3.3: it specializes the theorem to a concrete regression mixture and verifies its abstract conditions under transparent sufficient conditions.”
>
> ## Placement of the paragraph on p. 3.
>
> We agree that the paragraph beginning “Averaging simplex-valued ...” is better placed near the discussion of permutation invariance / empirical-measure representation, and we will move it accordingly in the final version.
>
> ---
>
> Thank you again for these very helpful comments. We would be happy to clarify any concerns or answer any questions that may come up during the discussion period.

---

> > ### Author Rebuttal · Reviewer_UrpP · 2026-04-07
> >
> > My questions have been satisfactorily  responded to.  I do not  revise my  grading.

---

> > > ### Author Response · Authors · 2026-04-08
> > >
> > > Thank you for the acknowledgement. We appreciate your careful reading and feedback.

---

### Official Review · Reviewer_fYNC · 2026-03-12

**Soundness:** 2
**Presentation:** 3
**Significance:** 2
**Originality:** 2
**Overall Recommendation:** 2
**Confidence:** 3

**Summary:**

In this paper, the authors study the ICL problem with Transformers. They decompose the total ICL error into two terms: (i) Bayes Gap (which captures model approximation error) and (ii) Posterior variance (which represents model-independent uncertainty)

The authors then establish bounds on the two errors, with non-asymptotic dependence on context length $p$ and number of prompts $N$.  (Theorem 3.2) The authors also try to bound the error in multi-task setup (Theorem 3.3) as well as distribution shift (Theorem 3.4).

**Compliance With Llm Reviewing Policy:**

Affirmed.

**Key Questions For Authors:**

Line 080: "the result suggests that uniform-attention Transformers select an optimal meta-algorithm during pretraining" What is the notion of optimality here? The authors did not seem to exploit any structure of the neural network, nor structure of the learning task. The only significant model assumption is a lipschitz assumption, which is well-known to give unrealistically bad bounds for deep neural networks. Can the authors elaborate on why their bound is optimal?

**Strengths And Weaknesses:**

## Strength
The proofs appear to be reasonable, though I did not check the appendix carefully.

## Weaknesses
- The paper claims to look at a transformer, but they do not actually look at a transformer. On line 157, the authors note "we adopt a specialized uniform-attention (Q=K=0) Transformer architecture." **Setting Q=K=0 removes the most essential aspect of the attention module**. Regardless of what the authors show for "uniform-attention", it most likely has no relevance for actual attention-basedd Transformers.
- The authors assume a "feature encoder network $\phi_\theta$" and a "decoder network $\rho_\theta$", which consist of multiple ReLU layers. Their construction requires that $\rho$ and $\phi$ learn to **conveniently learn to extract the exact features needed for their bayesian inference task**. (Line 982 below proposition C.6)
- The authors appeal to the fact that any feature map can be well approximated by a deep relu network (Yarotsky 2017) on line 1580

In summary, the proof in this paper
1. Has little to do with a real transformer, because their "uniform attention" is not attention.
2. Appeals very heavily on multi-layer ReLu networks being able to approximate just the right feature maps for Bayesian inference (via a result known to have very unrealistic sample and parameter complexity)
3. Applies very standard learning theory bounds based on lipschitz function classes.

Therefore I do not believe the results in this paper advance our understanding of how ICL is related to bayesian inference, nor provides useful complexity bounds.

I am not asking for a "polynomial rate" for deep learning models as that would be unreasonable. But at least, your theorem should take into account the Transformer architecture?

---

> ### Author Rebuttal · Authors · 2026-03-31
>
> We thank you for the careful reading. We agree that our wording should be sharpened, but we would also like to clarify more explicitly what is proved in this work.
>
> ## Q. What is the notion of optimality?
>
> In that sentence (line 80), “optimal” is meant in the **Bayes-risk** sense under squared loss, not as a claim of computational optimality or optimality among all Transformer architectures. Concretely, the *Bayes-optimal* in-context predictor is the posterior mean $$M_{\mathrm{Bayes}}(P^k)=\arg\min_M R(M),$$ and Prop. 3.1 decomposes the ICL risk into $ R(M)=R_{\mathrm{BG}}(M)+R_{\mathrm{PV}}.$
>
> Thm. 3.2 upper-bounds the Bayes Gap $R_{\mathrm{BG}}(M_{\hat\theta})$, i.e., how far the pretrained model (the empirical risk minimizer, ERM) is from this Bayes-optimal in-context predictor. Thus, our statement that “uniform-attention Transformers select an optimal meta-algorithm during pretraining” means that finite-sample pretraining drives the ERM toward the *Bayes-optimal* in-context predictor for the underlying meta-distribution. We agree that this wording is broader than necessary, and we will revise it to the more precise phrase: **“uniform-attention Transformers select a Bayes-optimal meta-algorithm during pretraining”**.
>
> ## On the three weakness points
>
> To clarify both our contribution and our limitation, let us also comment on the points raised under Weaknesses.
>
> ### 1. Why analyze uniform attention?
> Our choice of the uniform-attention subclass is principled in the exchangeable regime we study. Appendix C proves that: (i) permutation-averaging never increases risk, (ii) the Bayes predictor is permutation-invariant, and (iii) it factors through the empirical measure of the context. In this regime, a permutation-invariant / mean-pooled Transformer is therefore consistent with the symmetry of the Bayes-optimal predictor. In other words, in the i.i.d. or exchangeable setting, *uniform attention is sufficient to aggregate the context information relevant for the Bayes-optimal prediction.*
>
> We agree that this does not cover softmax Transformers. If order itself carries information, then non-uniform attention can indeed be necessary. We will make this limitation more explicit: our claim is a principled theorem for the i.i.d./exchangeable prompt regime, not for arbitrary order-sensitive prompts.
>
> ### 2. On “conveniently learn to extract the exact features”
>
> We respectfully believe this point should be interpreted differently. The ReLU networks are used here as a constructive approximation tool, and we do not assume that the needed features are given for free. Specifically, Lem. G.3 explicitly **constructs** the representation needed for Bayes prediction: a mollified partition-of-unity (“soft histogram”) encoder over the example domain, mean-pooled across context points, together with a decoder obtained via a McShane extension over a discrete Wasserstein metric. Then, the network size is chosen to match the smoothness of the Bayes predictor. Hence, the ReLU networks are used only as a **constructive realization** of this encoder/decoder pair.
>
> Moreover, Thm. 3.2 shows that ERM over this size-controlled Transformer class statistically yields a predictor with small Bayes Gap (the pretraining error). In this sense, the theorem *does* take the uniform-attention Transformer architecture into account: once this architecture class is specified, the learned predictor converges to the *Bayes-optimal* in-context predictor at an explicit rate. So the novelty is not “ReLU networks can approximate anything,” but “a constructive feasibility proof” plus “a statistical learning guarantee for this Transformer subclass with a moderate sample complexity” (see next comment). We will revise *Proof Idea* in line 280 to make our technical novelty clearer.
>
> ### 3. Why our bound is not merely a standard Lipschitz bound
>
> The generalization part (estimation error part) in Thm. 3.2 is also more specific than a black-box Lipschitz-class argument. Lem. G.2 decomposes the architecture into encoder and decoder, exploits the mean-pooled structure, and derives a **sequential covering number tailored to the uniform-attention Transformer class**. This is what produces the coupled $m/(pN)$ term in Thm. 3.2. Thus, the result is not merely a generic deep-net Lipschitz bound; it relies on the permutation-invariant, mean-pooled set-function structure of the model class.
>
> Overall, we agree that Thm. 3.2 does not yet cover full softmax attention, and we will sharpen that limitation. But we believe the contribution of Thm. 3.2 is **(1) to precisely identify an i.i.d. regime in which the Bayes predictor is permutation-invariant**, **(2) to prove that a corresponding Transformer subclass can statistically realize the required Bayes features,** and **(3) to derive explicit rates for ERM in that architecture class**.
>
> We hope this clarifies our intended scope and contribution, and we would be happy to clarify any further concerns during the discussion period.

---

> > ### Author Rebuttal · Reviewer_fYNC · 2026-04-04
> >
> > After rebuttal, I agree the authors have clarified scope, but my main concerns remain:
> >
> > 1. I appreciate the author's explanations about permutation-averaging being suitable for the problem they study. However at the end of the day the authors are still setting Q=K=0. My issue remains that any conclusions drawn here would be irrelevant to an actual Transformer, because *the entire advantage of a Transformer over something like MLP is learning inter-tokenn interactions via Q/K matrices.*
> >
> > 2. I agree that the authors' ReLU proof is constructive. However, we have known that even a 2-layer ReLU can approximate arbitrary functions, and one can indeed construct an explicit histogram/discretization based proof. My problem is that such a construction (a) has little relevance to how neural networks actually work, and (b) really does most of the heavy-lifting. If one is already appealing to the universal-approximation power of histograms, why not just use a 2-layer ReLU? Why bother sticking a uniform Transformer on top of that?
> >
> > 3. The style of proof in point 2 above This necessarily leads to a curse-of-dimensionality exponential dependence in effective dimension. This is a somewhat generic Holder-type rate, and is again of limited relevance to understanding how Transformers (or even general neural networks) work.
> >
> > I am aware that the other reviewers generally have a positive view of the results in this paper. I would discuss further with the AC and other reviewers before finalizing my own score. I would likely not be increasing my score to "weak accept", but I may be willing to increase to "weak reject".

---

> > > ### Author Response · Authors · 2026-04-04
> > >
> > > Thank you again for the thoughtful follow-up. We would like to clarify that **the uniform-attention mechanism plays an essential role in controlling the dependence on the context length $p$ and thereby in improving the pretraining generalization error rate in Thm 3.2**, which consists of both approximation and estimation errors. We will elaborate on this point below.
> > >
> > > ### **2. Why not just use a 2-layer ReLU? Why use a uniform Transformer?**
> > >
> > > Thank you for raising important questions. In one sentence, the reason is that **the uniform attention leads to improved pretraining accuracy** as the context length $p$ increases.
> > >
> > > * If one naively uses a 2-layer ReLU network and feeds the entire prompt $P=(x_1,y_1,\ldots,x_p,y_p,x_{p+1})$ as input, then the input dimension becomes $p(d_{\mathrm{feat}}+1)+d_{\mathrm{feat}}$, which leads to a curse of dimensionality in the context length $p$ (i.e., the number of examples).
> > >
> > > * In contrast, even with $Q=K=0$, a uniform attention mechanism (encoder $\to$ mean-pool $\to$ decoder) compresses an arbitrary-length context into a fixed-dimensional summary statistic. This is why the rate in Thm. 3.2 does not deteriorate as $p$ increases, but rather improves.
> > >
> > > The key takeaway is that the attention not only ***enables a rate that avoids the curse of dimensionality*** with respect to the context length $p$, but also ***improves the effective sample size***, yielding the pretraining generalization bound of order $(pN)^{-\frac{d_{\mathrm{eff}}}{d_{\mathrm{eff}}+2\alpha}}$ in Thm.3.2. This illustrates an advantage of the uniform-attention model over a 2-layer ReLU network.
> > >
> > > ### **3. This necessarily leads to a curse-of-dimensionality exponential dependence in effective dimension. This is a somewhat generic Holder-type rate, and is again of limited relevance to understanding how Transformers (or even general neural networks) work.**
> > >
> > > It is important to note that this dependence is on the dimension $d_{\mathrm{feat}}+1$ of a single token, not on the full prompt dimension $p(d_{\mathrm{feat}}+1)+d_{\mathrm{feat}}$. In our exchangeable-prompt setting, **uniform attention avoids the curse of dimensionality** with respect to the context length $p$. We would like to emphasize that **this is a significant difference between Transformers and 2-layer NNs**: Transformers achieve better accuracy as $p$ increases, while 2-layer NNs suffer from a severe curse of dimensionality as $p$ grows (even polynomial-order rates would be lost).
> > >
> > > Moreover, we derive a Hölder-type rate that jointly depends on $p$ and $N$. As discussed in lines 291--295 of the paper, this is a rate that prior works did not achieve. We were able to obtain it because of:
> > > - **the usage of uniform attention**, which removes the dependence on context length, and
> > > - technical novelties in the generalization bound for the encoder and decoder, namely **a Wasserstein-metric-based approximation technique that does not depend on context length**, together with **a sequential-complexity-based estimation-error analysis.**
> > >
> > > If your concern is that “this can be viewed as a classical Hölder rate and hence not novel,” we believe this reflects an intrinsic difficulty of the problem, rather than a limitation of the architecture or the proof technique. At least for ICL pretraining, as explained above, there are contributions both in the rate itself and in the proof techniques.
> > >
> > > ### **1. any conclusions drawn here would be irrelevant to an actual Transformer, because the entire advantage of a Transformer over something like MLP is learning inter-token interactions via Q/K matrices.**
> > >
> > > We agree that inter-token interactions induced by the $Q/K$ matrices are a critical property of attention. However, in light of the above discussion, even when $Q=K=0$, we find that *the uniform attention model still has an advantage in that it effectively controls dependence on context length $p$.*
> > >
> > > ---
> > >
> > > For these reasons, **the uniform-attention structure plays a key role in avoiding the curse of dimensionality** with respect to the context length $p$, and in **improving the effective sample size** for **Thm. 3.2 (pretraining generalization bound)**. Together with Thm. 3.3 (ICL’s inference-time behavior) and Thm. 3.4 (effect of OOD shift), we believe this work provides concrete contributions to the statistical understanding of ICL that are absent from prior work.
> > >
> > > We appreciate this important follow-up comment.

---

### Official Review · Reviewer_FGNG · 2026-03-18

**Soundness:** 3
**Presentation:** 4
**Significance:** 3
**Originality:** 2
**Overall Recommendation:** 5
**Confidence:** 4

**Summary:**

This paper frames in-context learning as Bayesian inference within a meta-learning framework and develops a finite-sample statistical theory for it.
Following recent related papers, which already frame ICL as Bayesian inference, this paper seeks to accomodate heterogenous task types and to analyze the relationship of pretraining size and prompt length.
The authors propose and build on a risk identity that decomposes the ICL risk into two independent terms, the Bayes gap and the posterior variance.
The Bayes gap measures the excess risk of the pretrained model relative to the Bayes-optimal in-context predictor, which they define as the posterior mean given the context.
The posterior variance is model-independent and represents the uncertainty of the Bayes predictor itself. It can only be reduced with more in-context examples.
The authors use a simplified transformer architecture with mean pooling (uniform attention) instead of the regular attention mechanism in their analysis.

The theory is studied under a mixture of task types regression model, where each prompt is generated by first sampling a task family, then a specific function from that family, and the i.i.d. input-label examples for a context plus a query. The authors use a simplified uniform attention (mean-pooling) transformer as the model, rather than a model with standard attention.

The three contributions are a theorem which provides non-asymptotic upper bounds that couple the number of pretraining prompts $N$ and their context length $p$, a theorem which explains in-context error via the test-task difficulty, and a theorem which characterizes stability under input-distribution shift.

**Compliance With Llm Reviewing Policy:**

Affirmed.

**Final Justification:**

The paper is technically solid and makes an interesting contribution to the theory of ICL. The rebuttal and reply rebuttal addressed my concerns well and added some quantitative empirical support to the paper's main claim. I accept the absence of formal results for standard attention as a limitation and possible direction for future work, rather than a flaw. The authors have committed to making the scope and limitations explicit in the camera ready version.

**Key Questions For Authors:**

* Can you provide any formal result (even partial) for standard attention or can you characterize how the theory extends to standard attention or which barriers prevent said extension?
* How does the bound for the posterior variance scale as $T$ grows? Is it still a useful bound as $T\to\infty$ or does the utility of the framework break down?
* Can you provide a quantitative comparison of the experimental observations with the theoretical predictions? Testing the tightness of the bounds rather than just qualitative agreement would mnake them significantly more useful.
* You mention that Ma et al. (2025) have "independently and concurrently" analyzed ICL adaptivity from a similar standpoint. Can you elaborate on the specific similarities and differences? This seems relevant to asses the novelty and magnitude of your contribution.
* What are the practical implications of your work?

**Limitations:**

yes

**Strengths And Weaknesses:**

**Strengths**
* Although quite simple, the decomposition of the ICL risk into a component which pretraining can improve and another component which only more context can improve, is a useful perspective.
* Rigorous and sound mathematical construction throughout the paper.
* Clear language and structured presentation. The progression from setup to risk identity to analysis of each term is logical and easy to follow and the paper sometimes provides helpful intuition alongside formalisms.
* The papers has interesting and novel contributions. Providing bounds for each term of the risk decomposition and analyzing the OOD stability are interesting and novel as far as I know.
* Related work is comprehensively cited and similarities are (mostly) acknowledged.

**Weaknesses**
* The theoretical results apply to a uniform-attention/mean-pooling transformer, which is substantially different from the standard attention layers used in practice. While the authors justify this choice in the appendix via the permutation invariance of the Bayes predictor, there is still a significant argumentative gap. The paper does not provide theoretical results for standard attention or a proof that the mean pooling results hold for standard attention. The experimental results on GPT2 are encouraging but do not fully close the gap.
* The experimental scope is quite limited. Experiments are performed on a single synthetic dataset consisting of a two-family mixture. I assume this setup is chosen to have exact Bayes baselines, but surely a more challenging or realistic scenario could have been chosen under this constraint.
* The experiments "support the main qualitative predictions of the theory" but there is no evaluation of whether the theory and experiments also align quantitatively. Since the central contribution of this paper are bounds for the different terms, the tightness of these bounds seems very relevant.
* The novelty of the central proposition (Proposition 3.1) which the authors build on is somewhat limited. Decomposing the risk into a Bayes gap and a posterior variance is well known in sstandard statistical learning theory. What is new is only the application to the ICL setting and bounding each term.
* Ma et al. (2025) provide similar decomposition and bounds (see e.g. their Proposition 1). This leads me to question the novelty of this paper's contributions.
* The assumption of a small number $T$ of task families is quite restrictive and unrealistic. The paper does not discuss how the results scale with $T$ or whether the framework holds for an effectively infinite number of tasks types encountered in real settings.
* Minor point for readability: the expected value as simply \mathbb{E} is sometimes ugly / difficult to read when there is a symbol immediately following it without a space. Consider defining a math operator, manually inserting a small space after \mathbb{E} or using parenthesis around the argument.

---

> ### Author Rebuttal · Authors · 2026-03-31
>
> We thank you for your positive assessment of our contributions and your constructive feedback.
>
> ## Q1 (extension to standard attention)
>
> We agree that the scope should be stated more explicitly. We first clarify the scope of the theorems:
>
> **Model-agnostic:** Prop. 3.1, Thm. 3.3, and the Appendix C symmetrization / permutation-invariance results depend only on the exchangeable prompt model, not on the architecture.
>
> **Architecture-specific:** Thm. 3.2 is the only theorem tied to the uniform-attention class, because its proof uses a mean-pooled empirical-summary approximation of the Bayes predictor.
>
> **Intermediate:** Thm. 3.4 depends on the model only through regularity terms (e.g., prompt-Lipschitz constants), so it can in principle extend once analogous bounds are established for another architecture.
>
> Thus, the part that does not currently transfer directly to a standard softmax-attention Transformer is Thm. 3.2. The barrier to extending Thm. 3.2 to standard softmax attention is that attention weights are prompt-dependent and couple all context tokens through logits and normalization, making the effective class data-dependent. A plausible route is to combine our symmetry argument with sequential covering / sequential Rademacher tools [1,2] for data-dependent compositional classes, together with Lipschitz control of softmax attention.
>
> We believe that this is a key direction for future work. We will make this more explicit in the revised manuscript and briefly describe the technical route.
>
> ## Q2 (T-dependence)
>
> For finite $T$, Thm. 3.3 already implies only logarithmic dependence on $T$:
> $$
> 5B_f^2\Big(\frac{1-\alpha_{i^\star}}{\alpha_{i^\star}}e^{-D_{\min}k/2}+(T-1)e^{-Ck}\Big).
> $$
> Hence the context length needed to keep the identification term small grows as $O(\log T)$.
>
> We believe this viewpoint extends beyond finite $T$: the factor $(T-1)$ can be replaced by a complexity measure of task variety, such as a covering number or a prior-mass term over the task space. We will add this discussion.
>
> ## Q3 (quantitative comparison)
>
> We agree and will strengthen this point. The synthetic setup was chosen because it provides exact Bayes and oracle baselines, so the Bayes Gap can be measured directly. In the revision we will add a more explicit quantitative comparison to the predicted $p$-$N$ pretraining scaling (e.g., a log-log plot / discussion of the slope), while keeping the claim modest because the theorem is for the uniform-attention class and the GPT-2 experiment serves as external qualitative validation.
>
>
> ## Q4 (Prop. 3.1 and Ma et al. [3])
>
> We agree that Prop. 3.1 itself is not the main novelty; we will sharpen this in Section 1. Our novelty is the three non-asymptotic results (Thm. 3.2-3.4) built on top of it.
>
> On the comparison to [3], the two works are complementary. [3] studies softmax-attention Transformers and a $\chi^2$-robust excess-risk bound for mixtures of difficulty levels from a Bayesian view. Our paper restricts the architecture, but adds: (a) an explicit **finite-sample $p$-$N$ pretraining scaling** theorem (Thm. 3.2), (b) a reduction of test-time risk to the true-family minimax risk plus **an exponentially decaying task-identification term** (Thm. 3.3), and (c) **a prompt-level Wasserstein OOD theorem** (Thm. 3.4).
>
> ## Q5 (practical implications)
>
> Thank you for the question. Our theory offers several practical insights.
>
> | Topic | Implication |
> |---|---|
> | Inference-time prompts | Very long prompts are not always necessary, but the model needs a few examples to identify the task (about three in our experiments; right panel of Fig. 2). For harder problems, more context helps insofar as it reduces the true-family minimax risk (Thm. 3.3). |
> | Distribution shift | Match the test input distribution to pretraining when possible. Thm. 3.4 shows that shift primarily degrades the Bayes Gap. Under moderate unavoidable shift, controlling Transformer Lipschitz constants (e.g., spectral normalization / output clipping) and then increasing $p,N$ is a more principled strategy. |
> | Benchmark design | Small-$k$ benchmarks mainly probe task identification, while larger-$k$ evaluations test learning within the identified family. |
>
> As these are spread across the main text, we will consolidate them into a dedicated discussion.
>
> ## Additional comment:
> We have fixed the notation/readability issue around $\mathbb{E}$.
>
> ---
>
> We would be happy to clarify any further questions during the discussion period.
>
> ## References
> [1] Rakhlin, A., Sridharan, K., and Tewari, A. Online Learning via Sequential Complexities. *Journal of Machine Learning Research* 16:155–186, 2015.
>
> [2] Rakhlin, A., Sridharan, K., and Tewari, A. Sequential Complexities and Uniform Martingale Laws of Large Numbers. *Probability Theory and Related Fields* 161:111–153, 2015.
>
> [3] Ma, T., Wang, T., and Samworth, R. J. Provable test-time adaptivity and distributional robustness of in-context learning. 2025. URL https://arxiv.org/abs/2510.23254v1.

---

> > ### Author Rebuttal · Reviewer_FGNG · 2026-04-03
> >
> > Thank you for your thorough and constructive rebuttal. Several of my concerns have been addressed well. I have some remaining questions:
> >
> > - The clarification that Prop. 3.1, Thm. 3.3 are model-agnostic is helpful. That said, Thm. 3.2 is arguably your most relevant result, and it remains tied to uniform attention. If this is made clear to the reader, I can accept this as a limitation rather than a flaw. Could you please briefly sketch how you plan to "make this more explicit in the revised manuscript and briefly describe the technical route"?
> > - The point that the experimental scope is quite limited has not been addressed yet. Can you provide results on more realistic/challenging scenarios?
> > - I appreciate the promise of a quantitative comparison with log-log plots in revision. Can you provide any preliminary (or completed) comparison during the discussion period? Your promise seems plausible, but seeing actual results is always a bit more convincing.

---

> > > ### Author Response · Authors · 2026-04-05
> > >
> > > Thank you for your thoughtful follow-up questions.
> > > ### Q1. Could you sketch how you plan to make this more explicit and briefly describe the technical route?
> > > We will make this clarification in three places.
> > >
> > > Immediately before Thm. 3.2, we will add:
> > >
> > > > The next result is architecture-specific and applies to the uniform-attention class in Definition 2.2. Its proof relies on the mean-pooling approximation of the Bayes predictor developed in Appendix C.
> > >
> > > At the end of Section 3 (theory section), we will add a dedicated paragraph to clarify the scope of the results, distinguishing the *model-agnostic*, *architecture-specific*, and *intermediate parts* as noted in our previous response.
> > >
> > > Moreover, in the limitations paragraph, we will replace the current high-level statement with the following more explicit text:
> > >
> > > > Our finite-sample pretraining bound (Theorem 3.2) is proved only for the uniform-attention class. The main obstacle to extending this result to standard softmax attention is that the attention weights are prompt-dependent and couple all context tokens through logits and normalization, so the effective hypothesis class becomes data-dependent. A plausible route is to treat softmax attention as a data-dependent weighted empirical summary and combine Lipschitz control of the attention map with sequential covering / sequential Rademacher bounds for this compositional class.
> > >
> > > We believe this makes the scope and limitation clear to the reader.
> > >
> > > ### Q2. Can you provide results on more realistic/challenging scenarios?
> > > To address this concern, we performed a synthetic but more challenging experiment. Specifically, we replaced the original 2-family mixture by a 3-family mixture consisting of (i) linear regression, (ii) tanh-feature regression, and (iii) odd-quadratic-feature regression ($\phi(x)=x|x|$). We further introduced a difficulty sweep by increasing the observation noise over $\sigma\in\\{0.1,0.3,0.5\\}$. This keeps exact Bayes baselines available while making both task-family identification and within-family prediction more challenging.
> > >
> > > | Noise $\sigma$ | Wrong-family mass @ $k=2$ | Wrong-family mass @ $k=4$ | Transformer / Bayes(mix) / Bayes(oracle) MSE @ $k=1$ | Transformer / Bayes(mix) / Bayes(oracle) MSE @ $k=4$ | Transformer / Bayes(mix) / Bayes(oracle) MSE @ $k=10$ |
> > > |---|---:|---:|---:|---:|---:|
> > > | 0.1 | 0.067 | 0.0010 | 19.15 / 18.027 / 4.063 | 1.814 / 1.738 / 1.136 | 0.156 / 0.109 / 0.013 |
> > > | 0.3 | 0.080 | 0.0025 | 23.27 / 22.737 / 4.773 | 3.135 / 2.879 / 2.110 | 0.282 / 0.224 / 0.152 |
> > > | 0.5 | 0.093 | 0.0054 | 23.94 / 22.426 / 6.124 | 3.291 / 2.908 / 2.364 | 0.338 / 0.317 / 0.314 |
> > >
> > > The results show two clear patterns.
> > >
> > > - First, as difficulty increases $(\sigma \uparrow)$, the **posterior mass outside the true family (wrong-family mass) decays more slowly** at the same context length $k$, indicating that harder settings require more in-context examples for task identification.
> > > - Second, the **Transformer exhibits the same qualitative behavior as the Bayes-mixture predictor**: prediction error drops rapidly in the first few in-context examples, and the Transformer quickly approaches the Bayes oracle. This indicates that **the Transformer quickly identifies the true task** after only a few examples, after which the remaining error is largely the task-specific intrinsic error.
> > >
> > > Overall, the main qualitative mechanism predicted by Thm. 3.3 persists in the harder setting and under a difficulty sweep.
> > > ### Q3. Can you provide any preliminary comparison during the discussion period?
> > > We provide a quantitative comparison here.
> > >
> > > **1. Power-law fit.**
> > > First, for each fixed-$p$ and fixed-$N$ sweep results (the $N$-sweeps in the left panel and the $p$-sweeps in the middle panel of Figure 2), a power law fits the (empirical) Bayes gap well:
> > > $$ \mathrm{BG}(N;p)\approx a_p+b_pN^{-\beta_p}, \qquad \mathrm{BG}(p;N)\approx a_N+c_Np^{-\gamma_N}. $$
> > > Across the fixed-$p$ sweeps, the fits have coefficients of determination $R^2=0.96$-$0.98$. Across the fixed-$N$ sweeps, the fits have $R^2=0.94$-$0.99$. Thus, the results show a power-law-like decay, consistent with Thm. 3.2.
> > >
> > > **2. $p$-$N$ joint dependence.**
> > > Second, using the pooled data from both sweeps (i.e., all points from the left and middle panels of Figure 2), we fit a $p$-$N$ coupled model motivated by Thm. 3.2:
> > > $$\mathrm{BG}(p,N)\approx a+b(pN)^{-\beta}+\frac{c}{N}.$$
> > > On the pooled dataset, this model fits the empirical Bayes-gap values well and outperforms $N$-only and $p$-only surrogates:
> > > | Model | Surrogate | $R^2$ |
> > > |---|---|---:|
> > > | joint | $a+b(pN)^{-\beta}+c/N$ | 0.912 |
> > > | N-only | $a+bN^{-\beta}$ | 0.030 |
> > > | p-only | $a+cp^{-\gamma}$ | 0.016 |
> > >
> > > The takeaway is that the coupled $p$-$N$ power-law dependence is quantitatively visible in the results. We will present the findings as order-level support for Thm. 3.2, rather than as a claim of tight constants.
> > >
> > > ---
> > > We appreciate your questions and suggestions; they have helped us improve this work.

---

### Decision · Program_Chairs · 2026-04-30

**Decision:**

Accept (regular)

**Comment:**

The manuscript received four reviewers, which identified strengths but also some weaknesses. In particular, the reviewers appreciated the effort of a formal treatment of in-context-learning in a Bayesian context and the quality of presentation. Aspects of concerns were the restriction to uniform-attention transformers, and uncertainty about the specifics of the contribution, given that the decomposition in Prop. 3.1 is not specific to the studied situation. The authors provided a response and engaged in discussion, in which the justified their choices and clarified some aspects. They also added some additional empirical evidence. After the discussion phase, three of the reviewers recommended to accept the work, while one reviewer remained critical. Ultimately, I believe the merits of the submission outweigh the limitations, so I recommend acceptance.